# Analyzing Generalization of Neural Networks through Loss Path Kernels

**Yilan Chen**
UCSD CSE
yilan@ucsd.edu

**Wei Huang**
RIEKN AIP
wei.huang.vr@riken.jp

**Hao Wang**
MIT-IBM Watson AI Lab
hao@ibm.com

**Charlotte Loh**
MIT EECS
cloh@mit.edu

**Akash Srivastava**
MIT-IBM Watson AI Lab
akash.srivastava@ibm.com

**Lam M. Nguyen**
IBM Research
LamNguyen.MLTD@ibm.com

**Tsui-Wei Weng**[*]
UCSD HDSI
lweng@ucsd.edu

## Abstract

Deep neural networks have been increasingly used in real-world applications, making it critical to ensure their ability to adapt to new, unseen data. In this paper, we study the generalization capability of neural networks trained with (stochastic) gradient flow. We establish a new connection between the loss dynamics of gradient flow and general kernel machines by proposing a new kernel, called loss path kernel. This kernel measures the similarity between two data points by evaluating the agreement between loss gradients along the path determined by the gradient flow. Based on this connection, we derive a new generalization upper bound that applies to general neural network architectures. This new bound is tight and strongly correlated with the true generalization error. We apply our results to guide the design of neural architecture search (NAS) and demonstrate favorable performance compared with state-of-the-art NAS algorithms through numerical experiments.

## 1 Introduction

Deep learning models have been increasingly used in applications with significant societal impact. Therefore, it is crucial to ensure that these models perform well not only on the training data but also on the new and unseen data. Classical learning theory attributes the generalization ability of machine learning (ML) models to the small complexity of the hypothesis class [56]. However, modern ML models, such as deep neural networks (NNs), can have billions of parameters yet still exhibit strong generalization abilities [26, 9]. This is because various elements of the learning algorithms, including optimization methods, training data, and neural architectures, can all influence the inductive bias, which in turn shapes the generalization abilities of neural networks [29, 40]. While the overall hypothesis class may be large, the "effective domain" of this class, which ultimately determines the model's generalization abilities, is often much smaller [54, 24, 50, 15]. Hence, it is vital to develop algorithmic generalization bounds to capture this effective domain of the hypothesis class.

There has been significant work investigating the generalization of neural networks in their infinite-width regime through kernel methods [28, 3, 32, 2, 11]. They showed that an infinite-width NN

---

[*]Correspondence to: Yilan Chen and Tsui-Wei Weng.

37th Conference on Neural Information Processing Systems (NeurIPS 2023).

Table 1: Comparison with existing NTK-based generalization bounds. $\mathbf{\Theta} = \Theta(\mathbf{X}, \mathbf{X})$ is the NTK on training samples and $\mathbf{H}^\infty$ is the NTK of the first layer. $L$ represents the number of NN layers. We highlight some unique properties of our results in blue color. "During training" means our bound can be calculated at any time during training while existing NTK-based bounds only hold for the NNs at convergence. "Multi-outputs" means our bound holds for NNs with multiple outputs.

|  | Arora et al. [2] | Cao & Gu [11] | **Ours** |
|---|---|---|---|
| Bound | $\sqrt{\frac{2\mathbf{Y}^\top(\mathbf{H}^\infty)^{-1}\mathbf{Y}}{n}}$ | $\tilde{O}(L \cdot \sqrt{\frac{\mathbf{Y}^\top(\mathbf{\Theta})^{-1}\mathbf{Y}}{n}})$ | Theorem 3, Theorem 5 |
| Model | Ultra-wide two-layer FCNN | Ultra-wide FCNN | **General continuously differentiable NN** |
| Data | i.i.d. data with $\|\boldsymbol{x}\| = 1$ | i.i.d. data with $\|\boldsymbol{x}\| = 1$ | i.i.d. data |
| Loss | Square loss | Logistic loss | **Continuously differentiable & bounded loss** |
| During training | No | No | **Yes** |
| Multi-outputs | No | No | **Yes** |
| Training algorithm | GD | SGD | (Stochastic) gradient flow |

trained by gradient flow and squared loss is equivalent to a kernel regression with neural tangent kernel (NTK) [28, 3]. Moreover, Arora et al. [2], Cao & Gu [11] further characterized the generalization behaviors of such ultra-wide NNs by deriving data-dependent and NTK-based generalization bounds. However, they only considered ultra-wide fully connected NNs with a square (or logistic) loss function. In practice, NNs are usually not ultra-wide and have more complex architectures such as CNNs. Hence, it is crucial to establish generalization bounds that hold in a more general setting.

In this paper, we analyze the generalization capability of NNs trained using (stochastic) gradient flow across a wide range of NN architectures. Our key technical contribution is to establish a new connection between the loss dynamics of (stochastic) gradient flow and general kernel machines with a special kernel that we named the *loss path kernel*. This new kernel calculates the similarity between two data points by integrating the inner product of the loss gradient evaluated at these points, along the path determined by the gradient flow. Based on this connection, we develop a novel generalization bound by analyzing the complexity of kernel machines induced by various training sets. Our generalization bound is tight and can be applied to a broad class of NN architectures, not restricted to ultra-wide NNs (see Table 1 for a comparison with existing results derived from NTK theory). Numerical experiments demonstrate that our bound maintains a strong correlation with the true generalization error of NNs trained with gradient descent (GD) (see Figure 1 and 2 in Sec. 6 & 7). Given this observation, we use our generalization bound to guide the design of neural architecture search (NAS) and demonstrate through numerical experiments that our approach can achieve a favorable performance compared with state-of-the-art training-free and minimum-training NAS algorithms [37, 13, 39]. In summary, our contributions are:

- In Sec. 4.1 and 5, we show for the first time that the loss of NNs trained by (stochastic) gradient flow is equivalent to a *general* kernel machine. This result enables us to investigate the generalization capability of NNs from the perspective of kernel theory.

- In Sec. 4.2 and 5, we derive tight generalization bounds for NNs based on the aforementioned equivalence. Our result is very general as it holds for any continuously differentiable NN architectures including finite-width and infinite-width NNs. Experiments demonstrate that our bounds are tight ($4690\times$ tighter than existing norm-based bounds and $55\times$ tighter than existing NTK-based bounds as shown in Appendix A.2) and highly correlated with the true generalization error.

- In Sec. 6 and 7, we apply our theory to study special cases including infinite-width NNs, stable algorithms, and norm-constrained NNs. We apply our bound to guide the design of NAS. Numerical experiments demonstrate that our approach achieves a favorable performance compared with state-of-the-art NAS algorithms on NAS-Bench-201 benchmark [19].

## 2 Related Work

**Generalization theory in deep learning.** Generalization is a crucial aspect of deep learning theory and various techniques have been proposed to study it. For example, Bartlett et al. [7] derived tight bounds for the VC dimension of NNs with ReLU activation functions. There is also a line of work that measures the capacity of NNs based on different norms, margins [5, 41, 6, 44], and sharpness-based measures [42, 43, 1] to explain the generalization behaviors of NNs. Additionally, there are theories studying the generalization of NNs from PAC-Bayes [35, 22] and information-theoretical approach

[48, 58]. For example, Dziugaite & Roy [22] numerically evaluated and optimized the PAC-Bayes bound of stochastic NN and obtained a non-vacuous generalization bound. In contrast, we study the generalization of NNs by building a new connection between NNs and kernel machines. We refer readers to Valle-Pérez & Louis [55] for a more comprehensive review of the generalization theory of NNs.

**Neural tangent kernel (NTK).** NTK was first introduced in Jacot et al. [28], where the authors demonstrated that a fully-trained, infinite-width NN follows kernel gradient descent in the function space with respect to the NTK. Under gradient flow and squared loss, the fully-trained infinite-width NN is equivalent to kernel regression with the NTK [28, 3]. Chen et al. [14] further established the equivalence between infinite-width NNs and regularized kernel machines. Arora et al. [2] studied the generalization capacity of ultra-wide, two-layer NNs trained by GD and square loss, while Cao & Gu [11] examined the generalization of deep, ultra-wide NNs trained by stochastic gradient descent (SGD) and logistic loss. Both studies derived generalization bounds of converged NNs based on NTK. Besides, Huang et al. [27] studied the convergence and generalization of PAC-Bayesian learning for deep, ultra-wide NNs. Later, Domingos [17] showed that every model trained by gradient flow is a "kernel machine" with the weights and bias as functions of input data, which however can be much more complex than a typical kernel machine and our general kernel machine in Definition 2. Chen et al. [14] showed that every NN trained by gradient flow is a general kernel machine but their kernel is valid only in very limited cases – when the loss gradient of output is a constant. Otherwise, the kernel is not symmetric and not valid. In this paper, we consider an equivalence between the loss of NNs and general kernel machines, which resolves the previous asymmetric problem of the kernel function and also makes the generalization analysis of multi-outputs easier.

**Neural Architecture Search (NAS).** NAS aims to automate the discovery of top-performance neural networks to reduce human efforts. However, most existing NAS algorithms require heavy training of a supernet or intensive architecture evaluations, suffering from heavy resource consumption [47, 34, 18, 33]. Thus, it is crucial to develop training-free or minimum-training NAS algorithms to reduce the computational cost and select the best architecture at the same time [37, 13, 39]. Since our generalization bound has a strong correlation with the true generalization error, we apply it to design a new minimum-training NAS algorithm. We demonstrate in Table 2 that with a simple random search algorithm, our approach can achieve a favorable performance compared with state-of-the-art training-free and minimum-training NAS algorithms.

# 3 Kernel Machine and Loss Path Kernel

In this section, we define notation, provide a brief overview of kernel methods, and introduce the main concept of interest—the loss path kernel.

## 3.1 Preliminaries

Consider a supervised learning problem where the task is to predict an output variable in $\mathcal{Y} \subseteq \mathbb{R}^k$ using a vector of input variables in $\mathcal{X} \subseteq \mathbb{R}^d$. Let $\mathcal{Z} \triangleq \mathcal{X} \times \mathcal{Y}$. We denote the training set by $\mathcal{S} \triangleq \{z_i\}_{i=1}^n$ with $z_i \triangleq (x_i, y_i) \in \mathcal{Z}$. We assume each point is drawn i.i.d. from an underlying distribution $\mu$. Let $\mathbf{X} = [x_1, \cdots, x_n]^T \in \mathbb{R}^{n \times d}$, $\mathbf{Y} = [y_1, \cdots, y_n]^T \in \mathbb{R}^{n \times k}$, and $\mathbf{Z} = [\mathbf{X}, \mathbf{Y}] \in \mathbb{R}^{n \times (d+k)}$.

We express a neural network in a general form $f(w, x) : \mathbb{R}^p \times \mathbb{R}^d \to \mathbb{R}^k$ where $w \in \mathbb{R}^p$ represents its parameters and $x \in \mathbb{R}^d$ is an input variable. The goal of a learning algorithm is to find a set of parameters that minimizes a population risk $L_\mu(w) = \mathbb{E}_{z \sim \mu}[\ell(w, z)]$ where $\ell(w, z) \triangleq \ell(f(w, x), y)$ is a loss function. Throughout this paper, we assume that $\ell(w, z) \in [0, 1]$ and is continuously differentiable. In practice, the underlying distribution $\mu$ is unknown so the learning algorithm minimizes an empirical risk on the training set $\mathcal{S}$ instead: $L_\mathcal{S}(w) = \frac{1}{n} \sum_{i=1}^n \ell(w, z_i)$. The *generalization gap* is defined as $L_\mu(w) - L_\mathcal{S}(w)$. The loss gradient with respect to the parameters $w$ is $\nabla_w \ell(w, z) = \nabla_w f(w, x)^\top \nabla_f \ell(f(w, x), y) \in \mathbb{R}^{p \times 1}$. Our analysis only requires that $\ell(w, z)$ is continuously differentiable w.r.t. $w$ and $z$ so $f$ can be either a fully connected neural network or a convolutional network or a residual network.

## 3.2 Kernel Method

Kernel methods [16, 51, 53] search for linear relations in high-dimensional feature space by using kernel functions. Rather than computing the coordinates in the feature space explicitly, they only need to calculate the inner product between data pairs, making it computationally easier.

A kernel is a function $K : \mathcal{X} \times \mathcal{X} \to \mathbb{R}$, such that for all $\boldsymbol{x}, \boldsymbol{x}' \in \mathcal{X}$, $K(\boldsymbol{x}, \boldsymbol{x}') = \langle \Phi(\boldsymbol{x}), \Phi(\boldsymbol{x}') \rangle$, where $\Phi : \mathcal{X} \to \mathcal{F}$ is a mapping from $\mathcal{X}$ to an (inner product) feature space $\mathcal{F}$.

**Proposition 1** (Shawe-Taylor et al. [53]). *A function $K : \mathcal{X} \times \mathcal{X} \to \mathbb{R}$, which is either continuous or has a finite domain, is a kernel function if and only if it is a symmetric function and, for any finite subset of $\mathcal{X}$, $\boldsymbol{x}_1, \cdots, \boldsymbol{x}_n \in \mathcal{X}$, the matrix $K(\mathbf{X}, \mathbf{X})$ is positive semi-definite, where $K(\mathbf{X}, \mathbf{X})$ is a $n \times n$ matrix whose $(i, j)$-th entry is $K(\boldsymbol{x}_i, \boldsymbol{x}_j)$.*

**Definition 1** (Kernel machine). Let $\mathcal{H}$ be the reproducing kernel Hilbert space (RKHS) corresponding to a kernel $K(\boldsymbol{x}, \boldsymbol{x}') = \langle \Phi(\boldsymbol{x}), \Phi(\boldsymbol{x}') \rangle$. A kernel machine $g : \mathcal{X} \to \mathbb{R}$ is a linear function in $\mathcal{H}$ such that its weight vector $\boldsymbol{\beta}$ can be expressed as a linear combination of the training points, i.e. $g(\boldsymbol{x}) = \langle \boldsymbol{\beta}, \Phi(\boldsymbol{x}) \rangle + b = \sum_{i=1}^{n} a_i K(\boldsymbol{x}_i, \boldsymbol{x}) + b$, where $\boldsymbol{\beta} = \sum_{i=1}^{n} a_i \Phi(\boldsymbol{x}_i)$ and $b$ is a constant. The RKHS norm of $g$ is $\|g\|_{\mathcal{H}} = \|\sum_{i=1}^{n} a_i \Phi(\boldsymbol{x}_i)\| = \sqrt{\sum_{i,j} a_i a_j K(\boldsymbol{x}_i, \boldsymbol{x}_j)}$.

Next, we introduce general kernel machine, which generalizes the concept of kernel machine.

**Definition 2** (General kernel machine). A general kernel machine $g : \mathcal{X} \to \mathbb{R}$ with a kernel $K(\boldsymbol{x}, \boldsymbol{x}')$ is $g(\boldsymbol{x}) = \sum_{i=1}^{n} a_i K(\boldsymbol{x}_i, \boldsymbol{x}) + h(\boldsymbol{x})$, where $h : \mathcal{X} \to \mathbb{R}$ is a function of $\boldsymbol{x}$. When $h(\boldsymbol{x})$ is a constant, $g(\boldsymbol{x})$ reduces to a kernel machine in Definition 1.

## 3.3 Neural Tangent Kernel and Loss Path Kernel

Neural tangent kernel (NTK) has been introduced by Jacot et al. [28] to establish an equivalence between infinite-width NNs and kernel regression. After then, there is a growing line of work applying NTK theory to study properties of over-parameterized NNs, such as optimization convergence [21, 20] and generalization capability [2, 11]. The neural tangent kernel [28] associated with a NN $f(\boldsymbol{w}, \boldsymbol{x})$ at $\boldsymbol{w}$ is defined as $\hat{\Theta}(\boldsymbol{w}; \boldsymbol{x}, \boldsymbol{x}') = \nabla_{\boldsymbol{w}} f(\boldsymbol{w}, \boldsymbol{x}) \nabla_{\boldsymbol{w}} f(\boldsymbol{w}, \boldsymbol{x}')^{\top} \in \mathbb{R}^{k \times k}$. Under certain conditions, such as infinite width limit and NTK parameterization, the NTK converges to a deterministic limit kernel $\Theta(\boldsymbol{x}, \boldsymbol{x}') \cdot \mathbf{I}_k$ that remains constant during training: $\hat{\Theta}(\boldsymbol{w}; \boldsymbol{x}, \boldsymbol{x}') \to \Theta(\boldsymbol{x}, \boldsymbol{x}') \cdot \mathbf{I}_k$, where $\Theta(\boldsymbol{x}, \boldsymbol{x}') : \mathbb{R}^d \times \mathbb{R}^d \to \mathbb{R}$ is a scalar kernel and $\mathbf{I}_k$ is a $k \times k$ identity matrix. Next, we introduce the main concepts of interest in this paper: the *loss tangent kernel* and the *loss path kernel*. They are central to characterizing the generalization behaviors of NNs trained by (stochastic) gradient flow.

**Definition 3** (Loss Tangent Kernel (LTK) $\bar{\mathsf{K}}$). The loss tangent kernel associated with the loss function $\ell(\boldsymbol{w}, \boldsymbol{z})$ is defined as $\bar{\mathsf{K}}(\boldsymbol{w}; \boldsymbol{z}, \boldsymbol{z}') = \langle \nabla_{\boldsymbol{w}} \ell(\boldsymbol{w}, \boldsymbol{z}), \nabla_{\boldsymbol{w}} \ell(\boldsymbol{w}, \boldsymbol{z}') \rangle \in \mathbb{R}$.

The LTK $\bar{\mathsf{K}}$ has a natural connection with the NTK $\hat{\Theta}$ by applying the chain rule:

$$\bar{\mathsf{K}}(\boldsymbol{w}; \boldsymbol{z}, \boldsymbol{z}') = \nabla_f \ell(\boldsymbol{w}, \boldsymbol{z})^{\top} \hat{\Theta}(\boldsymbol{w}; \boldsymbol{x}, \boldsymbol{x}') \nabla_f \ell(\boldsymbol{w}, \boldsymbol{z}').$$

Next, we introduce the loss path kernel, which integrates the LTK along a given path of the parameters. Later, we will characterize this path via the gradient flow dynamics.

**Definition 4** (Loss Path Kernel (LPK) $\mathsf{K}_T$). Suppose the weights follow a continuous path $\boldsymbol{w}(t) : [0, T] \to \mathbb{R}^p$ in their domain with a starting point $\boldsymbol{w}(0) = \boldsymbol{w}_0$, where $T$ is a predetermined constant. This path is determined by the training set $\mathcal{S}$ and the training time $T$. We define the loss path kernel associated with the loss function $\ell(\boldsymbol{w}, \boldsymbol{z})$ along the path as $\mathsf{K}_T(\boldsymbol{z}, \boldsymbol{z}'; \mathcal{S}) \triangleq \int_0^T \bar{\mathsf{K}}(\boldsymbol{w}(t); \boldsymbol{z}, \boldsymbol{z}') \mathrm{d}t$.

In Appendix B, we show LTK is Riemann integrable so the integral in the above definition is well-defined. Intuitively, the LTK $\bar{\mathsf{K}}(\boldsymbol{w}; \boldsymbol{z}, \boldsymbol{z}')$ measures the similarity between data points $\boldsymbol{z}$ and $\boldsymbol{z}'$ by comparing their loss gradients when evaluated using a fixed neural network parameter $\boldsymbol{w}$. The LPK $\mathsf{K}_T(\boldsymbol{z}, \boldsymbol{z}'; \mathcal{S})$ measures the overall similarity during the entire training time.

## 3.4 Rademacher Complexity

Rademacher complexity [52] measures the complexity of a hypothesis class. It takes into account the data distribution and is a central concept in statistical learning theory. Next, we recall its definition and a generalization upper bound via Rademacher complexity.

**Definition 5** (Empirical Rademacher complexity $\hat{\mathcal{R}}_S(\mathcal{G})$). Let $\mathcal{F} = \{f : \mathcal{X} \to \mathbb{R}^k\}$ be a hypothesis class. We denote $\mathcal{G}$ as the set of loss functions associated with each function in $\mathcal{F}$, defined as $\mathcal{G} = \{g : (\boldsymbol{x}, \boldsymbol{y}) \to \ell(f(\boldsymbol{x}), \boldsymbol{y}), f \in \mathcal{F}\}$. The empirical Rademacher complexity of $\mathcal{G}$ with respect to a sample set $\mathcal{S}$ is defined as: $\hat{\mathcal{R}}_S(\mathcal{G}) = \frac{1}{n} \mathbb{E}_{\boldsymbol{\sigma}} \left[ \sup_{g \in \mathcal{G}} \sum_{i=1}^n \sigma_i g(\boldsymbol{z}_i) \right]$, where $\boldsymbol{\sigma} = (\sigma_1, \ldots, \sigma_n)$ and $\sigma_i$ are independent uniform random variables taking values in $\{+1, -1\}$.

**Theorem 1** (Theorem 3.3 in Mohri et al. [38]). *Let $\mathcal{G}$ be a family of functions mapping from $\mathcal{Z}$ to* $[0, 1]$*. Then for any $\delta \in (0, 1)$, with probability at least $1 - \delta$ over the draw of an i.i.d. sample set $\mathcal{S} = \{\boldsymbol{z}_1, \ldots, \boldsymbol{z}_n\}$, the following holds for all $g \in \mathcal{G}$:* $\mathbb{E}_z[g(\boldsymbol{z})] - \frac{1}{n} \sum_{i=1}^n g(\boldsymbol{z}_i) \leq 2\hat{\mathcal{R}}_S(\mathcal{G}) + 3\sqrt{\frac{\log(2/\delta)}{2n}}$.

## 4 Gradient Flow

In this section, we establish a new connection between the loss dynamics of gradient flow and a general kernel machine equipped with the LPK. Using this result, we introduce a new generalization bound by analyzing the complexity of the collection of kernel machines induced by all possible training sets. Our analysis applies to a wide range of neural network architectures, as long as they are continuously differentiable. Our numerical experiments validate the tightness of our bound and its strong correlation with the true generalization error.

### 4.1 Loss Dynamics of Gradient Flow and Its Equivalence with General Kernel Machine

Consider the gradient flow dynamics (gradient descent with infinitesimal step size):

$$\frac{\mathrm{d}\boldsymbol{w}(t)}{\mathrm{d}t} = -\nabla_{\boldsymbol{w}} L_S(\boldsymbol{w}(t)) = -\frac{1}{n} \sum_{i=1}^n \nabla_{\boldsymbol{w}} \ell(\boldsymbol{w}(t), \boldsymbol{z}_i). \tag{1}$$

The above ODE is well-defined for a wide variety of conditions, e.g. local Lipschitz-continuity of the gradient or semi-convexity of the loss function [49, 23]. Next, we establish its connection with the general kernel machine (KM) in the following theorem.

**Theorem 2** (Equivalence with general KM.). *Suppose $\boldsymbol{w}(T) = \boldsymbol{w}_T$ is a solution of (1) at time $T$ with initialization $\boldsymbol{w}(0) = \boldsymbol{w}_0$. Then for any $\boldsymbol{z} \in \mathcal{Z}$,*

$$\ell(\boldsymbol{w}_T, \boldsymbol{z}) = \sum_{i=1}^n -\frac{1}{n} \mathsf{K}_T(\boldsymbol{z}, \boldsymbol{z}_i; \mathcal{S}) + \ell(\boldsymbol{w}_0, \boldsymbol{z}),$$

*where $\mathsf{K}_T$ is defined in Definition 4.*

The above theorem demonstrates that the loss of the NN at a certain fixed time is a general kernel machine. Herein, $\mathsf{K}_T$ is the LPK and we prove in Appendix C.1 that it is a valid kernel. Unlike previous NTK works that establish the equivalence between *infinite-width* NNs and kernel machines, our equivalence is much more general and holds for any NN that is continuously differentiable. Based on this equivalence, we characterize the generalization of NNs from the perspective of kernels. Note that $\mathsf{K}_T$ is a function of $\mathcal{S}$ and this property enables us to establish a data-dependent generalization bound shortly.

### 4.2 Generalization Bounds

We introduce the main result in this section: a generalization bound for NNs whose weights follow gradient flow in (1) at time $T$. We derive this bound by analyzing the Rademacher complexity of the function class of kernel machines induced by different training sets with constrained RKHS norms. Recall that each training set yields a distinct LPK. We define the collection of all such LPKs by

$$\mathcal{K}_T \triangleq \{\mathsf{K}_T(\cdot, \cdot; \mathcal{S}') : \mathcal{S}' \in \mathsf{supp}(\mu^{\otimes n}), \frac{1}{n^2} \sum_{i,j} \mathsf{K}_T(\boldsymbol{z}_i', \boldsymbol{z}_j'; \mathcal{S}') \leq B^2\}, \tag{2}$$

where $B > 0$ is some constant, $\mathcal{S}' = \{\boldsymbol{z}_1', \ldots, \boldsymbol{z}_n'\}$, $\mu^{\otimes n}$ is the joint distribution of $n$ i.i.d. samples drawn from $\mu$, and $\mathsf{supp}(\mu^{\otimes n})$ is the support set of $\mu^{\otimes n}$. Recall that $\mathcal{S} = \{\boldsymbol{z}_1, \ldots, \boldsymbol{z}_n\}$ is the training

set. Note the set in (2) includes the case of $\mathcal{S}' = \mathcal{S}$ if $\frac{1}{n^2} \sum_{i,j} \mathsf{K}_T(\boldsymbol{z}_i, \boldsymbol{z}_j; \mathcal{S}) \leq B^2$. Then we introduce a class of general kernel machines, corresponding to all different kernels in $\mathcal{K}_T$.

$$\mathcal{G}_T \triangleq \left\{ g(\boldsymbol{z}) = \sum_{i=1}^n -\frac{1}{n} \mathsf{K}(\boldsymbol{z}, \boldsymbol{z}_i'; \mathcal{S}') + \ell(\boldsymbol{w}_0, \boldsymbol{z}) : \mathsf{K}(\cdot, \cdot; \mathcal{S}') \in \mathcal{K}_T \right\}.$$

Note that $g(\boldsymbol{z}) \in \mathcal{G}_T$ corresponds to $\ell(\boldsymbol{w}_T, \boldsymbol{z})$ trained from one possible dataset $\mathcal{S}' \in \mathsf{supp}(\mu^{\otimes n})$. Next, we compute the Rademacher complexity of $\mathcal{G}_T$ and use it to obtain a generalization bound.

**Theorem 3.** $\hat{\mathcal{R}}_{\mathcal{S}}(\mathcal{G}_T) \leq \min\{U_1, U_2\}$. *Here*

$$U_1 = \frac{B}{n} \sqrt{\sup_{\mathsf{K}(\cdot, \cdot; \mathcal{S}') \in \mathcal{K}_T} \mathrm{Tr}(\mathsf{K}(\mathbf{Z}, \mathbf{Z}; \mathcal{S}')) + \sum_{i \neq j} \Delta(\boldsymbol{z}_i, \boldsymbol{z}_j)},$$

$$U_2 = \inf_{\epsilon > 0} \left( \frac{\epsilon}{n} + \sqrt{\frac{2 \ln \mathcal{N}(\mathcal{G}_T^{\mathcal{S}}, \epsilon, \|\|_1)}{n}} \right),$$

*where* $\mathcal{G}_T^{\mathcal{S}} = \{g(\mathbf{Z}) = (g(\boldsymbol{z}_1), \ldots, g(\boldsymbol{z}_n)) : g \in \mathcal{G}_T\}$, $\mathcal{N}(\mathcal{G}_T^{\mathcal{S}}, \epsilon, \|\|_1)$ *is the covering number of* $\mathcal{G}_T^{\mathcal{S}}$ *with the* $\ell_1$-*norm and*

$$\Delta(\boldsymbol{z}_i, \boldsymbol{z}_j) = \frac{1}{2} \left[ \sup_{\mathsf{K}(\cdot, \cdot; \mathcal{S}') \in \mathcal{K}_T} \mathsf{K}(\boldsymbol{z}_i, \boldsymbol{z}_j; \mathcal{S}') - \inf_{\mathsf{K}(\cdot, \cdot; \mathcal{S}') \in \mathcal{K}_T} \mathsf{K}(\boldsymbol{z}_i, \boldsymbol{z}_j; \mathcal{S}') \right].$$

The term $U_1$ is composed by two components $\sup_{\mathsf{K}(\cdot, \cdot; \mathcal{S}') \in \mathcal{K}_T} \mathrm{Tr}(\mathsf{K}(\mathbf{Z}, \mathbf{Z}; \mathcal{S}'))$ and $\Delta(\boldsymbol{z}_i, \boldsymbol{z}_j)$. The first component, according to the definition of LPK, quantifies the maximum magnitude of the loss gradient in $\mathcal{K}_T$ evaluated with the set $\mathcal{S}$ throughout the training trajectory. The second component assesses the range of variation of LPK within the set $\mathcal{K}_T$. The term $U_2$ is obtained from analyzing the covering number of $\mathcal{G}_T$. It shows that if the variation of the loss dynamics of gradient flow with different training data is small, then the complexity of $\mathcal{G}_T$ will also be small. The norm constraint $\frac{1}{n^2} \sum_{i,j} \mathsf{K}_T(\boldsymbol{z}_i', \boldsymbol{z}_j'; \mathcal{S}') \leq B^2$ balances a tradeoff between the tightness of the bound and the expressiveness of the set $\mathcal{G}_T$ (the number of datasets covered). Combining these two bounds with Theorem 1, we obtain the following generalization bound.

**Corollary 1** (Generalization bound for NN). *Fix* $B > 0$. *Let* $\hat{\mathcal{R}}_{\mathcal{S}}^{gf}(\mathcal{G}_T) = \min(U_1, U_2)$ *where* $U_1$ *and* $U_2$ *are defined in Theorem 3. For any* $\delta \in (0, 1)$, *with probability at least* $1 - \delta$ *over the draw of an i.i.d. sample set* $\mathcal{S} = \{\boldsymbol{z}_i\}_{i=1}^n$, *if* $\frac{1}{n^2} \sum_{i,j} \mathsf{K}_T(\boldsymbol{z}_i, \boldsymbol{z}_j; \mathcal{S}) \leq B^2$, *the following holds for* $\ell(\boldsymbol{w}_T, \boldsymbol{z})$ *that trained from* $\mathcal{S}$,

$$L_\mu(A_T(\mathcal{S})) - L_S(A_T(\mathcal{S})) \leq 2\hat{\mathcal{R}}_{\mathcal{S}}^{gf}(\mathcal{G}_T) + 3\sqrt{\frac{\log(2/\delta)}{2n}},$$

*where* $\boldsymbol{w}_T = A_T(\mathcal{S})$ *is the output from the gradient flow* (1) *at time* $T$ *by using* $\mathcal{S}$ *as input.*

Our result owns many compelling properties.

- First, our bound holds in a general setting as it does not hedge on a special NN architecture. In contrast, existing works [2, 11] only consider fully connected NNs and require NN to be ultra-wide.

- Our bound depends on the data distribution through the quantities in $U_1$ and $U_2$. This property not only significantly tightens our bound but can also help explain some empirical observations of NNs. For example, different from classical generalization theory, e.g. VC dimension, our complexity bounds depend on the labels directly, which helps explain the random label phenomenon [59] as shown in Figure 3 in Sec. 7.

- Our experiments in Sec. 7 (Figure 2) demonstrate the tightness of the generalization bound. Intuitively, our bound is tight because (1) instead of considering the entire hypothesis class, we focus on the subset of interest characterized by running gradient flow from a starting point $\boldsymbol{w}_0$; (2) we get the bound from an equivalence between NNs and general kernel machines, whose generalization bounds are tighter. Finally, we compare our generalization bound with two existing NTK-based bounds in Table 1.

# 5  Stochastic Gradient Flow

In the previous section, we derived a generalization bound for NNs trained from full-batch gradient flow. Here we extend our analysis to stochastic gradient flow and derive a corresponding generalization bound. To start with, we recall the dynamics of stochastic gradient flow (SGD with infinitesimal step size). Let $\mathcal{S}_t \subseteq \{1, \ldots, n\}$ be the indices of batch data used in time interval $[t, t+1]$ and $|\mathcal{S}_t| = m$ be the batch size. We establish a new connection between the loss dynamics of stochastic gradient flow and a general kernel machine. Then we investigate the complexity of the collection of such kernel machines that can be induced by various training sets.

**Theorem 4.** *Suppose $\boldsymbol{w}(T) = \boldsymbol{w}_T$ is a solution of stochastic gradient flow at time $T \in \mathbb{N}$ with initialization $\boldsymbol{w}(0) = \boldsymbol{w}_0$. Then for any $\boldsymbol{z} \in \mathcal{Z}$,*

$$\ell(\boldsymbol{w}_T, \boldsymbol{z}) = \sum_{t=0}^{T-1} \sum_{i \in \mathcal{S}_t} -\frac{1}{m} \mathsf{K}_{t,t+1}(\boldsymbol{z}, \boldsymbol{z}_i; \mathcal{S}) + \ell(\boldsymbol{w}_0, \boldsymbol{z}),$$

*where $\mathsf{K}_{t,t+1}(\boldsymbol{z}, \boldsymbol{z}_i; \mathcal{S}) = \int_t^{t+1} \bar{\mathsf{K}}(\boldsymbol{w}(t); \boldsymbol{z}, \boldsymbol{z}_i) dt$ with $\bar{\mathsf{K}}$ defined in Definition 3.*

The above theorem shows that the loss of the NN in stochastic gradient flow dynamics can be characterized by a sum of general kernel machines. In particular, when we use the full batch at each time interval (i.e., $m = n$), the above result recovers Theorem 2. To study its generalization behavior, we introduce the class of kernel machines induced by different training sets $\mathcal{S}' \in \mathsf{supp}(\mu^{\otimes n})$ with constrained RKHS norms. Specifically, given $B_t > 0$ for $t = 0, \cdots, T-1$, we define

$$\mathcal{K}_T = \{(\mathsf{K}_{0,1}(\cdot, \cdot; \mathcal{S}'), \cdots, \mathsf{K}_{T-1,T}(\cdot, \cdot; \mathcal{S}')) : \mathcal{S}' \in \mathsf{supp}(\mu^{\otimes n}), \frac{1}{m^2} \sum_{i,j \in \mathcal{S}_t} \mathsf{K}_{t,t+1}(\boldsymbol{z}_i', \boldsymbol{z}_j'; \mathcal{S}') \leq B_t^2\}.$$

Note this set includes the kernel induced by the training set $\mathcal{S}$ if it satisfies the constraints. Then $\ell(\boldsymbol{w}_T, \boldsymbol{z})$ trained from all feasible $\mathcal{S}' \in \mathsf{supp}(\mu^{\otimes n})$ form a function class

$$\mathcal{G}_T \triangleq \Big\{ \sum_{t=0}^{T-1} \sum_{i \in \mathcal{S}_t} -\frac{1}{m} \mathsf{K}_{t,t+1}(\boldsymbol{z}, \boldsymbol{z}_i'; \mathcal{S}') + \ell(\boldsymbol{w}_0, \boldsymbol{z}) : \mathsf{K}(\cdot, \cdot; \mathcal{S}') \in \mathcal{K}_T \Big\}. \tag{3}$$

Next, we upper bound the Rademacher complexity of the function class $\mathcal{G}_T$. This bound can naturally translate into a generalization bound by equipping with Theorem 1.

**Theorem 5.** *The Rademacher complexity of $\mathcal{G}_T$ defined in (3) has an upper bound:*

$$\hat{\mathcal{R}}_{\mathcal{S}}(\mathcal{G}_T) \leq \sum_{t=0}^{T-1} \frac{B_t}{n} \sqrt{\sup_{\mathsf{K}(\cdot, \cdot; \mathcal{S}') \in \mathcal{K}_T} \mathrm{Tr}(\mathsf{K}_{t,t+1}(\mathbf{Z}, \mathbf{Z}); \mathcal{S}') + \sum_{i \neq j} \Delta_t(\boldsymbol{z}_i, \boldsymbol{z}_j)}.$$

*where $\Delta_t(\boldsymbol{z}_i, \boldsymbol{z}_j) = \frac{1}{2} \Big[ \sup_{\mathsf{K}(\cdot, \cdot; \mathcal{S}') \in \mathcal{K}_T} \mathsf{K}_{t,t+1}(\boldsymbol{z}_i, \boldsymbol{z}_j; \mathcal{S}') - \inf_{\mathsf{K}(\cdot, \cdot; \mathcal{S}') \in \mathcal{K}_T} \mathsf{K}_{t,t+1}(\boldsymbol{z}_i, \boldsymbol{z}_j; \mathcal{S}') \Big]$.*

We assumed that mini-batches indices $\mathcal{S}_t$ are chosen before training. However, our analysis can be extended to accommodate random mini-batch selections of any sampling strategy by enumerating all potential $\mathcal{S}_t$ in $\mathcal{K}_T$.

# 6  Case Study & Use Case

In the previous sections, we derived generalization bounds for NNs trained with (stochastic) gradient flow. These bounds may initially appear complex due to their dependence on the training process. Here we show that these bounds can be significantly simplified by applying them to infinite-width NNs (and stable algorithms in Appendix E.2, norm-constraint NNs in Appendix E.3). Moreover, we demonstrate that our generalization bounds maintain a high correlation with the true generalization error. As a result, we use them to guide the design of NAS, and our experimental results demonstrate that this approach has a favorable performance compared with state-of-the-art algorithms.

## 6.1 Infinite-width NN

In this subsection, we consider a special case of infinite-width NNs trained by gradient flow and derive pre-computed generalization bounds. We focus on gradient flow to simplify the presentations but our results can be directly extended to stochastic gradient flow. For an infinite-width NN, under certain conditions, the neural tangent kernel keeps unchanged during training: $\hat{\Theta}(\boldsymbol{w}_t; \boldsymbol{x}, \boldsymbol{x}') \to \Theta(\boldsymbol{x}, \boldsymbol{x}') \cdot \mathbf{I}_k$. Consider a $\rho$-Lipschitz loss function, i.e. $\|\nabla_f \ell(\boldsymbol{w}, \boldsymbol{z})\| \le \rho$. The Rademacher complexity in Theorem 3 can be bounded by $\hat{\mathcal{R}}_{\mathcal{S}}(\mathcal{G}_T) \le U_\infty$, where

$$U_\infty = \frac{\rho B \sqrt{T}}{n} \sqrt{\sum_{i,j} |\Theta(\boldsymbol{x}_i, \boldsymbol{x}_j)|}. \tag{4}$$

In this infinite-width regime, our bound has no dependence on the initialization $\boldsymbol{w}_0$ since the NTK converges to a deterministic limit and has no dependence on the parameters. That means the bound holds for all possible $\boldsymbol{w}_0$ of infinite-width NNs trained by gradient flow from initialization. Compared with the bound $\tilde{O}(L \cdot \sqrt{\frac{\mathbf{Y}^\top (\boldsymbol{\Theta})^{-1} \mathbf{Y}}{n}})$ in [11], $U_\infty$ has several advantages: (1) it has no dependence on the number of layers $L$; (2) it holds for NNs with multiple outputs.

## 6.2 Correlation Analysis and NAS

As a practical application, we apply our generalization bounds to guide the design of NAS. We first introduce a quantity $U_{\text{sgd}}$ simplified from the bound in Theorem 5, defined as

$$U_{\text{sgd}} = \sum_{t=0}^{T-1} \frac{1}{n} \sqrt{\frac{1}{m^2} \sum_{i,j \in \mathcal{S}_t} \mathsf{K}_{t,t+1}(\boldsymbol{z}_i, \boldsymbol{z}_j; \mathcal{S})} \sqrt{\text{Tr}(\mathsf{K}_{t,t+1}(\mathbf{Z}, \mathbf{Z}); \mathcal{S})}.$$

$U_{\text{sgd}}$ can be computed along with training a NN via SGD on a training set $\mathcal{S}$. Combining it with the training loss, we define the following quantity as an estimate of the population loss:

$$\text{Gene}(\boldsymbol{w}, \mathcal{S}) = L_{\mathcal{S}}(\boldsymbol{w}) + 2U_{\text{sgd}}. \tag{5}$$

We analyze the correlation between $\text{Gene}(\boldsymbol{w}, \mathcal{S})$ and the true generalization error by randomly sampling 100 NN architectures from NAS-Bench-201 [19]. For each, we compute both $\text{Gene}(\boldsymbol{w}, \mathcal{S})$ and the true generalization error. Since solving the gradient flow ODE is computationally infeasible for the large NNs in NAS-Bench-201, we apply a trapezoidal rule to approximate the integration in LPK $K_{t,t+1}$. This approximation enables us to compute $U_{\text{sgd}}$ efficiently. Figure 1 demonstrates the correlation between $\text{Gene}(\boldsymbol{w}, \mathcal{S})$ and the test error. The left figures plot the test error at epoch 1 or 2 against $\text{Gene}(\boldsymbol{w}, \mathcal{S})$ of the respective epochs, showing a strong positive correlation between them. The right figures plot the test error at convergence against $\text{Gene}(\boldsymbol{w}, \mathcal{S})$ at epoch 1 or 2, which also demonstrate a positive correlation. The outlier is caused by some architecture with large loss gradients. This experiment shows that $\text{Gene}(\boldsymbol{w}, \mathcal{S})$ at the initial training stage can predict the performance of NNs at convergence. Based

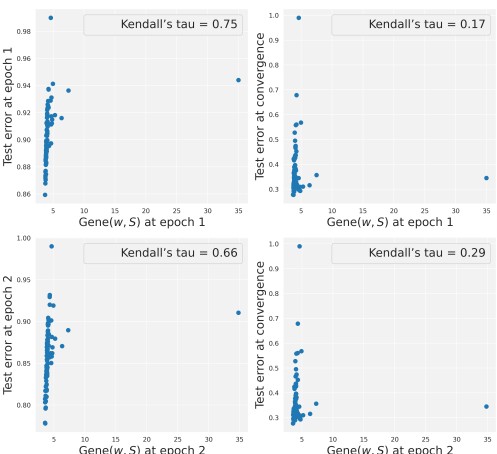

Figure 1: Correlation between $\text{Gene}(\boldsymbol{w}, \mathcal{S})$ and the test error on CIFAR-100 at epoch 1 and epoch 2. Kendall's tau shows they have a strong positive correlation.

on this observation, we use $\text{Gene}(\boldsymbol{w}, \mathcal{S})$ as a metric in NAS for selecting architectures at the initial training stage (see Table 2). This approach significantly reduces computational costs compared with training-based NAS algorithms [47, 34, 18, 33].

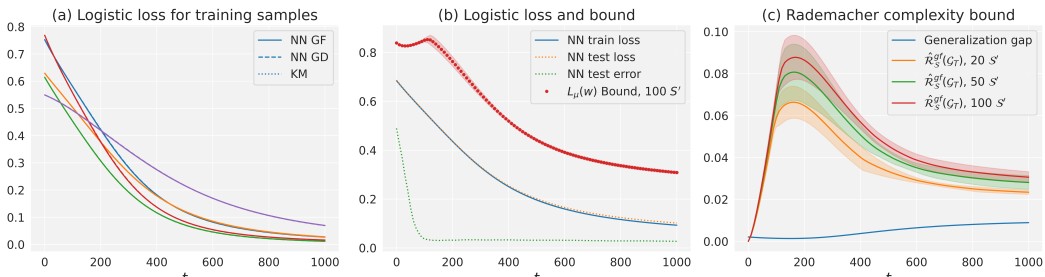

Figure 2: **Experiment (I)**. (a) shows the dynamics of logistic loss for 5 randomly selected training samples for NN trained by gradient flow (NN GF), NN trained by GD (NN GD), and the equivalent general kernel machine (KM) in Theorem 2. The dynamics of NN GF and KM overlap and thus verify the equivalence in Theorem 2. The dynamics of NN GF and NN GD are consistently close throughout the training process. (b) shows NN GF's training loss, test loss, test error, and upper bound for $L_\mu(\boldsymbol{w}_T)$ in Corollary 1. (c) shows that the complexity bound $\hat{\mathcal{R}}_{\mathcal{S}}^{gf}(\mathcal{G}_T)$ in Corollary 1 captures the generalization gap $L_\mu(\boldsymbol{w}_T) - L_{\mathcal{S}}(\boldsymbol{w}_T)$ well. It first increases and then converges after sufficient training time.

## 7 Numerical Experiments

We conduct comprehensive numerical experiments to demonstrate our generalization bounds. We observe that our complexity bounds are tight with respect to the generalization gap and can capture how noisy label influences the generalization behaviors of NNs. Moreover, we apply $\text{Gene}(\boldsymbol{w}, \mathcal{S})$ in (5) to NAS and demonstrate favorable performance compared with state-of-the-art algorithms.

**(I) Generalization bound in Corollary 1.** In Figure 2 (more detailed in Figure A.4), we use a logistic loss to train a two-layer NN with 100 hidden nodes for binary classification on MNIST 1 and 7 [31] by full-batch gradient flow and compute its generalization bound. Due to the computational cost of solving the gradient flow ODE and computing the kernel, we only train and calculate the bound on $n = 1000$ training samples. The bound would be tighter with more training samples. The NN is initialized using the NTK parameterization [28]. We use the Softplus activation function, defined as $\text{Softplus}(x) = \frac{1}{\beta} \ln(1 + e^{\beta x})$. This function is continuously differentiable and serves as a smooth approximation to the ReLU activation function. In our experiments, we set $\beta = 10$. To train the NN via gradient flow, we solve the gradient flow ODE given by (1) to decide the NN parameters. For the equivalent general KM, we compute the LTK using the NN parameters and integrate it to get the LPK $\mathsf{K}_T$. These ODEs are computed with torchdiffeq [12]. To estimate the generalization bound, we train the NN on (20, 50, 100) independently sampled training sets $\mathcal{S}'$ to estimate the $\mathcal{K}_T$ and $\mathcal{G}_T$, and the supremum in the bound $U_1$ is estimated by taking the maximum over the finite set $\mathcal{K}_T$. For $U_2$, we compute an upper bound of it by setting $\epsilon$ as the largest $\ell_1$ distance between any two $g(\mathbf{Z}) \in \mathcal{G}_T^{\mathcal{S}}$ and $\mathcal{N}(\mathcal{G}_T^{\mathcal{S}}, \epsilon, \|\|_1) = 1$ because in this case any $g(\mathbf{Z}) \in \mathcal{G}_T^{\mathcal{S}}$ will satisfy as an $\epsilon$-cover. We run each experiment five times and plot the mean and standard deviation. The numerical experiments demonstrate that our complexity bound is tight and can capture the generalization gap well. As the number of $\mathcal{S}'$ increases, our bound converges to the true supremum value. To estimate the true supremum value, we apply the extreme value theory in Appendix A.3 and show the gap between the finite maximum and supremum is small, validating using a finite maximum as an estimate for our bound.

We train the NN using GD with a finite learning rate $\eta = 10$ to compare with the NN trained by gradient flow. The training time $t = \eta \times$ training steps. In Figure 2 (a) and Figure A.4 (a)(b), we observe that the loss of the NN trained by GD and that trained by gradient flow are consistently close throughout the entire training process. Consequently, while we established upper bounds for NNs trained by gradient flow, these results can serve as an (approximate) upper bound for NNs trained by GD with a finite learning rate.

Notably, for the NN in this experiment, the VC dimension [7] is 55957.3, the norm-based bound in [6] is 140.7 at $T = 1000$, and the NTK-based bound for an ultra-wide NN in [11] is 1.44, which are all vacuous (larger than 1), while our bound is tight (0.03 at $T = 1000$). See a detailed comparison in Appendix A.2. We also conduct an experiment of three-layer NN (3072-100-100-1) trained on binary CIFAR-10 (cat and dog) [30] in Figure A.5, where there is a larger generalization gap.

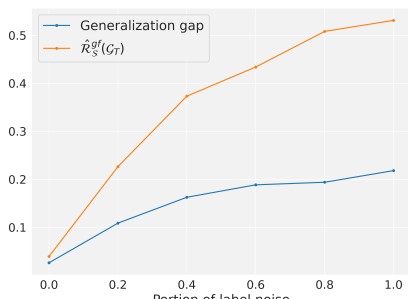

Figure 3: **Experiment (II)**. Generalization bound with label noise at $T = 20000$.

Table 2: **Experiment (III)**. Comparison with state-of-the-art training-free/minimum-training NAS methods on NAS-Bench-201. Test accuracy with mean and deviation are reported. "Best" is the best accuracy over the four runs. "Optimal" indicates the best test accuracy achievable in NAS-Bench-201 search space. RS: randomly sample 100 architectures and select the one with the best metric value.

| Algorithm | CIFAR-10 Accuracy | Best | CIFAR-100 Accuracy | Best |
|---|---|---|---|---|
| **Baselines** | | | | |
| TENAS [13] | 93.08±0.15 | 93.25 | 70.37±2.40 | **73.16** |
| RS + LGA$_3$ [39] | 93.64 | | 69.77 | |
| **Ours** | | | | |
| RS + Gene$(\boldsymbol{w}, \mathcal{S})_1$ | 93.68±0.12 | 93.84 | 72.02±1.43 | 73.15 |
| RS + Gene$(\boldsymbol{w}, \mathcal{S})_2$ | **93.79**±0.18 | **94.02** | **72.76**±0.33 | 73.15 |
| Optimal | 94.37 | | 73.51 | |

**(II) Generalization bound with label noise.** The settings are similar to Experiment (I) except we corrupt the labels with different portions of noise and calculate the bound after training NN until $T = 20000$. We estimate the bound with 20 training sets $\mathcal{S}'$. The results in Figure 3 show that our bound has a strong correlation with the generalization gap and increases with the portion of label noise. Unlike classical generalization theory, e.g. VC dimension, our generalization bound can help explain the random label phenomenon [59].

**(III) Neural architecture search (NAS).** We apply Gene$(\boldsymbol{w}, \mathcal{S})$ in Eq. (5) to guide the design of NAS. The results are shown in Table 2. We use a simple random search (RS) with Gene$(\boldsymbol{w}, \mathcal{S})$, where 100 architectures are sampled from the search space for evaluation, and the architecture with smallest Gene$(\boldsymbol{w}, \mathcal{S})$ is selected. $U_{\mathrm{sgd}}$ is estimated with a batch of data of size 600. Build upon Sec. 6.2, we apply Gene$(\boldsymbol{w}, \mathcal{S})_1$ and Gene$(\boldsymbol{w}, \mathcal{S})_2$ (Gene$(\boldsymbol{w}, \mathcal{S})$ after training 1 and 2 epochs) to select NN architectures at the initial training stage in order to reduce computational costs. We compare our method with state-of-the-art training-free/minimum-training NAS algorithms [13, 39]. We run the experiments four times with different random seeds and report the mean and standard deviation. We reproduce the results in Chen et al. [13] using their released code and directly adopt the results reported in Mok et al. [39] as they did not release the code. The results show our approach of RS + Gene$(\boldsymbol{w}, \mathcal{S})$ can achieve favorable performance compared with state-of-the-art training-free/minimum-training NAS algorithms.

# 8 Conclusion and Future Work

In this paper, we establish a new connection between the loss dynamics of (stochastic) gradient flow and a general kernel machine. Building upon this result, we introduce generalization bounds for NNs trained from (stochastic) gradient flow. Our bounds hold for any continuously differentiable NN architectures (both finite-width and ultra-wide) and are generally tighter than existing bounds. Moreover, for infinite-width NNs, we obtain a pre-computed generalization bound for the whole training process. Finally, we apply our results to NAS and demonstrate favorable performance compared with state-of-the-art NAS algorithms.

There are several directions for future research. First, evaluating our generation bounds relies on the loss gradient, which may contain private and sensitive information. One potential fix would be accessing such information in a differentially private manner and it would be interesting to investigate how this "noisy" observation of gradient information influences our generalization bounds. Second, it is worth exploring how other optimization algorithms and different model architectures influence the generalization bounds. Finally, our bounds provide worst-case guarantees to the generalization of NNs and it would be interesting to extend our results to obtain expected bounds for further sharpening the results.

# 9 Acknowledgement

We thank the anonymous reviewers for valuable suggestions to improve the paper. We also thank the San Diego Supercomputer Center and the MIT-IBM Watson AI Lab for computing resources. Y. Chen and T.-W. Weng are supported by National Science Foundation under Grant No. 2107189 and 2313105.

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

# Appendices

## A  Additional Experiments

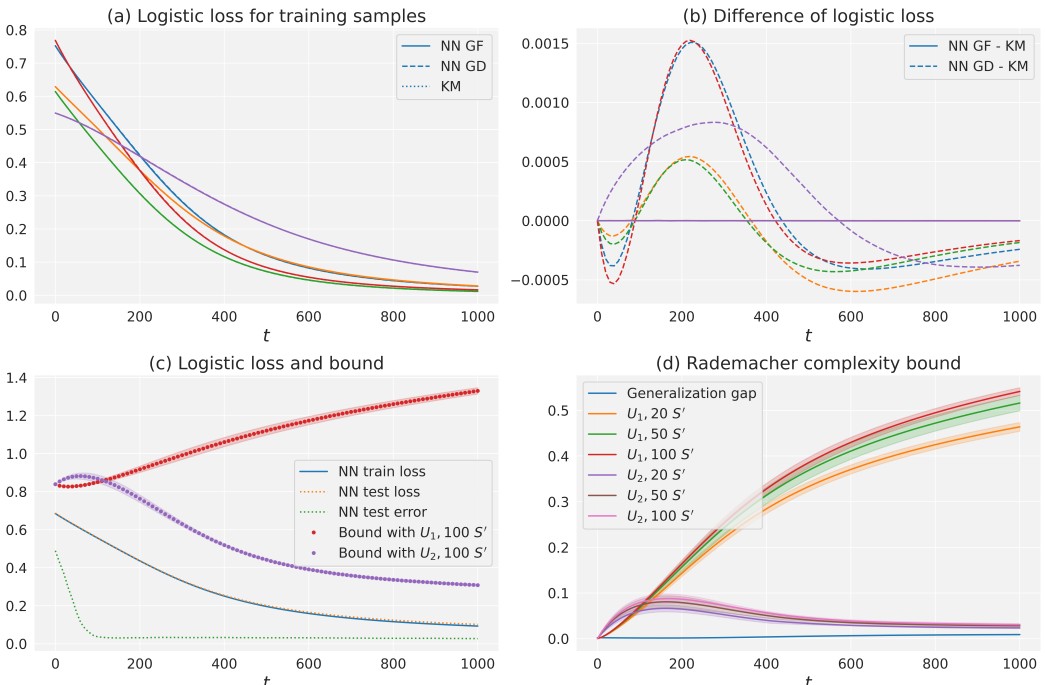

Figure A.4: **Experiment (I)**. (a) shows the dynamics of logistic loss for 5 randomly selected training samples for NN trained by gradient flow (NN GF), NN trained by gradient descent (NN GD), and the equivalent general kernel machine (KM) in Theorem 2. The dynamics of NN GF and KM overlap and verify the equivalence in Theorem 2. The dynamics of NN GF and NN GD are consistently close throughout the training process. (b) shows the differences between NN GF and KM are 0, which verifies our equivalence in Theorem 2. The differences between NN GD and KM are also small. (c) shows NN GF's training loss, test loss, test error, and population loss bounds we estimated. Bound with $U_1/U_2$ is the bound for $L_\mu(w)$ by applying $U_1/U_2$ in Corollary 1. (d) shows $U_1$ and $U_2$ in Theorem 3 first increase then converge after sufficient training time. The numerical experiments demonstrate that our complexity bound is tight and can capture the generalization gap well. As the number of $\mathcal{S}'$ increases, our bound converges to the true supremum value.

Experiments are implemented with PyTorch [46] on 24G A5000 and 32G V100 GPUs.

### A.1  Computation Cost of Experiments

In Experiment (I), estimating the bound with 20 $S'$ (solving 20 gradient flow ODE) costs 500s and training NN costs 0.29s. The GPU memory required by estimating the bound is 2406MB and training NN requires 1044MB.

| GPU hours | CIFAR-10 | CIFAR-100 |
|---|---|---|
| RS + Gene($\boldsymbol{w}, \mathcal{S}$) (Ours) | 0.036 | 0.037 |
| Training one NN architecture to convergence | 1.83 | 2.56 |

Table 3: Averaged computational cost (GPU hours) for one architecture in Experiment (III).

For Experiment (III), we report the averaged computational cost (GPU hours) of our approach for one architecture and the computational cost of training one NN architecture to convergence in

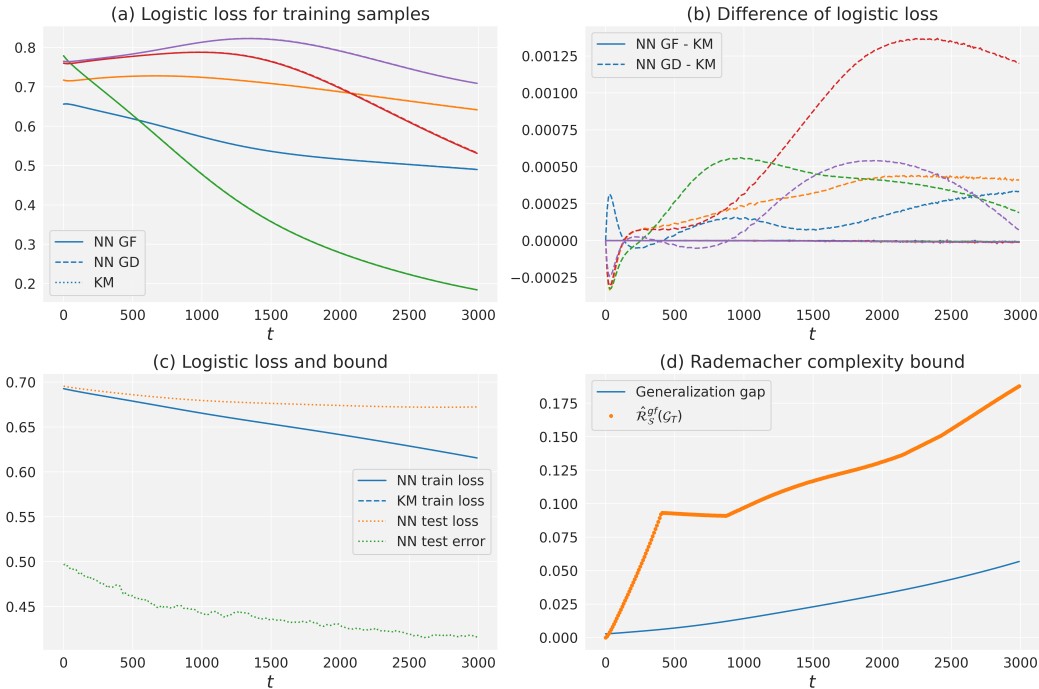

Figure A.5: **Experiment (I) on CIFAR-10**. Generalization bound for three-layer NN (3072-100-100-1) trained on binary CIFAR-10 (cat and dog). The experiment demonstrates that our complexity bound is tight and can capture the generalization gap well.

Table 3. Note our approach calculates $\text{Gene}(\boldsymbol{w}, \mathcal{S})$ only after training for 1 or 2 epochs, which saves computational cost a lot.

## A.2 Comparison with Existing Generalization Bounds

To make a stronger comparison with existing work, we have conducted two additional experiments below to show that our bound is much tighter than prior work on both finite-width NNs and infinite-width NNs.

**(1)** For the finite-width NN, we compare with previous uniform convergence bounds (VC dimension [7] and norm-based bounds [6]) as the NTK-based bounds [2, 11] are limited to infinite or ultra-wide NNs. Note our bounds are also uniform convergence bounds. We follow the same setting in our Experiment I, and calculate the bound for two-layer NN with 100 hidden nodes (finite-width) at training time T=1000 and sample size n=1000. Here $L$ is the number of layers, $p$ is the number of parameters, $W^i$ is the weight matrix for the $i$-th layer, and $m$ is the largest width of NN including the input dimension. Constants in big $O$ are ignored in the calculation. The results in Table 4 show that our bound is much tighter than previous VC dimensions and norm-based bounds.

Finally, we remark that existing works have observed that VC dimension and norm-based bounds are mostly vacuous (see e.g., Figure 5 of [44] and Figure 4 in [1]) while our bound is non-vacuous as shown above and in Figure 2 and Figure A.5.

**(2)** For the infinite-width NN, we compare with the NTK-based bound in Cao & Gu [11] in a similar setting as Experiment I – two-layer NN and binary MNIST (1 vs. 7) with n=1000. Note that their bound only holds for an ultra-wide NN. We compute the infinite-width NTK using Neural Tangents [45] and calculate their bound accordingly. For our bound, we train a two-layer NN with 1000 hidden nodes (to approximate an ultra-wide NN) and compute our bound at $T = 1000$ (almost convergence). Constants in big $O$ are ignored in the calculation. The results in Table 5 show that our bound is much tighter than previous NTK-based bounds.

Table 4: Comparison with existing uniform convergence bounds (VC dimension [7] and norm-based bounds [6]). "Finite sample estimate": Bound estimated from 100 $\mathcal{S}'$. "Extreme value estimate": Bound estimated from extreme value theory. See more detail in Appendix A.3.

| Method | Bound | Value | Tighter over [6] |
|---|---|---|---|
| VC dimension [7] | $O(\frac{Lplog(p)}{\sqrt{n}})$ | 55957.3 | - |
| Norm-based bound [6] | $\tilde{O}\left(\frac{1}{\sqrt{n}}\prod_{i=1}^{L}\left\|W^i\right\|_2 \left(\sum_{i=1}^{L}\frac{\|W^{i\top}\|_{2,1}^{2/3}}{\|W^i\|_2^{2/3}}\right)^{3/2}\ln(m)\right)$ | 140.7 | - |
| **Ours** | | | |
| Finite sample estimate | $\hat{\mathcal{R}}_\mathcal{S}^{gf}(\mathcal{G}_T)$ | 0.03 | 4690× |
| Extreme value estimate | $\hat{\mathcal{R}}_\mathcal{S}^{gf}(\mathcal{G}_T)$ | 0.06 | 2345× |

Table 5: Comparison with NTK-based bounds [11].

| Method | Bound | Value | Tighter over [11] |
|---|---|---|---|
| Cao & Gu [11] | $\tilde{O}(L\cdot\sqrt{\frac{Y^\top(\mathbf{\Theta})^{-1}Y}{n}})$ | 1.44 | - |
| **Ours** | | | |
| Finite sample estimate | $\hat{\mathcal{R}}_\mathcal{S}^{gf}(\mathcal{G}_T)$ | 0.026 | 55.38× |
| Extreme value estimate | $\hat{\mathcal{R}}_\mathcal{S}^{gf}(\mathcal{G}_T)$ | 0.08 | 18.00× |

## A.3 Estimate Supremum with Extreme Value Theory

In the all above experiments, we estimate the supremum in the Rademacher bound in Theorem 3 with the maximum over a finite set. To get the true supremum value of the bound, we apply the extreme value theory to estimate it, similar to estimating the local Lipschitz of NN in [57]. We show the finite maximum is close to the true supremum value, validating using a finite maximum as an estimate for our bound. We first state the following result from extreme value theory and then explain how we can apply it to estimate our bounds.

**Theorem 6** (Fisher–Tippett–Gnedenko theorem). *Let $X_1, X_2, \cdots, X_m$ be a sequence of independent and identically-distributed random variables with cumulative distribution function $F$. Suppose that there exist two sequences of real numbers $a_m > 0$ and $b_m \in \mathbb{R}$ such that the following limits converge to a non-degenerate distribution function:*

$$\lim_{m\to\infty} \mathbb{P}(\frac{\max\{X_1, X_2, \cdots, X_m\} - a_m}{b_m} \le x) = G(x).$$

*Then the limit distribution $G$ belongs to either the Gumbel class (Type I), the Fréchet class (Type II), or the Reverse Weibull class (Type III):*

$$\textit{Gumbel class (Type I):} \quad G(x) = exp\left\{-exp\left(-\frac{x-a_w}{b_w}\right)\right\}, x \in \mathbb{R},$$

$$\textit{Fréchet class (Type II):} \quad G(x) = \begin{cases} 0, & x < a_w, \\ exp\left\{-\left(\frac{x-a_w}{b_w}\right)^{-c_w}\right\}, & x \ge a_w, \end{cases}$$

$$\textit{Reverse Weibull class (Type III):} \quad G(x) = \begin{cases} exp\left\{-\left(\frac{a_w-x}{b_w}\right)^{-c_w}\right\}, & x < a_w, \\ 1, & x \ge a_w, \end{cases}$$

*where $a_w \in \mathbb{R}, b_w > 0$ and $c_w > 0$ are the location, scale, and shape parameters, respectively.*

Our bounds are finite as long as $\ell(\boldsymbol{w}, \boldsymbol{z})$ is $L_\ell$-Lipschitz (will explain in detail later). Thus we are particularly interested in the reverse Weibull class, as its CDF has a finite right end-point (denoted as $a_w$). The right end-point reveals the upper limit of the distribution, known as the *extreme value*. In our case, the extreme value is exactly the supremum we want to estimate.

To compute $U_2$, we compute an upper bound of it by setting $\epsilon$ as the largest $\ell_1$ distance between any two $g(\mathbf{Z}) \in \mathcal{G}_T^{\mathcal{S}}$ and $\mathcal{N}(\mathcal{G}_T^{\mathcal{S}}, \epsilon, \|\|_1) = 1$ because in this case any $g(\mathbf{Z}) \in \mathcal{G}_T^{\mathcal{S}}$ will satisfy as an $\epsilon$ cover.

$$U_2 = \inf_{\epsilon > 0} \left( \frac{\epsilon}{n} + \sqrt{\frac{2 \ln \mathcal{N}(\mathcal{G}_T^{\mathcal{S}}, \epsilon, \|\|_1)}{n}} \right) \le \sup_{g_1, g_2 \in \mathcal{G}_T} \frac{1}{n} \|g_1(\mathbf{Z}) - g_2(\mathbf{Z})\|_1$$

Denote the right hand side as $U_2^*$. Since each $\mathsf{K}(z_i, z_j; \mathcal{S}') \le L_\ell T^2$, $U_2^* \le 2L_\ell T^2$.

Note $\mathbf{Z}$ is fixed. Then for all $g_1, g_2 \in \mathcal{G}_T$, $\|g_1(\mathbf{Z}) - g_2(\mathbf{Z})\|_1$ are i.i.d. random variables, because $g \in \mathcal{G}_T$ are trained from i.i.d. $\mathcal{S}'$. Consider a finite set $\mathcal{G}_{N_g} \subset \mathcal{G}_T$ with size $N_g = |\mathcal{G}_{N_g}|$. As $N_g \to \infty$,

$$\hat{U}_2 \triangleq \max_{g_1, g_2 \in \mathcal{G}_{N_g}} \|g_1(\mathbf{Z}) - g_2(\mathbf{Z})\|_1 \xrightarrow{P} U_2^*, \quad N_g \to \infty.$$

Then we can apply extreme value theory to estimate $U_2^*$. We generate $N_b$ batch of $\mathcal{G}_{N_g}$, compute their $\hat{U}_2$ and store them in a set. Then with these $\hat{U}_2$'s, we perform a maximum likelihood estimation of reverse Weibull distribution parameters, and the location estimate $a_w$ is used as an estimate of the $U_2^*$. To validate that reverse Weibull distribution is a good fit for the empirical distribution of the $\hat{U}_2$'s, we conduct Kolmogorov-Smirnov goodness-of-fit test (a.k.a. K-S test) to calculate the K-S test statistics D and corresponding p-values. The null hypothesis is that $\hat{U}_2$'s follow a reverse Weibull distribution.

We follow the same setting as Experiment I and want to estimate $U_2^*$ at $T = 1000$. Figure A.6 shows a result of estimating $U_2^*$ with $N_b = 50$ and $N_g = 1000$. The estimated $U_2^* = 0.06$ is quite close to the finite maximum, validating using a finite maximum as an estimate for our bound.

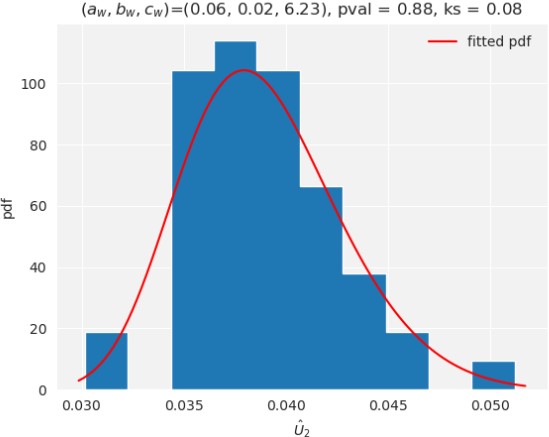

Figure A.6: Estimating $U_2^*$ with $N_b = 50$ finite maximum over sets of size $N_g = 1000$. The red line is the fitted probability distribution function (PDF) of the reverse Weibull distribution. The small D-statistics of K-S test (ks) and large p-values (pval) show the hypothesized reverse Weibull distribution fits the empirical distribution of $\hat{U}_2$ well. The estimated $U_2^* = 0.06$ is quite close to the finite maximum, validating using a finite maximum as an estimate for our bound.

For $U_1$,

$$U_1 = \frac{B}{n} \sqrt{\sup_{\mathsf{K}(\cdot, \cdot; \mathcal{S}') \in \mathcal{K}_T} \mathrm{Tr}(\mathsf{K}(\mathbf{Z}, \mathbf{Z}; \mathcal{S}')) + \sum_{i \ne j} \Delta(\boldsymbol{z}_i, \boldsymbol{z}_j)}$$

where

$$\Delta(\boldsymbol{z}_i, \boldsymbol{z}_j) = \frac{1}{2} \left[ \sup_{\mathsf{K}(\cdot, \cdot; \mathcal{S}') \in \mathcal{K}_T} \mathsf{K}(\boldsymbol{z}_i, \boldsymbol{z}_j; \mathcal{S}') - \inf_{\mathsf{K}(\cdot, \cdot; \mathcal{S}') \in \mathcal{K}_T} \mathsf{K}(\boldsymbol{z}_i, \boldsymbol{z}_j; \mathcal{S}') \right].$$

$\mathrm{Tr}(\mathsf{K}(\mathbf{Z}, \mathbf{Z}; \mathcal{S}'))$ and $\mathsf{K}(\boldsymbol{z}_i, \boldsymbol{z}_j; \mathcal{S}')$ for $i, j \in [n]$ are random variables that only depends on $\mathcal{S}'$. For different $\mathcal{S}', \mathcal{S}'' \in \mathrm{supp}(\mu^{\otimes n})$, $\mathrm{Tr}(\mathsf{K}(\mathbf{Z}, \mathbf{Z}; \mathcal{S}'))$ and $\mathrm{Tr}(\mathsf{K}(\mathbf{Z}, \mathbf{Z}; \mathcal{S}''))$ are i.i.d. random variables. Similarly for $\mathsf{K}(\boldsymbol{z}_i, \boldsymbol{z}_j; \mathcal{S}'), i, j \in [n]$. We assume the finite maximum of each random variable follows a reverse Weibull distribution and estimate their supremum then sum them together to get

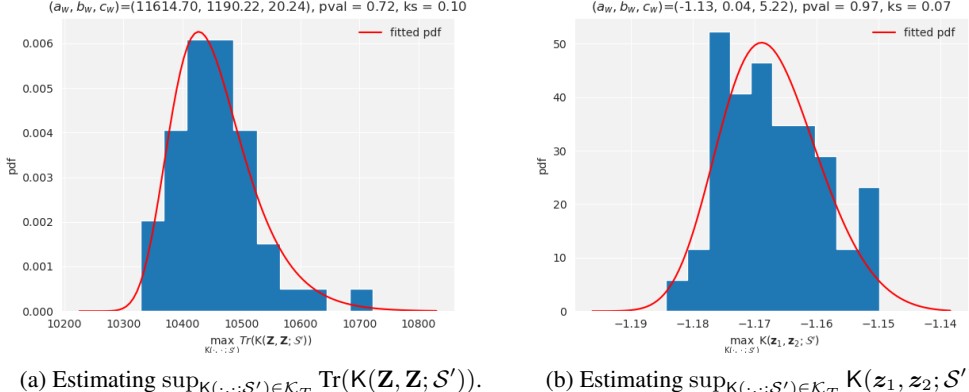

(a) Estimating $\sup_{\mathsf{K}(\cdot,\cdot;\mathcal{S}')\in\mathcal{K}_T} \mathrm{Tr}(\mathsf{K}(\mathbf{Z},\mathbf{Z};\mathcal{S}'))$.  (b) Estimating $\sup_{\mathsf{K}(\cdot,\cdot;\mathcal{S}')\in\mathcal{K}_T} \mathsf{K}(\mathbf{z}_1,\mathbf{z}_2;\mathcal{S}')$.

Figure A.7: Estimating the components in $U_1$ with $N_b = 50$ finite maximum over sets of size $N_g = 1000$. The estimated extreme values are close to the finite maximum, validating using a finite maximum as an estimate for our bound.

$U_1$. Figure A.7 shows estimating $\sup_{\mathsf{K}(\cdot,\cdot;\mathcal{S}')\in\mathcal{K}_T} \mathrm{Tr}(\mathsf{K}(\mathbf{Z},\mathbf{Z};\mathcal{S}'))$ and $\sup_{\mathsf{K}(\cdot,\cdot;\mathcal{S}')\in\mathcal{K}_T} \mathsf{K}(\mathbf{z}_1,\mathbf{z}_2;\mathcal{S}')$. The estimated extreme values are close to the finite maximum, validating using a finite maximum as an estimate for our bound. Due to the computational cost of estimating $n^2$ extreme values, we leave estimating $U_1$ as future work.

# B    Integrability of Loss Tangent Kernel

By the assumption that the loss is continuously differentiable, the loss gradient $\nabla_w \ell(\boldsymbol{w}(t), \boldsymbol{z})$ is continuous w.r.t. $\boldsymbol{w}(t)$. Together with $\boldsymbol{w}(t)$ is continuous w.r.t. $t$, $\nabla_w \ell(\boldsymbol{w}(t), \boldsymbol{z})$ is continuous w.r.t. $t$. After the inner product, the LTK $\bar{\mathsf{K}}(\boldsymbol{w}(t); \boldsymbol{z}, \boldsymbol{z}')$ is still continuous w.r.t. $t$. By the continuity of LTK on the compact set $[0, T]$, LTK is bounded and Riemann integrable on $[0, T]$. Therefore, the integral in LPK is well-defined.

For the full-batch gradient flow (1) we considered in this paper, $\boldsymbol{w}(t)$ is differentiable therefore continuous w.r.t. $t$ by the gradient flow equation. Hence, the integral in LPK is well-defined.

For stochastic gradient flow, the continuity of the path $\boldsymbol{w}(t)$ can be argued as follows. For each time interval $[t, t + 1]$, we assume that the same batch of data is used. Within this time interval, $\boldsymbol{w}(t)$ is continuous w.r.t. $t$. For the next time interval $[t + 1, t + 2]$, even if a different data batch is used, the gradient flow ODE initializes the $\boldsymbol{w}(t + 1)$ with the solution from the end point of the previous interval, $[t, t + 1]$. This ensures the continuity of $\boldsymbol{w}(t)$ across distinct time intervals. In short, the continuity of $\boldsymbol{w}(t)$ within each time interval, combined with the initialization of the ODE, will ensure the continuity of $\boldsymbol{w}(t)$ in the entire time interval $[0, T]$.

# C    Complete Proofs for Gradient Flow

## C.1    Proof of Theorem 2

**Theorem 2.** *Suppose $\boldsymbol{w}(T) = \boldsymbol{w}_T$ is a solution of (1) at time $T$ with initialization $\boldsymbol{w}(0) = \boldsymbol{w}_0$. Then for any $\boldsymbol{z} \in \mathcal{Z}$,*

$$\ell(\boldsymbol{w}_T, \boldsymbol{z}) = \sum_{i=1}^{n} -\frac{1}{n} \mathsf{K}_T(\boldsymbol{z}, \boldsymbol{z}_i; \mathcal{S}) + \ell(\boldsymbol{w}_0, \boldsymbol{z}),$$

*where $\mathsf{K}_T$ is defined in Definition 4.*

*Proof.* Consider the gradient flow:

$$\frac{d\boldsymbol{w}_t}{dt} = -\nabla_{\boldsymbol{w}} L_S(\boldsymbol{w}_t) = -\frac{1}{n} \sum_{i=1}^{n} \nabla_{\boldsymbol{w}} \ell(\boldsymbol{w}_t, \boldsymbol{z}_i). \tag{6}$$

For any differentiable loss function $\ell(\boldsymbol{w}, \boldsymbol{z})$, by chain rule,

$$\frac{d\ell(\boldsymbol{w}_t, \boldsymbol{z})}{dt} = \left\langle \nabla_{\boldsymbol{w}} \ell(\boldsymbol{w}_t, \boldsymbol{z}), \frac{d\boldsymbol{w}_t}{dt} \right\rangle. \tag{7}$$

Plug in the gradient flow expression of $\frac{d\boldsymbol{w}_t}{dt}$ in (6) into (7). We have

$$\begin{aligned}
\frac{d\ell(\boldsymbol{w}_t, \boldsymbol{z})}{dt} &= \langle \nabla_{\boldsymbol{w}} \ell(\boldsymbol{w}_t, \boldsymbol{z}), -\nabla_{\boldsymbol{w}} L_S(\boldsymbol{w}_t) \rangle \\
&= \left\langle \nabla_{\boldsymbol{w}} \ell(\boldsymbol{w}_t, \boldsymbol{z}), -\frac{1}{n} \sum_{i=1}^{n} \nabla_{\boldsymbol{w}} \ell(\boldsymbol{w}_t, \boldsymbol{z}_i) \right\rangle && \text{(by definition of } L_S(\boldsymbol{w}_t)) \\
&= -\frac{1}{n} \sum_{i=1}^{n} \langle \nabla_{\boldsymbol{w}} \ell(\boldsymbol{w}_t, \boldsymbol{z}), \nabla_{\boldsymbol{w}} \ell(\boldsymbol{w}_t, \boldsymbol{z}_i) \rangle && \text{(by the linearity of inner product)} \\
&= -\frac{1}{n} \sum_{i=1}^{n} \bar{\mathsf{K}}(\boldsymbol{w}_t; \boldsymbol{z}, \boldsymbol{z}_i). && \text{(by Definition 3 of the LTK)}
\end{aligned}$$

Integrate both sides from 0 to $T$ over the path $\boldsymbol{w}(t)$ taken by the parameters during the gradient flow,

$$\ell(\boldsymbol{w}_T, \boldsymbol{z}) - \ell(\boldsymbol{w}_0, \boldsymbol{z}) = \int_0^T -\frac{1}{n}\sum_{i=1}^n \bar{\mathsf{K}}(\boldsymbol{w}_t; \boldsymbol{z}, \boldsymbol{z}_i)dt$$

$$= -\frac{1}{n}\sum_{i=1}^n \int_0^T \bar{\mathsf{K}}(\boldsymbol{w}_t; \boldsymbol{z}, \boldsymbol{z}_i)dt \qquad \text{(By linearity of integration)}$$

$$= \sum_{i=1}^n -\frac{1}{n}\mathsf{K}_T(\boldsymbol{z}, \boldsymbol{z}_i; \mathcal{S}). \qquad \text{(By Definition 4 of LPK)}$$

Thus we have

$$\ell(\boldsymbol{w}_T, \boldsymbol{z}) = \sum_{i=1}^n -\frac{1}{n}\mathsf{K}_T(\boldsymbol{z}, \boldsymbol{z}_i; \mathcal{S}) + \ell(\boldsymbol{w}_0, \boldsymbol{z}).$$

Below, we prove that $\mathsf{K}_T(\boldsymbol{z}, \boldsymbol{z}'; \mathcal{S}) : \mathcal{Z} \times \mathcal{Z} \to \mathbb{R}$ is a valid kernel. We can show $\mathsf{K}_T(\boldsymbol{z}, \boldsymbol{z}'; \mathcal{S})$ is a valid kernel by proving $\mathsf{K}_T(\boldsymbol{z}, \boldsymbol{z}'; \mathcal{S})$ is continuous and the kernel matrix $\mathsf{K}_T(\mathbf{Z}, \mathbf{Z}; \mathcal{S}) \in \mathbb{R}^{n \times n}$ is positive semi-definite (PSD) for any finite subset of $\mathcal{Z}$, $\boldsymbol{z}_1, \cdots, \boldsymbol{z}_n \in \mathcal{Z}$ [53]. By Definition 4 of the loss path kernel,

$$\mathsf{K}_T(\boldsymbol{z}, \boldsymbol{z}'; \mathcal{S}) = \int_0^T \bar{\mathsf{K}}(\boldsymbol{w}_t; \boldsymbol{z}, \boldsymbol{z}')dt$$

$$= \int_0^T \langle \nabla_{\boldsymbol{w}}\ell(\boldsymbol{w}_t, \boldsymbol{z}), \nabla_{\boldsymbol{w}}\ell(\boldsymbol{w}_t, \boldsymbol{z}') \rangle \, dt.$$

Since $\ell(\boldsymbol{w}_t, \boldsymbol{z})$ is continuously differentiable for $\boldsymbol{w}_t$ and $\boldsymbol{z}$, $\nabla_{\boldsymbol{w}}\ell(\boldsymbol{w}_t, \boldsymbol{z})$ is continuous for $\boldsymbol{z}$. After the inner product and integration, it is still a continuous function. Thus $\mathsf{K}_T(\boldsymbol{z}, \boldsymbol{z}'; \mathcal{S})$ is a continuous function for $\boldsymbol{z}$.

Denote $\Phi_t(\boldsymbol{z}) = \nabla_{\boldsymbol{w}}\ell(\boldsymbol{w}_t, \boldsymbol{z})$, then by Definition 4 of the loss path kernel,

$$\mathsf{K}_T(\boldsymbol{z}, \boldsymbol{z}'; \mathcal{S}) = \int_0^T \bar{\mathsf{K}}(\boldsymbol{w}_t; \boldsymbol{z}, \boldsymbol{z}')dt$$

$$= \int_0^T \langle \nabla_{\boldsymbol{w}}\ell(\boldsymbol{w}_t, \boldsymbol{z}), \nabla_{\boldsymbol{w}}\ell(\boldsymbol{w}_t, \boldsymbol{z}') \rangle \, dt$$

$$= \int_0^T \langle \Phi_t(\boldsymbol{z}), \Phi_t(\boldsymbol{z}') \rangle \, dt.$$

For $\forall \boldsymbol{u} \in \mathbb{R}^n$ and $\forall \{\boldsymbol{z}_i\}_{i=1}^n \subseteq \mathcal{Z}$,

$$\boldsymbol{u}^T \mathsf{K}_T(\mathbf{Z}, \mathbf{Z}; \mathcal{S})\boldsymbol{u} = \sum_{i=1}^n \sum_{j=1}^n u_i u_j \mathsf{K}_T(\boldsymbol{z}_i, \boldsymbol{z}_j; \mathcal{S})$$

$$= \sum_{i=1}^n \sum_{j=1}^n u_i u_j \int_0^T \langle \Phi_t(\boldsymbol{z}_i), \Phi_t(\boldsymbol{z}_j) \rangle \, dt$$

$$= \int_0^T \sum_{i=1}^n \sum_{j=1}^n u_i u_j \langle \Phi_t(\boldsymbol{z}_i), \Phi_t(\boldsymbol{z}_j) \rangle \, dt$$

$$= \int_0^T \left\langle \sum_{i=1}^n u_i \Phi_t(\boldsymbol{z}_i), \sum_{j=1}^n u_j \Phi_t(\boldsymbol{z}_j) \right\rangle \, dt$$

$$= \int_0^T \left\| \sum_i^n u_i \Phi_t(\boldsymbol{z}_i) \right\|^2 \, dt$$

$$\geq 0.$$

Thus the matrix $\mathsf{K}_T(\mathbf{Z}, \mathbf{Z}; \mathcal{S})$ is PSD and $\mathsf{K}_T(\boldsymbol{z}, \boldsymbol{z}'; \mathcal{S})$ is therefore a valid kernel.

$\square$

## C.2 Proof of Theorem 3

**Theorem 3.** $\hat{\mathcal{R}}_{\mathcal{S}}(\mathcal{G}_T) \leq \min\{U_1, U_2\}$. *Here*

$$U_1 = \frac{B}{n} \sqrt{\sup_{\mathsf{K}(\cdot,\cdot;\mathcal{S}')\in\mathcal{K}_T} \mathrm{Tr}(\mathsf{K}(\mathbf{Z},\mathbf{Z};\mathcal{S}')) + \sum_{i\neq j} \Delta(\boldsymbol{z}_i,\boldsymbol{z}_j)},$$

$$U_2 = \inf_{\epsilon>0} \left( \frac{\epsilon}{n} + \sqrt{\frac{2\ln\mathcal{N}(\mathcal{G}_T^{\mathcal{S}},\epsilon,\|\|_1)}{n}} \right),$$

*where $\mathcal{G}_T^{\mathcal{S}} = \{g(\mathbf{Z}) = (g(\boldsymbol{z}_1),\ldots,g(\boldsymbol{z}_n)) : g \in \mathcal{G}_T\}$, $\mathcal{N}(\mathcal{G}_T^{\mathcal{S}},\epsilon,\|\|_1)$ is the covering number of $\mathcal{G}_T^{\mathcal{S}}$ with the $\ell_1$-norm and*

$$\Delta(\boldsymbol{z}_i,\boldsymbol{z}_j) = \frac{1}{2} \left[ \sup_{\mathsf{K}(\cdot,\cdot;\mathcal{S}')\in\mathcal{K}_T} \mathsf{K}(\boldsymbol{z}_i,\boldsymbol{z}_j;\mathcal{S}') - \inf_{\mathsf{K}(\cdot,\cdot;\mathcal{S}')\in\mathcal{K}_T} \mathsf{K}(\boldsymbol{z}_i,\boldsymbol{z}_j;\mathcal{S}') \right].$$

### C.2.1 Proof of $U_1$

*Proof.* Recall the definition of $\mathcal{G}_T$,

$$\mathcal{G}_T = \{g(\boldsymbol{z}) = \sum_{i=1}^{n} -\frac{1}{n}\mathsf{K}(\boldsymbol{z},\boldsymbol{z}_i';\mathcal{S}') + \ell(\boldsymbol{w}_0,\boldsymbol{z}) : \mathsf{K}(\cdot,\cdot;\mathcal{S}') \in \mathcal{K}_T\}.$$

where

$$\mathcal{K}_T = \{\mathsf{K}_T(\cdot,\cdot;\mathcal{S}') : \mathcal{S}' \in \mathsf{supp}(\mu^{\otimes n}), \frac{1}{n^2}\sum_{i,j}\mathsf{K}_T(\boldsymbol{z}_i',\boldsymbol{z}_j';\mathcal{S}') \leq B^2\}.$$

Suppose $\mathsf{K}(\boldsymbol{z},\boldsymbol{z}';\mathcal{S}') = \langle \Phi_{\mathcal{S}'}(\boldsymbol{z}), \Phi_{\mathcal{S}'}(\boldsymbol{z}')\rangle$. Define

$$\mathcal{G}_T^U = \{g(\boldsymbol{z}) = \langle \boldsymbol{\beta}, \Phi_{\mathcal{S}'}(\boldsymbol{z})\rangle + \ell(\boldsymbol{w}_0,\boldsymbol{z}) : \|\boldsymbol{\beta}\| \leq B, \mathsf{K}(\cdot,\cdot;\mathcal{S}') \in \mathcal{K}_T\}.$$

We first show $\mathcal{G}_T \subseteq \mathcal{G}_T^U$. For $\forall g(\boldsymbol{z}) \in \mathcal{G}_T$,

$$
\begin{aligned}
g(\boldsymbol{z}) &= \sum_{i=1}^{n} -\frac{1}{n}\mathsf{K}(\boldsymbol{z},\boldsymbol{z}_i';\mathcal{S}') + \ell(\boldsymbol{w}_0,\boldsymbol{z}) \\
&= \sum_{i=1}^{n} -\frac{1}{n}\langle \Phi_{\mathcal{S}'}(\boldsymbol{z}), \Phi_{\mathcal{S}'}(\boldsymbol{z}_i')\rangle + \ell(\boldsymbol{w}_0,\boldsymbol{z}) \quad \text{(by definition } \mathsf{K}(\boldsymbol{z},\boldsymbol{z}';\mathcal{S}') = \langle \Phi_{\mathcal{S}'}(\boldsymbol{z}), \Phi_{\mathcal{S}'}(\boldsymbol{z}')\rangle) \\
&= \left\langle \Phi_{\mathcal{S}'}(\boldsymbol{z}), \sum_{i=1}^{n} -\frac{1}{n}\Phi_{\mathcal{S}'}(\boldsymbol{z}_i')\right\rangle + \ell(\boldsymbol{w}_0,\boldsymbol{z}) \\
&= \langle \Phi_{\mathcal{S}'}(\boldsymbol{z}), \boldsymbol{\beta}_{\mathcal{S}'}\rangle + \ell(\boldsymbol{w}_0,\boldsymbol{z}) \quad\quad\quad\quad (\text{denote } \boldsymbol{\beta}_{\mathcal{S}'} = \sum_{i=1}^{n} -\frac{1}{n}\Phi_{\mathcal{S}'}(\boldsymbol{z}_i')) \\
&= \langle \boldsymbol{\beta}_{\mathcal{S}'}, \Phi_{\mathcal{S}'}(\boldsymbol{z})\rangle + \ell(\boldsymbol{w}_0,\boldsymbol{z})
\end{aligned}
$$

We know by definition of $\mathcal{G}_T$, $\|\boldsymbol{\beta}_{\mathcal{S}'}\|^2 = \frac{1}{n^2}\sum_{i,j}\mathsf{K}(\boldsymbol{z}_i',\boldsymbol{z}_j';\mathcal{S}') \leq B^2$. Thus $g(\boldsymbol{z}) \in \mathcal{G}_T^U$. Since $\forall g(\boldsymbol{z}) \in \mathcal{G}_T$, $g(\boldsymbol{z}) \in \mathcal{G}_T^U$, $\mathcal{G}_T \subseteq \mathcal{G}_T^U$. But $\mathcal{G}_T^U$ is strictly larger than $\mathcal{G}_T$ because $\boldsymbol{\beta}_{\mathcal{S}'}$ is a fixed vector for a fixed $\mathsf{K}(\cdot,\cdot;\mathcal{S}')$ while $\boldsymbol{\beta}$ in $\mathcal{G}_T^U$ is a vector of any direction.

Then by the property of Rademacher complexity,

$$
\begin{aligned}
\hat{\mathcal{R}}_{\mathcal{S}}(\mathcal{G}_T) &\le \hat{\mathcal{R}}_{\mathcal{S}}(\mathcal{G}_T^U) \\
&= \frac{1}{n} \mathbb{E}_{\boldsymbol{\sigma}} \left[ \sup_{g \in \mathcal{G}_T^U} \sum_{i=1}^n \sigma_i g(\boldsymbol{z}_i) \right] \\
&= \frac{1}{n} \mathbb{E}_{\boldsymbol{\sigma}} \left[ \sup_{\mathsf{K}(\cdot,\cdot;\mathcal{S}') \in \mathcal{K}_T} \sum_{i=1}^n \sigma_i \left( \langle \boldsymbol{\beta}, \Phi_{\mathcal{S}'}(\boldsymbol{z}_i) \rangle + \ell(\boldsymbol{w}_0, \boldsymbol{z}_i) \right) \right] \\
&= \frac{1}{n} \mathbb{E}_{\boldsymbol{\sigma}} \left[ \sup_{\mathsf{K}(\cdot,\cdot;\mathcal{S}') \in \mathcal{K}_T} \sum_{i=1}^n \sigma_i \langle \boldsymbol{\beta}, \Phi_{\mathcal{S}'}(\boldsymbol{z}_i) \rangle \right] + \frac{1}{n} \mathbb{E}_{\boldsymbol{\sigma}} \left[ \sup_{\mathsf{K}(\cdot,\cdot;\mathcal{S}') \in \mathcal{K}_T} \sum_{i=1}^n \sigma_i \ell(\boldsymbol{w}_0, \boldsymbol{z}_i) \right] \\
&= \frac{1}{n} \mathbb{E}_{\boldsymbol{\sigma}} \left[ \sup_{\mathsf{K}(\cdot,\cdot;\mathcal{S}') \in \mathcal{K}_T} \left\langle \boldsymbol{\beta}, \sum_{i=1}^n \sigma_i \Phi_{\mathcal{S}'}(\boldsymbol{z}_i) \right\rangle \right] \\
&= \frac{B}{n} \mathbb{E}_{\boldsymbol{\sigma}} \left[ \sup_{\mathsf{K}(\cdot,\cdot;\mathcal{S}') \in \mathcal{K}_T} \left\| \sum_{i=1}^n \sigma_i \Phi_{\mathcal{S}'}(\boldsymbol{z}_i) \right\| \right] \\
&= \frac{B}{n} \mathbb{E}_{\boldsymbol{\sigma}} \left[ \sup_{\mathsf{K}(\cdot,\cdot;\mathcal{S}') \in \mathcal{K}_T} \left( \sum_{i=1}^n \sum_{j=1}^n \sigma_i \sigma_j \mathsf{K}(\boldsymbol{z}_i, \boldsymbol{z}_j; \mathcal{S}') \right)^{\frac{1}{2}} \right] \\
&= \frac{B}{n} \mathbb{E}_{\boldsymbol{\sigma}} \left[ \left( \sup_{\mathsf{K}(\cdot,\cdot;\mathcal{S}') \in \mathcal{K}_T} \sum_{i=1}^n \sum_{j=1}^n \sigma_i \sigma_j \mathsf{K}(\boldsymbol{z}_i, \boldsymbol{z}_j; \mathcal{S}') \right)^{\frac{1}{2}} \right] \\
&\le \frac{B}{n} \left( \mathbb{E}_{\boldsymbol{\sigma}} \left[ \sup_{\mathsf{K}(\cdot,\cdot;\mathcal{S}') \in \mathcal{K}_T} \sum_{i=1}^n \sum_{j=1}^n \sigma_i \sigma_j \mathsf{K}(\boldsymbol{z}_i, \boldsymbol{z}_j; \mathcal{S}') \right] \right)^{\frac{1}{2}} \quad \text{(Jensen's inequality)} \\
&= \frac{B}{n} \left( \mathbb{E}_{\boldsymbol{\sigma}} \left[ \sup_{\mathsf{K}(\cdot,\cdot;\mathcal{S}') \in \mathcal{K}_T} \left( \sum_{i=1}^n \mathsf{K}(\boldsymbol{z}_i, \boldsymbol{z}_i) + \sum_{i \ne j} \sigma_i \sigma_j \mathsf{K}(\boldsymbol{z}_i, \boldsymbol{z}_j; \mathcal{S}') \right) \right] \right)^{\frac{1}{2}} \\
&\le \frac{B}{n} \left( \mathbb{E}_{\boldsymbol{\sigma}} \left[ \sup_{\mathsf{K}(\cdot,\cdot;\mathcal{S}') \in \mathcal{K}_T} \sum_{i=1}^n \mathsf{K}(\boldsymbol{z}_i, \boldsymbol{z}_i) + \sup_{\mathsf{K}(\cdot,\cdot;\mathcal{S}') \in \mathcal{K}_T} \sum_{i \ne j} \sigma_i \sigma_j \mathsf{K}(\boldsymbol{z}_i, \boldsymbol{z}_j; \mathcal{S}') \right] \right)^{\frac{1}{2}} \\
&= \frac{B}{n} \left( \sup_{\mathsf{K}(\cdot,\cdot;\mathcal{S}') \in \mathcal{K}_T} \mathrm{Tr}(\mathsf{K}(\mathbf{Z}, \mathbf{Z}; \mathcal{S}')) + \mathbb{E}_{\boldsymbol{\sigma}} \left[ \sup_{\mathsf{K}(\cdot,\cdot;\mathcal{S}') \in \mathcal{K}_T} \sum_{i \ne j} \sigma_i \sigma_j \mathsf{K}(\boldsymbol{z}_i, \boldsymbol{z}_j; \mathcal{S}') \right] \right)^{\frac{1}{2}} .
\end{aligned}
$$

The second term in the square root is

$$\mathbb{E}_{\boldsymbol{\sigma}}\left[\sup_{\mathsf{K}(\cdot,\cdot;\mathcal{S}')\in\mathcal{K}_T}\sum_{i\neq j}\sigma_i\sigma_j\mathsf{K}(\boldsymbol{z}_i,\boldsymbol{z}_j;\mathcal{S}')\right]$$

$$\leq\mathbb{E}_{\boldsymbol{\sigma}}\left[\sum_{i\neq j}\sup_{\mathsf{K}(\cdot,\cdot;\mathcal{S}')\in\mathcal{K}_T}\sigma_i\sigma_j\mathsf{K}(\boldsymbol{z}_i,\boldsymbol{z}_j;\mathcal{S}')\right]$$

$$=\sum_{i\neq j}\mathbb{E}_{\boldsymbol{\sigma}}\left[\sup_{\mathsf{K}(\cdot,\cdot;\mathcal{S}')\in\mathcal{K}_T}\sigma_i\sigma_j\mathsf{K}(\boldsymbol{z}_i,\boldsymbol{z}_j;\mathcal{S}')\right]$$

$$=\sum_{i\neq j}\left(\mathbb{P}\left(\sigma_i\sigma_j=+1\right)\left[\sup_{\mathsf{K}(\cdot,\cdot;\mathcal{S}')\in\mathcal{K}_T}\mathsf{K}(\boldsymbol{z}_i,\boldsymbol{z}_j;\mathcal{S}')\right]+\mathbb{P}\left(\sigma_i\sigma_j=-1\right)\left[\sup_{\mathsf{K}(\cdot,\cdot;\mathcal{S}')\in\mathcal{K}_T}-\mathsf{K}(\boldsymbol{z}_i,\boldsymbol{z}_j;\mathcal{S}')\right]\right)$$

$$=\sum_{i\neq j}\frac{1}{2}\left[\sup_{\mathsf{K}(\cdot,\cdot;\mathcal{S}')\in\mathcal{K}_T}\mathsf{K}(\boldsymbol{z}_i,\boldsymbol{z}_j;\mathcal{S}')-\inf_{\mathsf{K}(\cdot,\cdot;\mathcal{S}')\in\mathcal{K}_T}\mathsf{K}(\boldsymbol{z}_i,\boldsymbol{z}_j;\mathcal{S}')\right]$$

$$=\sum_{i\neq j}\Delta(\boldsymbol{z}_i,\boldsymbol{z}_j)$$

where we define $\Delta(\boldsymbol{z}_i,\boldsymbol{z}_j)=\frac{1}{2}\left[\sup_{\mathsf{K}(\cdot,\cdot;\mathcal{S}')\in\mathcal{K}_T}\mathsf{K}(\boldsymbol{z}_i,\boldsymbol{z}_j;\mathcal{S}')-\inf_{\mathsf{K}(\cdot,\cdot;\mathcal{S}')\in\mathcal{K}_T}\mathsf{K}(\boldsymbol{z}_i,\boldsymbol{z}_j;\mathcal{S}')\right]$. Thus in total,

$$\hat{\mathcal{R}}_{\mathcal{S}}(\mathcal{G}_T)\leq\hat{\mathcal{R}}_{\mathcal{S}}(\mathcal{G}_T^U)\leq\frac{B}{n}\left(\sup_{\mathsf{K}(\cdot,\cdot;\mathcal{S}')\in\mathcal{K}_T}\mathrm{Tr}(\mathsf{K}(\mathbf{Z},\mathbf{Z};\mathcal{S}'))+\sum_{i\neq j}\Delta(\boldsymbol{z}_i,\boldsymbol{z}_j)\right)^{\frac{1}{2}}$$

$\square$

### C.2.2 Proof of $U_2$

*Proof.* To simplify the notation, denote $\mathsf{K}_{\mathcal{S}'}(\cdot,\cdot)=\mathsf{K}_T(\cdot,\cdot;\mathcal{S}')$. Denote $g_{\mathsf{K}_{\mathcal{S}'}}(\boldsymbol{z})=\sum_{i=1}^{n}-\frac{1}{n}\mathsf{K}_{\mathcal{S}'}(\boldsymbol{z},\boldsymbol{z}_i')+\ell(\boldsymbol{w}_0,\boldsymbol{z})$ for a fixed $\mathsf{K}_{\mathcal{S}'}$, which is a singleton hypothesis class. Then $\mathcal{G}_T$ is a union set of such function classes with different kernels,

$$\mathcal{G}_T=\left\{g_{\mathsf{K}_{\mathcal{S}'}}(\boldsymbol{z})=\sum_{i=1}^{n}-\frac{1}{n}\mathsf{K}_{\mathcal{S}'}(\boldsymbol{z},\boldsymbol{z}_i')+\ell(\boldsymbol{w}_0,\boldsymbol{z}):\mathsf{K}_{\mathcal{S}'}\in\mathcal{K}_T\right\}=\bigcup_{\mathsf{K}_{\mathcal{S}'}\in\mathcal{K}_T}\{g_{\mathsf{K}_{\mathcal{S}'}}(\boldsymbol{z})\}$$

Then we can rewrite $\mathcal{G}_T^{\mathcal{S}}=\left\{g_{\mathsf{K}_{\mathcal{S}'}}(\mathbf{Z})=(g_{\mathsf{K}_{\mathcal{S}'}}(\boldsymbol{z}_1),\ldots,g_{\mathsf{K}_{\mathcal{S}'}}(\boldsymbol{z}_n)):\mathsf{K}_{\mathcal{S}'}\in\mathcal{K}_T\right\}$. Suppose $\mathcal{G}_T^{\epsilon}\subseteq\mathcal{G}_T^{\mathcal{S}}$ is a minimal $\epsilon$-cover of $\mathcal{G}_T^{\mathcal{S}}$ with $\|\|_1$. Denote $\widetilde{g}_{\mathsf{K}_{\mathcal{S}'}}(\mathbf{Z})=(\widetilde{g}_{\mathsf{K}_{\mathcal{S}'}}(\boldsymbol{z}_1),\ldots,\widetilde{g}_{\mathsf{K}_{\mathcal{S}'}}(\boldsymbol{z}_n))\in\mathcal{G}_T^{\epsilon}$ as the closest element to $g_{\mathsf{K}_{\mathcal{S}'}}(\mathbf{Z})\in\mathcal{G}_T^{\mathcal{S}}$ and $\widetilde{g}_{\mathsf{K}_{\mathcal{S}'}}(\boldsymbol{z})=\sum_{i=1}^{n}-\frac{1}{n}\widetilde{K}_{\mathcal{S}'}(\boldsymbol{z},\boldsymbol{z}_i')+\ell(\boldsymbol{w}_0,\boldsymbol{z})=g_{\widetilde{K}_{\mathcal{S}'}}(\boldsymbol{z})$ with $\widetilde{K}_{\mathcal{S}'}\in\mathcal{K}_T$. Denote the set of all $\widetilde{K}_{\mathcal{S}'}$ as $\mathcal{K}_T^{\epsilon}$. Since one $\widetilde{g}_{\mathsf{K}_{\mathcal{S}'}}$ corresponds to one $\widetilde{K}_{\mathcal{S}'}$, $|\mathcal{K}_T^{\epsilon}|=|\mathcal{G}_T^{\epsilon}|=\mathcal{N}(\mathcal{G}_T^{\mathcal{S}},\epsilon,\|\|_1)$.

Based on the above, we can write the Rademacher complexity of $\mathcal{G}_T$ as

$$\hat{\mathcal{R}}_{\mathcal{S}}(\mathcal{G}_T)=\frac{1}{n}\mathbb{E}_{\boldsymbol{\sigma}}\left[\sup_{\mathsf{K}_{\mathcal{S}'}\in\mathcal{K}_T}\sum_{i=1}^{n}\sigma_i g_{\mathsf{K}_{\mathcal{S}'}}(\boldsymbol{z}_i)\right].$$

For any $\lambda>0$, by Jensen's inequality,

$$e^{\lambda\hat{\mathcal{R}}_{\mathcal{S}}(\mathcal{G}_T)}\leq\mathbb{E}_{\boldsymbol{\sigma}}\left[e^{\lambda\left[\frac{1}{n}\sup_{\mathsf{K}_{\mathcal{S}'}\in\mathcal{K}_T}\sum_{i=1}^{n}\sigma_i g_{\mathsf{K}_{\mathcal{S}'}}(\boldsymbol{z}_i)\right]}\right]$$

$$=\mathbb{E}_{\boldsymbol{\sigma}}\left[\sup_{\mathsf{K}_{\mathcal{S}'}\in\mathcal{K}_T}e^{\lambda\left[\frac{1}{n}\sum_{i=1}^{n}\sigma_i g_{\mathsf{K}_{\mathcal{S}'}}(\boldsymbol{z}_i)\right]}\right]. \tag{8}$$

Utilizing the definition of $\epsilon$-covering, the quantity on the exponent in Eq. (8) is

$$\frac{1}{n}\sum_{i=1}^{n}\sigma_i g_{\mathsf{K}_{\mathcal{S}'}}(\boldsymbol{z}_i)$$

$$=\frac{1}{n}\sum_{i=1}^{n}\sigma_i\left(g_{\mathsf{K}_{\mathcal{S}'}}(\boldsymbol{z}_i)-\widetilde{g}_{\mathsf{K}_{\mathcal{S}'}}(\boldsymbol{z}_i)+\widetilde{g}_{\mathsf{K}_{\mathcal{S}'}}(\boldsymbol{z}_i)\right)$$

$$=\frac{1}{n}\sum_{i=1}^{n}\sigma_i\widetilde{g}_{\mathsf{K}_{\mathcal{S}'}}(\boldsymbol{z}_i)+\frac{1}{n}\sum_{i=1}^{n}\sigma_i\left(g_{\mathsf{K}_{\mathcal{S}'}}(\boldsymbol{z}_i)-\widetilde{g}_{\mathsf{K}_{\mathcal{S}'}}(\boldsymbol{z}_i)\right)$$

$$\leq\frac{1}{n}\sum_{i=1}^{n}\sigma_i\widetilde{g}_{\mathsf{K}_{\mathcal{S}'}}(\boldsymbol{z}_i)+\frac{1}{n}\sum_{i=1}^{n}\left|g_{\mathsf{K}_{\mathcal{S}'}}(\boldsymbol{z}_i)-\widetilde{g}_{\mathsf{K}_{\mathcal{S}'}}(\boldsymbol{z}_i)\right|$$

$$=\frac{1}{n}\sum_{i=1}^{n}\sigma_i\widetilde{g}_{\mathsf{K}_{\mathcal{S}'}}(\boldsymbol{z}_i)+\frac{1}{n}\left\|g_{\mathsf{K}_{\mathcal{S}'}}(\mathbf{Z})-\widetilde{g}_{\mathsf{K}_{\mathcal{S}'}}(\mathbf{Z})\right\|_1$$

$$\leq\frac{1}{n}\sum_{i=1}^{n}\sigma_i\widetilde{g}_{\mathsf{K}_{\mathcal{S}'}}(\boldsymbol{z}_i)+\frac{\epsilon}{n}. \qquad\text{(by the definition of the $\epsilon$-covering)}$$

Substitute this inequality into Eq. (8). We have

$$e^{\lambda\hat{\mathcal{R}}_{\mathcal{S}}(\mathcal{G}_T)}\leq\mathbb{E}_{\boldsymbol{\sigma}}\left[\sup_{\mathsf{K}_{\mathcal{S}'}\in\mathcal{K}_T}e^{\lambda\left[\frac{1}{n}\sum_{i=1}^{n}\sigma_i\widetilde{g}_{\mathsf{K}_{\mathcal{S}'}}(\boldsymbol{z}_i)+\frac{\epsilon}{n}\right]}\right]$$

$$=e^{\frac{\lambda\epsilon}{n}}\mathbb{E}_{\boldsymbol{\sigma}}\left[\sup_{\mathsf{K}_{\mathcal{S}'}\in\mathcal{K}_T}e^{\lambda\left[\frac{1}{n}\sum_{i=1}^{n}\sigma_i g_{\widetilde{K}_{\mathcal{S}'}}(\boldsymbol{z}_i)\right]}\right] \qquad (\widetilde{g}_{\mathsf{K}_{\mathcal{S}'}}(\boldsymbol{z}_i)=g_{\widetilde{K}_{\mathcal{S}'}}(\boldsymbol{z}_i))$$

$$\overset{(i)}{=}e^{\frac{\lambda\epsilon}{n}}\mathbb{E}_{\boldsymbol{\sigma}}\left[\max_{\widetilde{K}_{\mathcal{S}'}\in\mathcal{K}_T^{\epsilon}}e^{\lambda\left[\frac{1}{n}\sum_{i=1}^{n}\sigma_i g_{\widetilde{K}_{\mathcal{S}'}}(\boldsymbol{z}_i)\right]}\right]$$

$$\leq e^{\frac{\lambda\epsilon}{n}}\sum_{\widetilde{K}_{\mathcal{S}'}\in\mathcal{K}_T^{\epsilon}}\mathbb{E}_{\boldsymbol{\sigma}}\left[e^{\lambda\left[\frac{1}{n}\sum_{i=1}^{n}\sigma_i g_{\widetilde{K}_{\mathcal{S}'}}(\boldsymbol{z}_i)\right]}\right] \qquad (9)$$

where $(i)$ is because we only use the $g_{\widetilde{K}_{\mathcal{S}'}}$ instead of $g_{\mathsf{K}_{\mathcal{S}'}}$, which corresponds to $\widetilde{K}_{\mathcal{S}'}\in\mathcal{K}_T^{\epsilon}$ instead of $\mathsf{K}_{\mathcal{S}'}\in\mathcal{K}_T$, so it is equivalent to take the maximum over $\widetilde{K}_{\mathcal{S}'}\in\mathcal{K}_T^{\epsilon}$. Denote $\xi(\sigma_1,\ldots,\sigma_n)=\frac{1}{n}\sum_{i=1}^{n}\sigma_i g_{\widetilde{K}_{\mathcal{S}'}}(\boldsymbol{z}_i)$. Note $g_{\widetilde{K}_{\mathcal{S}'}}(\boldsymbol{z}_i)\in[0,1]$ for $i\in[n]$. Then for all $i\in[n]$,

$$\sup_{\boldsymbol{\sigma}}|\xi(\sigma_1,\cdots,\sigma_i,\cdots,\sigma_n)-\xi(\sigma_1,\cdots,-\sigma_i,\cdots,\sigma_n)|$$

$$\leq\sup_{\boldsymbol{\sigma}}\left|\frac{2}{n}\sigma_i g_{\widetilde{K}_{\mathcal{S}'}}(\boldsymbol{z}_i)\right|$$

$$\leq\frac{2}{n}.$$

Thus $\xi(\sigma_1,\cdots,\sigma_n)$ satisfies the bounded difference property. Let $c_i=\frac{2}{n}$. By [36, 10], $\xi$ is a sub-Gaussian variable and satisfies

$$\mathbb{E}\left[e^{\lambda\xi}\right]\leq e^{\frac{\lambda^2}{8}\sum_{i=1}^{n}c_i^2}e^{\lambda\mathbb{E}[\xi]}$$

$$=e^{\frac{\lambda^2}{2n}}e^{\lambda\mathbb{E}[\xi]}.$$

Since $g_{\widetilde{K}_{\mathcal{S}'}}$ is a singleton function, $\mathbb{E}[\xi]=\frac{1}{n}\mathbb{E}_{\boldsymbol{\sigma}}\left[\sum_{i=1}^{n}\sigma_i g_{\widetilde{K}_{\mathcal{S}'}}(\boldsymbol{z}_i)\right]=0$. Thus

$$\mathbb{E}\left[e^{\lambda\xi}\right]\leq e^{\frac{\lambda^2}{2n}}.$$

Take this into Eq. (9), in total, we have

$$e^{\lambda\hat{\mathcal{R}}_{\mathcal{S}}(\mathcal{G}_T)}\leq e^{\frac{\lambda\epsilon}{n}}\sum_{\widetilde{K}_{\mathcal{S}'}\in\mathcal{K}_T^{\epsilon}}e^{\frac{\lambda^2}{2n}}=e^{\frac{\lambda\epsilon}{n}+\frac{\lambda^2}{2n}}|\mathcal{K}_T^{\epsilon}|.$$

Take the logarithm on both sides of the equation,

$$\lambda \hat{\mathcal{R}}_{\mathcal{S}}(\mathcal{G}_T) \leq \frac{\lambda \epsilon}{n} + \frac{\lambda^2}{2n} + \ln |\mathcal{K}_T^{\epsilon}|.$$

Divide $\lambda$ on both sides,

$$\hat{\mathcal{R}}_{\mathcal{S}}(\mathcal{G}_T) \leq \frac{\epsilon}{n} + \frac{\lambda}{2n} + \frac{1}{\lambda} \ln |\mathcal{K}_T^{\epsilon}|$$

By taking $\lambda = \sqrt{2n \ln |\mathcal{K}_T^{\epsilon}|}$, we get

$$\hat{\mathcal{R}}_{\mathcal{S}}(\mathcal{G}_T) \leq \frac{\epsilon}{n} + \sqrt{\frac{2 \ln |\mathcal{K}_T^{\epsilon}|}{n}}.$$

Take the infimum over $\epsilon > 0$ and note $|\mathcal{K}_T^{\epsilon}| = |\mathcal{G}_T^{\epsilon}| = \mathcal{N}(\mathcal{G}_T^{\mathcal{S}}, \epsilon, \|\|_1)$. We get

$$\hat{\mathcal{R}}_{\mathcal{S}}(\mathcal{G}_T) \leq \inf_{\epsilon > 0} \left( \frac{\epsilon}{n} + \sqrt{\frac{2 \ln \mathcal{N}(\mathcal{G}_T^{\mathcal{S}}, \epsilon, \|\|_1)}{n}} \right).$$

$\square$

## C.3 A lower bound of $\hat{\mathcal{R}}_{\mathcal{S}}(\mathcal{G}_T^U)$

Here we give a lower bound of $\hat{\mathcal{R}}_{\mathcal{S}}(\mathcal{G}_T^U)$. Similar lower bounds for a linear model were proved in [4, 6] without the supremum. Our lower bound match the trace term in the upper bound $U_1$, which shows the upper bound $U_1$ is relatively tight.

**Theorem 7.**

$$\hat{\mathcal{R}}_{\mathcal{S}}(\mathcal{G}_T^U) \geq \frac{B}{\sqrt{2}n} \sup_{\mathsf{K}_{\mathcal{S}'} \in \mathcal{K}_T} \sqrt{\mathrm{Tr}(\mathsf{K}_{\mathcal{S}'}(\mathbf{Z}, \mathbf{Z}))}.$$

*Proof.* Recall

$$\mathcal{G}_T^U = \{g(\boldsymbol{z}) = \langle \boldsymbol{\beta}, \Phi_{\mathcal{S}'}(\boldsymbol{z}) \rangle + \ell(\boldsymbol{w}_0, \boldsymbol{z}) : \|\boldsymbol{\beta}\| \leq B, \mathsf{K}_{\mathcal{S}'} \in \mathcal{K}_T\}.$$

The Rademacher complexity of $\mathcal{G}_T^U$ is

$$\hat{\mathcal{R}}_{\mathcal{S}}(\mathcal{G}_T^U) = \frac{1}{n} \mathbb{E}_{\boldsymbol{\sigma}} \left[ \sup_{g \in \mathcal{G}_T^U} \sum_{i=1}^n \sigma_i g(\boldsymbol{z}_i) \right]$$

$$= \frac{1}{n} \mathbb{E}_{\boldsymbol{\sigma}} \left[ \sup_{\mathsf{K}_{\mathcal{S}'} \in \mathcal{K}_T} \sup_{\|\boldsymbol{\beta}\| \leq B} \sum_{i=1}^n \sigma_i \left( \langle \boldsymbol{\beta}, \Phi_{\mathcal{S}'}(\boldsymbol{z}_i) \rangle + \ell(\boldsymbol{w}_0, \boldsymbol{z}_i) \right) \right]$$

$$= \frac{1}{n} \mathbb{E}_{\boldsymbol{\sigma}} \left[ \sup_{\mathsf{K}_{\mathcal{S}'} \in \mathcal{K}_T} \sup_{\|\boldsymbol{\beta}\| \leq B} \left\langle \boldsymbol{\beta}, \sum_{i=1}^n \sigma_i \Phi_{\mathcal{S}'}(\boldsymbol{z}_i) \right\rangle \right] + \mathbb{E}_{\boldsymbol{\sigma}} \left[ \sum_{i=1}^n \sigma_i \ell(\boldsymbol{w}_0, \boldsymbol{z}_i) \right]$$

$$= \frac{1}{n} \mathbb{E}_{\boldsymbol{\sigma}} \left[ \sup_{\mathsf{K}_{\mathcal{S}'} \in \mathcal{K}_T} \sup_{\|\boldsymbol{\beta}\| \leq B} \left\langle \boldsymbol{\beta}, \sum_{i=1}^n \sigma_i \Phi_{\mathcal{S}'}(\boldsymbol{z}_i) \right\rangle \right]$$

$$= \frac{B}{n} \mathbb{E}_{\boldsymbol{\sigma}} \left[ \sup_{\mathsf{K}_{\mathcal{S}'} \in \mathcal{K}_T} \left\| \sum_{i=1}^n \sigma_i \Phi_{\mathcal{S}'}(\boldsymbol{z}_i) \right\| \right] \qquad \text{(dual norm property)}$$

$$\geq \frac{B}{n} \sup_{\mathsf{K}_{\mathcal{S}'} \in \mathcal{K}_T} \mathbb{E}_{\boldsymbol{\sigma}} \left[ \left\| \sum_{i=1}^n \sigma_i \Phi_{\mathcal{S}'}(\boldsymbol{z}_i) \right\| \right]$$

$$\geq \frac{B}{n} \sup_{\mathsf{K}_{\mathcal{S}'} \in \mathcal{K}_T} \left\| \mathbb{E}_{\boldsymbol{\sigma}} \left[ \sum_{i=1}^n \sigma_i \Phi_{\mathcal{S}'}(\boldsymbol{z}_i) \right] \right\| \qquad \text{(norm sub-additivity)}$$

$$= \frac{B}{n} \sup_{\mathsf{K}_{\mathcal{S}'} \in \mathcal{K}_T} \left( \sum_{j \in \mathbb{N}_+} \left( \mathbb{E}_{\boldsymbol{\sigma}} \left[ \left| \sum_{i=1}^n \sigma_i [\Phi_{\mathcal{S}'}(\boldsymbol{z}_i)]_j \right| \right] \right)^2 \right)^{\frac{1}{2}} \qquad \text{(by the definition of 2-norm)}$$

$$\geq \frac{B}{n} \sup_{\mathsf{K}_{\mathcal{S}'} \in \mathcal{K}_T} \left( \sum_{j \in \mathbb{N}_+} \left( \frac{1}{\sqrt{2}} \left| \sum_{i=1}^n [\Phi_{\mathcal{S}'}(\boldsymbol{z}_i)]_j^2 \right|^{\frac{1}{2}} \right)^2 \right)^{\frac{1}{2}} \qquad \text{(Khintchine-Kahane inequality)}$$

$$= \frac{B}{\sqrt{2}n} \sup_{\mathsf{K}_{\mathcal{S}'} \in \mathcal{K}_T} \left( \sum_{j \in \mathbb{N}_+} \left| \sum_{i=1}^n [\Phi_{\mathcal{S}'}(\boldsymbol{z}_i)]_j^2 \right| \right)^{\frac{1}{2}}$$

$$= \frac{B}{\sqrt{2}n} \sup_{\mathsf{K}_{\mathcal{S}'} \in \mathcal{K}_T} \left( \sum_{i=1}^n \sum_{j \in \mathbb{N}_+} [\Phi_{\mathcal{S}'}(\boldsymbol{z}_i)]_j^2 \right)^{\frac{1}{2}} \qquad \text{(rearrange the summations)}$$

$$= \frac{B}{\sqrt{2}n} \sup_{\mathsf{K}_{\mathcal{S}'} \in \mathcal{K}_T} \left( \sum_{i=1}^n \|\Phi_{\mathcal{S}'}(\boldsymbol{z}_i)\|^2 \right)^{\frac{1}{2}}$$

$$= \frac{B}{\sqrt{2}n} \sup_{\mathsf{K}_{\mathcal{S}'} \in \mathcal{K}_T} \left( \sum_{i=1}^n \mathsf{K}_{\mathcal{S}'}(\boldsymbol{z}_i, \boldsymbol{z}_i) \right)^{\frac{1}{2}}$$

$$= \frac{B}{\sqrt{2}n} \sup_{\mathsf{K}_{\mathcal{S}'} \in \mathcal{K}_T} \sqrt{\operatorname{Tr}(\mathsf{K}_{\mathcal{S}'}(\mathbf{Z}, \mathbf{Z}))}.$$

$\square$

## C.4 Proof of Corollary 1

**Corollary 1.** *Fix $B > 0$. Let $\hat{\mathcal{R}}_{\mathcal{S}}^{gf}(\mathcal{G}_T) = \min(U_1, U_2)$ where $U_1$ and $U_2$ are defined in Theorem 3. For any $\delta \in (0, 1)$, with probability at least $1 - \delta$ over the draw of an i.i.d. sample set $\mathcal{S} = \{\boldsymbol{z}_i\}_{i=1}^n$, if $\frac{1}{n^2} \sum_{i,j} \mathsf{K}_T(\boldsymbol{z}_i, \boldsymbol{z}_j; \mathcal{S}) \leq B^2$, the following holds for $\ell(\boldsymbol{w}_T, \boldsymbol{z})$ that trained from $\mathcal{S}$,*

$$L_\mu(A_T(\mathcal{S})) - L_S(A_T(\mathcal{S})) \leq 2\hat{\mathcal{R}}_{\mathcal{S}}^{gf}(\mathcal{G}_T) + 3\sqrt{\frac{\log(2/\delta)}{2n}},$$

*where $\boldsymbol{w}_T = A_T(\mathcal{S})$ is the output from the gradient flow* (1) *at time $T$ by using $\mathcal{S}$ as input.*

*Proof.* Apply Theorem 1 to $\mathcal{G}_T$ and $\mathcal{S}$, for all $g \in \mathcal{G}_T$

$$\mathbb{E}_z [g(\boldsymbol{z})] - \frac{1}{n} \sum_{i=1}^n g(\boldsymbol{z}_i) \le 2\hat{\mathcal{R}}_{\mathcal{S}}(\mathcal{G}_T) + 3\sqrt{\frac{\log(2/\delta)}{2n}}.$$

Since $g(\boldsymbol{z}) \in \mathcal{G}_T$ corresponds to $\ell(A_T(\mathcal{S}'), \boldsymbol{z})$ for all $\mathcal{S}' \in \mathsf{supp}(\mu^{\otimes n})$ that $\frac{1}{n^2} \sum_{i,j} \mathsf{K}_T(\boldsymbol{z}_i', \boldsymbol{z}_j'; \mathcal{S}') \le B^2$, it holds for all these feasible $\mathcal{S}' \in \mathsf{supp}(\mu^{\otimes n})$ that

$$
\begin{aligned}
&\mathbb{E}_z [\ell(A_T(\mathcal{S}'), \boldsymbol{z})] - \frac{1}{n} \sum_{i=1}^n \ell(A_T(\mathcal{S}'), \boldsymbol{z}_i) \\
&= L_\mu(A_T(\mathcal{S}')) - L_S(A_T(\mathcal{S}')) \qquad \text{(by definition of } L_S(\boldsymbol{w}) \text{ and } L_\mu(\boldsymbol{w})) \\
&\le 2\hat{\mathcal{R}}_{\mathcal{S}}(\mathcal{G}_T) + 3\sqrt{\frac{\log(2/\delta)}{2n}}.
\end{aligned}
$$

From Theorem 3, we have $\hat{\mathcal{R}}_{\mathcal{S}}(\mathcal{G}_T) \le \min(U_1, U_2)$. Define $\hat{\mathcal{R}}_{\mathcal{S}}^{gf}(\mathcal{G}_T) = \min(U_1, U_2)$ and take into above inequality,

$$
\begin{aligned}
L_\mu(A_T(\mathcal{S}')) - L_S(A_T(\mathcal{S}')) &\le 2\hat{\mathcal{R}}_{\mathcal{S}}(\mathcal{G}_T) + 3\sqrt{\frac{\log(2/\delta)}{2n}} \\
&\le 2\hat{\mathcal{R}}_{\mathcal{S}}^{gf}(\mathcal{G}_T) + 3\sqrt{\frac{\log(2/\delta)}{2n}}.
\end{aligned}
$$

Since above holds for all $\mathcal{S}' \in \mathsf{supp}(\mu^{\otimes n})$ that $\frac{1}{n^2} \sum_{i,j} \mathsf{K}_T(\boldsymbol{z}_i', \boldsymbol{z}_j'; \mathcal{S}') \le B^2$, and $\mathcal{S}$ is also in $\mathsf{supp}(\mu^{\otimes n})$, then if $\frac{1}{n^2} \sum_{i,j} \mathsf{K}_T(\boldsymbol{z}_i, \boldsymbol{z}_j; \mathcal{S}) \le B^2$, we have

$$L_\mu(A_T(\mathcal{S})) - L_S(A_T(\mathcal{S})) \le 2\hat{\mathcal{R}}_{\mathcal{S}}^{gf}(\mathcal{G}_T) + 3\sqrt{\frac{\log(2/\delta)}{2n}}.$$

$\square$

## D  Complete Proofs for Stochastic Gradient Flow

### D.1  Proof of Theorem 4

**Theorem 4.** *Suppose $\boldsymbol{w}(T) = \boldsymbol{w}_T$ is a solution of stochastic gradient flow at time $T \in \mathbb{N}$ with initialization $\boldsymbol{w}(0) = \boldsymbol{w}_0$. Then for any $\boldsymbol{z} \in \mathcal{Z}$,*

$$\ell(\boldsymbol{w}_T, \boldsymbol{z}) = \sum_{t=0}^{T-1} \sum_{i \in \mathcal{S}_t} -\frac{1}{m} \mathsf{K}_{t,t+1}(\boldsymbol{z}, \boldsymbol{z}_i; \mathcal{S}) + \ell(\boldsymbol{w}_0, \boldsymbol{z}),$$

*where $\mathsf{K}_{t,t+1}(\boldsymbol{z}, \boldsymbol{z}_i; \mathcal{S}) = \int_t^{t+1} \bar{\mathsf{K}}(\boldsymbol{w}(t); \boldsymbol{z}, \boldsymbol{z}_i) dt$ with $\bar{\mathsf{K}}$ defined in Definition 3.*

*Proof.* For each time interval $[t, t+1]$ and data batch $\mathcal{S}_t$, Stochastic gradient flow can be treated as full-batch gradient flow. Applying Theorem 2 for each $[t, t+1]$ and data batch $\mathcal{S}_t$, we have

$$\ell(\boldsymbol{w}_{t+1}, \boldsymbol{z}) = \sum_{i \in \mathcal{S}_t} -\frac{1}{m} \mathsf{K}_{t,t+1}(\boldsymbol{z}, \boldsymbol{z}_i; \mathcal{S}) + \ell(\boldsymbol{w}_t, \boldsymbol{z}), \tag{10}$$

where

$$\mathsf{K}_{t,t+1}(\boldsymbol{z}, \boldsymbol{z}_i; \mathcal{S}) = \int_t^{t+1} \bar{\mathsf{K}}(\boldsymbol{w}_t; \boldsymbol{z}, \boldsymbol{z}_i) dt.$$

For time $T \in \mathbb{N}$,

$$\ell(\boldsymbol{w}_T, \boldsymbol{z}) - \ell(\boldsymbol{w}_0, \boldsymbol{z}) = \sum_{t=0}^{T-1} \ell(\boldsymbol{w}_{t+1}, \boldsymbol{z}) - \ell(\boldsymbol{w}_t, \boldsymbol{z})$$

$$= \sum_{t=0}^{T-1} \sum_{i \in \mathcal{S}_t} -\frac{1}{m} \mathsf{K}_{t,t+1}(\boldsymbol{z}, \boldsymbol{z}_i; \mathcal{S}). \qquad \text{(takes in Eq. (10))}$$

$\square$

## D.2 Proof of Theorem 5

**Theorem 5.** *The Rademacher complexity of $\mathcal{G}_T$ defined in (3) has an upper bound:*

$$\hat{\mathcal{R}}_{\mathcal{S}}(\mathcal{G}_T) \leq \sum_{t=0}^{T-1} \frac{B_t}{n} \sqrt{\sup_{\mathsf{K}(\cdot, \cdot; \mathcal{S}') \in \mathcal{K}_T} \mathrm{Tr}(\mathsf{K}_{t,t+1}(\mathbf{Z}, \mathbf{Z}); \mathcal{S}') + \sum_{i \neq j} \Delta_t(\boldsymbol{z}_i, \boldsymbol{z}_j)}.$$

*where $\Delta_t(\boldsymbol{z}_i, \boldsymbol{z}_j) = \frac{1}{2} \left[ \sup_{\mathsf{K}(\cdot, \cdot; \mathcal{S}') \in \mathcal{K}_T} \mathsf{K}_{t,t+1}(\boldsymbol{z}_i, \boldsymbol{z}_j; \mathcal{S}') - \inf_{\mathsf{K}(\cdot, \cdot; \mathcal{S}') \in \mathcal{K}_T} \mathsf{K}_{t,t+1}(\boldsymbol{z}_i, \boldsymbol{z}_j; \mathcal{S}') \right]$.*

*Proof.* Recall,

$$\mathcal{G}_T = \{ g(\boldsymbol{z}) = \sum_{t=0}^{T-1} \sum_{i \in \mathcal{S}_t} -\frac{1}{m} \mathsf{K}_{t,t+1}(\boldsymbol{z}, \boldsymbol{z}_i'; \mathcal{S}') + \ell(\boldsymbol{w}_0, \boldsymbol{z}) : \mathsf{K}(\cdot, \cdot; \mathcal{S}') \in \mathcal{K}_T \}.$$

where

$$\mathcal{K}_T = \{ (\mathsf{K}_{0,1}(\cdot, \cdot; \mathcal{S}'), \cdots, \mathsf{K}_{T-1,T}(\cdot, \cdot; \mathcal{S}')) : \mathcal{S}' \in \mathsf{supp}(\mu^{\otimes n}), \frac{1}{m^2} \sum_{i,j \in \mathcal{S}_t} \mathsf{K}_{t,t+1}(\boldsymbol{z}_i', \boldsymbol{z}_j'; \mathcal{S}') \leq B_t^2 \}.$$

For $t = 0, 1, \cdots, T-1$, let

$$\mathcal{G}_t = \{ g(\boldsymbol{z}) = \sum_{i \in \mathcal{S}_t} -\frac{1}{m} \mathsf{K}_{t,t+1}(\boldsymbol{z}, \boldsymbol{z}_i'; \mathcal{S}') : \mathsf{K}(\cdot, \cdot; \mathcal{S}') \in \mathcal{K}_T \},$$

Then we have

$$\mathcal{G}_T \subseteq \mathcal{G}_0 \oplus \mathcal{G}_1 \oplus \cdots \oplus \mathcal{G}_{T-1} \oplus \{ \ell(\boldsymbol{w}_0, \boldsymbol{z}) \}.$$

Since the set on the RHS involves combinations of kernels induced from distinct training set $\mathcal{S}'$, it is a strictly larger set than the LHS. Apply Theorem 3 bound $U_1$ for each $\mathcal{G}_t$ on $\mathcal{S}$,

$$\hat{\mathcal{R}}_{\mathcal{S}}(\mathcal{G}_t) \leq \frac{B_t}{n} \left( \sup_{\mathsf{K}(\cdot, \cdot; \mathcal{S}') \in \mathcal{K}_T} \mathrm{Tr}(\mathsf{K}_{t,t+1}(\mathbf{Z}, \mathbf{Z}); \mathcal{S}') + \sum_{i \neq j} \Delta_t(\boldsymbol{z}_i, \boldsymbol{z}_j) \right)^{\frac{1}{2}}, \qquad (11)$$

where $\Delta_t(\boldsymbol{z}_i, \boldsymbol{z}_j) = \frac{1}{2} \left[ \sup_{\mathsf{K}(\cdot, \cdot; \mathcal{S}') \in \mathcal{K}_T} \mathsf{K}_{t,t+1}(\boldsymbol{z}_i, \boldsymbol{z}_j; \mathcal{S}') - \inf_{\mathsf{K}(\cdot, \cdot; \mathcal{S}') \in \mathcal{K}_T} \mathsf{K}_{t,t+1}(\boldsymbol{z}_i, \boldsymbol{z}_j; \mathcal{S}') \right]$.

By the monotonicity and linear combination of Rademacher complexity [38] and take in (11),

$$\hat{\mathcal{R}}_{\mathcal{S}}(\mathcal{G}_T) \leq \hat{\mathcal{R}}_{\mathcal{S}}(\mathcal{G}_0 \oplus \mathcal{G}_1 \oplus \cdots \oplus \mathcal{G}_{T-1} \oplus \{ \ell(\boldsymbol{w}_0, \boldsymbol{z}) \})$$

$$= \sum_{t=0}^{T-1} \hat{\mathcal{R}}_{\mathcal{S}}(\mathcal{G}_t) + \hat{\mathcal{R}}_{\mathcal{S}}(\{ \ell(\boldsymbol{w}_0, \boldsymbol{z}) \})$$

$$\leq \sum_{t=0}^{T-1} \frac{B_t}{n} \left( \sup_{\mathsf{K}(\cdot, \cdot; \mathcal{S}') \in \mathcal{K}_T} \mathrm{Tr}(\mathsf{K}_{t,t+1}(\mathbf{Z}, \mathbf{Z}); \mathcal{S}') + \sum_{i \neq j} \Delta_t(\boldsymbol{z}_i, \boldsymbol{z}_j) \right)^{\frac{1}{2}}.$$

$\square$

# E   Complete Proofs for Case Study

## E.1   Generalization bounds for Infinite-width NNs

### E.1.1   Proof of Eq. (4)

*Proof.* We bound $U_1$ in Theorem 3 for an infinite-width NN. For an infinite-width NN, the NTK keeps unchanged during training:

$$\hat{\Theta}(\boldsymbol{w}_t; \boldsymbol{x}, \boldsymbol{x}') \to \Theta(\boldsymbol{x}, \boldsymbol{x}') \cdot \mathbf{I}_k.$$

Then for our loss path kernel, for any $\boldsymbol{z}, \boldsymbol{z}' \in \mathcal{Z}$ and any $\mathsf{K}(\cdot, \cdot; \mathcal{S}') \in \mathcal{K}_T$,

$$
\begin{aligned}
\mathsf{K}(\boldsymbol{z}, \boldsymbol{z}'; \mathcal{S}') &= \int_0^T \bar{\mathsf{K}}(\boldsymbol{w}_t; \boldsymbol{z}, \boldsymbol{z}') dt \\
&= \int_0^T \nabla_f \ell(\boldsymbol{w}_t, \boldsymbol{z})^\top \hat{\Theta}(\boldsymbol{w}_t; \boldsymbol{x}, \boldsymbol{x}') \nabla_f \ell(\boldsymbol{w}_t, \boldsymbol{z}') \, dt \\
&= \int_0^T \nabla_f \ell(\boldsymbol{w}_t, \boldsymbol{z})^\top \Theta(\boldsymbol{x}, \boldsymbol{x}') \cdot \mathbf{I}_k \nabla_f \ell(\boldsymbol{w}_t, \boldsymbol{z}') \, dt \\
&= \Theta(\boldsymbol{x}, \boldsymbol{x}') \cdot \int_0^T \nabla_f \ell(\boldsymbol{w}_t, \boldsymbol{z})^\top \nabla_f \ell(\boldsymbol{w}_t, \boldsymbol{z}') \, dt.
\end{aligned}
\tag{12}
$$

Consider a $\rho$-Lipschitz loss function, i.e. $\|\nabla_f \ell(\boldsymbol{w}_t, \boldsymbol{z})\| \leq \rho$, e.g. $\rho = 1$ for hinge loss and logistic loss, $\rho = \sqrt{2}$ for cross-entropy loss with one-hot labels. Then we have

$$-\rho^2 \leq \nabla_f \ell(\boldsymbol{w}_t, \boldsymbol{z})^\top \nabla_f \ell(\boldsymbol{w}_t, \boldsymbol{z}') \leq \rho^2.$$

Thus

$$-\rho^2 T \leq \int_0^T \nabla_f \ell(\boldsymbol{w}_t, \boldsymbol{z})^\top \nabla_f \ell(\boldsymbol{w}_t, \boldsymbol{z}') \, dt \leq \rho^2 T.$$

Then by this inequality and Eq. (12),

$$-\rho^2 T \left|\Theta(\boldsymbol{x}, \boldsymbol{x}')\right| \leq \mathsf{K}(\boldsymbol{z}, \boldsymbol{z}'; \mathcal{S}') \leq \rho^2 T \left|\Theta(\boldsymbol{x}, \boldsymbol{x}')\right|.$$

Since $\Theta(\boldsymbol{x}_i, \boldsymbol{x}_i) \geq 0$ for $i \in [n]$, $\mathsf{K}(\boldsymbol{z}_i, \boldsymbol{z}_i; \mathcal{S}') \leq \rho^2 T \cdot \Theta(\boldsymbol{x}_i, \boldsymbol{x}_i)$ and $\mathrm{Tr}(\mathsf{K}(\mathbf{Z}, \mathbf{Z}); \mathcal{S}') \leq \mathrm{Tr}(\rho^2 T \cdot \Theta(\mathbf{X}, \mathbf{X}))$.

$$
\begin{aligned}
\Delta(\boldsymbol{z}_i, \boldsymbol{z}_j) &= \frac{1}{2} \left[ \sup_{\mathsf{K}(\cdot, \cdot; \mathcal{S}') \in \mathcal{K}_T} \mathsf{K}(\boldsymbol{z}_i, \boldsymbol{z}_j; \mathcal{S}') - \inf_{\mathsf{K}(\cdot, \cdot; \mathcal{S}') \in \mathcal{K}_T} \mathsf{K}(\boldsymbol{z}_i, \boldsymbol{z}_j; \mathcal{S}') \right] \\
&\leq \frac{1}{2} \left[ \rho^2 T \left|\Theta(\boldsymbol{x}_i, \boldsymbol{x}_j)\right| - \left(-\rho^2 T \left|\Theta(\boldsymbol{x}, \boldsymbol{x}_i)\right|\right) \right] \\
&= \rho^2 T \left|\Theta(\boldsymbol{x}_i, \boldsymbol{x}_j)\right|
\end{aligned}
$$

Take these terms into $U_1$,

$$
\begin{aligned}
\hat{\mathcal{R}}_\mathcal{S}(\mathcal{G}_T) &\leq \frac{B}{n} \sqrt{\sup_{\mathsf{K}(\cdot, \cdot; \mathcal{S}') \in \mathcal{K}_T} \mathrm{Tr}(\mathsf{K}(\mathbf{Z}, \mathbf{Z}; \mathcal{S}')) + \sum_{i \neq j} \Delta(\boldsymbol{z}_i, \boldsymbol{z}_j)} \\
&\leq \frac{B}{n} \sqrt{\mathrm{Tr}(\rho^2 T \cdot \Theta(\mathbf{X}, \mathbf{X})) + \sum_{i \neq j} \rho^2 T \left|\Theta(\boldsymbol{x}_i, \boldsymbol{x}_j)\right|} \\
&= \frac{\rho B \sqrt{T}}{n} \sqrt{\sum_{i,j} \left|\Theta(\boldsymbol{x}_i, \boldsymbol{x}_j)\right|}.
\end{aligned}
\tag{13}
$$

$\square$

## E.2 Generalization bound for stable algorithms

Let $\mathcal{S}'$ and $S_1'$ be two datasets that only differ in one data point. We make the following stability assumption for gradient decent.

**Assumption 1** (uniform stability of GD.)**.** Assume $\|A_t(\mathcal{S}') - A_t(S_1')\| = \frac{ct}{n}$ for some constant $c > 0$. Assume $\ell(\boldsymbol{w}, \boldsymbol{z})$ is $L_\ell$-Lipschitz and $\beta_\ell$-smooth for any $\boldsymbol{z} \in \mathcal{Z}$. Then $\|\nabla_{\boldsymbol{w}}\ell(A_t(\mathcal{S}'), \boldsymbol{z}) - \nabla_{\boldsymbol{w}}\ell(A_t(S_1'), \boldsymbol{z})\| \leq \beta_\ell \|A_t(\mathcal{S}') - A_t(S_1')\| = c\beta_\ell \frac{t}{n}$.

This kind of stability results of GD and SGD are proved in [8, 25]. Under these stability and smoothness assumptions, we can bound the deviation of $g(\boldsymbol{z}_i)$ from its expectation with high probability and further bound the complexity based on Theorem 3.

**Theorem 8.** *For any $\delta \in (0, 1)$, let $\mathcal{G}_T^\delta \subset \mathcal{G}_T$ be a $1 - \delta$ subset of $\mathcal{G}_T$, i.e. $\left|\mathcal{G}_T^\delta\right| = (1 - \delta)\left|\mathcal{G}_T\right|$. Under Assumption 1, we have at least one of such $\mathcal{G}_T^\delta$,*

$$\hat{\mathcal{R}}_\mathcal{S}(\mathcal{G}_T^\delta) \leq \left(2L_\ell^2 T + cL_\ell\beta_\ell T^2\right)\sqrt{\frac{2\log(\frac{2n}{\delta})}{n}}.$$

This bound will naturally translate into a generalization bound by equipping with Theorem 1. This bound has a convergence rate of $\tilde{O}(1/\sqrt{n})$. It shows that the complexity of NN trained by GD has a polynomial dependence on $L_\ell$, $\beta_\ell$, and training time $T$.

*Proof.* Let $\mathcal{G}_T^{\delta, \mathcal{S}} = \left\{g(\mathbf{Z}) = (g(\boldsymbol{z}_1), \ldots, g(\boldsymbol{z}_n)) : g \in \mathcal{G}_T^\delta\right\}$. Then similar bound as $U_2$ in Theorem 3 holds for $\mathcal{G}_T^\delta$,

$$\hat{\mathcal{R}}_\mathcal{S}(\mathcal{G}_T^\delta) \leq \inf_{\epsilon > 0}\left(\frac{\epsilon}{n} + \sqrt{\frac{2\ln\mathcal{N}(\mathcal{G}_T^{\delta,\mathcal{S}}, \epsilon, \|\|_1)}{n}}\right) \tag{14}$$

We consider the covering of $\mathcal{G}_T^\delta$ and upper bound the right hand side. Without loss of generality, suppose $\mathcal{S}'$ and $S_1'$ differ in the first sample. That is $\mathcal{S}' = \{\boldsymbol{z}_1', \ldots, \boldsymbol{z}_n'\}$ and $S_1' = \{\hat{\boldsymbol{z}}_1', \ldots, \boldsymbol{z}_n'\}$. For any fixed $i \in [n]$ and $g_{\mathsf{K}_{S_1'}}, g_{\mathsf{K}_{\mathcal{S}'}} \in \mathcal{G}_T$,

$$g_{\mathsf{K}_{S_1'}}(\boldsymbol{z}_i) - g_{\mathsf{K}_{\mathcal{S}'}}(\boldsymbol{z}_i)$$
$$= \frac{1}{n}\mathsf{K}_{\mathcal{S}'}(\boldsymbol{z}_i, \boldsymbol{z}_1') - \frac{1}{n}\mathsf{K}_{S_1'}(\boldsymbol{z}_i, \hat{\boldsymbol{z}}_1') + \sum_{j=2}^n \frac{1}{n}\mathsf{K}_{\mathcal{S}'}(\boldsymbol{z}_i, \boldsymbol{z}_j') - \sum_{j=2}^n \frac{1}{n}\mathsf{K}_{S_1'}(\boldsymbol{z}_i, \boldsymbol{z}_j')$$

The first two terms are

$$\frac{1}{n}\mathsf{K}_{\mathcal{S}'}(\boldsymbol{z}_i, \boldsymbol{z}_1') - \frac{1}{n}\mathsf{K}_{S_1'}(\boldsymbol{z}_i, \hat{\boldsymbol{z}}_1')$$
$$= \frac{1}{n}\int_0^T \langle\nabla_{\boldsymbol{w}}\ell(A_t(\mathcal{S}'), \boldsymbol{z}_i), \nabla_{\boldsymbol{w}}\ell(A_t(\mathcal{S}'), \boldsymbol{z}_1')\rangle\,dt - \frac{1}{n}\int_0^T \langle\nabla_{\boldsymbol{w}}\ell(A_t(S_1'), \boldsymbol{z}_i), \nabla_{\boldsymbol{w}}\ell(A_t(S_1'), \hat{\boldsymbol{z}}_1')\rangle\,dt$$
$$= \frac{1}{n}\int_0^T \langle\nabla_{\boldsymbol{w}}\ell(A_t(\mathcal{S}'), \boldsymbol{z}_i), \nabla_{\boldsymbol{w}}\ell(A_t(\mathcal{S}'), \boldsymbol{z}_1')\rangle - \langle\nabla_{\boldsymbol{w}}\ell(A_t(S_1'), \boldsymbol{z}_i), \nabla_{\boldsymbol{w}}\ell(A_t(S_1'), \hat{\boldsymbol{z}}_1')\rangle\,dt$$
$$\leq \frac{1}{n}\int_0^T \|\nabla_{\boldsymbol{w}}\ell(A_t(\mathcal{S}'), \boldsymbol{z}_i)\|\,\|\nabla_{\boldsymbol{w}}\ell(A_t(\mathcal{S}'), \boldsymbol{z}_1')\| + \|\nabla_{\boldsymbol{w}}\ell(A_t(S_1'), \boldsymbol{z}_i)\|\,\|\nabla_{\boldsymbol{w}}\ell(A_t(S_1'), \hat{\boldsymbol{z}}_1')\|\,dt$$
$$\leq \frac{1}{n}\int_0^T L_\ell^2 + L_\ell^2\,dt$$
$$= \frac{1}{n}2L_\ell^2 T$$

The last two terms are

$$\sum_{j=2}^{n} \frac{1}{n} \mathsf{K}_{\mathcal{S}'}(\boldsymbol{z}_i, \boldsymbol{z}_j') - \sum_{j=2}^{n} \frac{1}{n} \mathsf{K}_{S_1'}(\boldsymbol{z}_i, \boldsymbol{z}_j')$$

$$= \frac{1}{n} \sum_{j=2}^{n} \mathsf{K}_{\mathcal{S}'}(\boldsymbol{z}_i, \boldsymbol{z}_j') - \mathsf{K}_{S_1'}(\boldsymbol{z}_i, \boldsymbol{z}_j')$$

$$= \frac{1}{n} \sum_{j=2}^{n} \int_0^T \left\langle \nabla_{\boldsymbol{w}} \ell(A_t(\mathcal{S}'), \boldsymbol{z}_i), \nabla_{\boldsymbol{w}} \ell(A_t(\mathcal{S}'), \boldsymbol{z}_j') \right\rangle dt - \int_0^T \left\langle \nabla_{\boldsymbol{w}} \ell(A_t(S_1'), \boldsymbol{z}_i), \nabla_{\boldsymbol{w}} \ell(A_t(S_1'), \boldsymbol{z}_j') \right\rangle dt$$

$$= \frac{1}{n} \sum_{j=2}^{n} \int_0^T \left\langle \nabla_{\boldsymbol{w}} \ell(A_t(\mathcal{S}'), \boldsymbol{z}_i), \nabla_{\boldsymbol{w}} \ell(A_t(\mathcal{S}'), \boldsymbol{z}_j') \right\rangle - \left\langle \nabla_{\boldsymbol{w}} \ell(A_t(S_1'), \boldsymbol{z}_i), \nabla_{\boldsymbol{w}} \ell(A_t(S_1'), \boldsymbol{z}_j') \right\rangle dt$$

$$= \frac{1}{n} \sum_{j=2}^{n} \int_0^T \left\langle \nabla_{\boldsymbol{w}} \ell(A_t(\mathcal{S}'), \boldsymbol{z}_i), \nabla_{\boldsymbol{w}} \ell(A_t(\mathcal{S}'), \boldsymbol{z}_j') \right\rangle - \left\langle \nabla_{\boldsymbol{w}} \ell(A_t(\mathcal{S}'), \boldsymbol{z}_i), \nabla_{\boldsymbol{w}} \ell(A_t(S_1'), \boldsymbol{z}_j') \right\rangle$$

$$\qquad + \left\langle \nabla_{\boldsymbol{w}} \ell(A_t(\mathcal{S}'), \boldsymbol{z}_i), \nabla_{\boldsymbol{w}} \ell(A_t(S_1'), \boldsymbol{z}_j') \right\rangle - \left\langle \nabla_{\boldsymbol{w}} \ell(A_t(S_1'), \boldsymbol{z}_i), \nabla_{\boldsymbol{w}} \ell(A_t(S_1'), \boldsymbol{z}_j') \right\rangle dt$$

$$= \frac{1}{n} \sum_{j=2}^{n} \int_0^T \left\langle \nabla_{\boldsymbol{w}} \ell(A_t(\mathcal{S}'), \boldsymbol{z}_i), \nabla_{\boldsymbol{w}} \ell(A_t(\mathcal{S}'), \boldsymbol{z}_j') - \nabla_{\boldsymbol{w}} \ell(A_t(S_1'), \boldsymbol{z}_j') \right\rangle$$

$$\qquad + \left\langle \nabla_{\boldsymbol{w}} \ell(A_t(\mathcal{S}'), \boldsymbol{z}_i) - \nabla_{\boldsymbol{w}} \ell(A_t(S_1'), \boldsymbol{z}_i), \nabla_{\boldsymbol{w}} \ell(A_t(S_1'), \boldsymbol{z}_j') \right\rangle dt$$

$$\leq \frac{1}{n} \sum_{j=2}^{n} \int_0^T \left\| \nabla_{\boldsymbol{w}} \ell(A_t(\mathcal{S}'), \boldsymbol{z}_i) \right\| \left\| \nabla_{\boldsymbol{w}} \ell(A_t(\mathcal{S}'), \boldsymbol{z}_j') - \nabla_{\boldsymbol{w}} \ell(A_t(S_1'), \boldsymbol{z}_j') \right\|$$

$$\qquad + \left\| \nabla_{\boldsymbol{w}} \ell(A_t(\mathcal{S}'), \boldsymbol{z}_i) - \nabla_{\boldsymbol{w}} \ell(A_t(S_1'), \boldsymbol{z}_i) \right\| \left\| \nabla_{\boldsymbol{w}} \ell(A_t(S_1'), \boldsymbol{z}_j') \right\| dt$$

$$\leq \frac{1}{n} \sum_{j=2}^{n} \int_0^T L_\ell \beta_\ell \frac{ct}{n} + \beta_\ell \frac{ct}{n} L_\ell dt \qquad \qquad \text{(by Assumption 1)}$$

$$= \left( 1 - \frac{1}{n} \right) c L_\ell \beta_\ell \frac{T^2}{n}$$

In total,

$$g_{\mathsf{K}_{S_1'}}(\boldsymbol{z}_i) - g_{\mathsf{K}_{\mathcal{S}'}}(\boldsymbol{z}_i)$$

$$\leq \frac{1}{n} 2 L_\ell^2 T + \left( 1 - \frac{1}{n} \right) c L_\ell \beta_\ell \frac{T^2}{n}$$

$$= \frac{1}{n} \left( 2 L_\ell^2 T + \left( 1 - \frac{1}{n} \right) c L_\ell \beta_\ell T^2 \right)$$

$$\leq \frac{1}{n} \left( 2 L_\ell^2 T + c L_\ell \beta_\ell T^2 \right)$$

Similarly $g_{\mathsf{K}_{\mathcal{S}'}}(\boldsymbol{z}_i) - g_{\mathsf{K}_{S_1'}}(\boldsymbol{z}_i) \leq \frac{1}{n} \left( 2 L_\ell^2 T + c L_\ell \beta_\ell T^2 \right)$. Thus $\left| g_{\mathsf{K}_{\mathcal{S}'}}(\boldsymbol{z}_i) - g_{\mathsf{K}_{S_1'}}(\boldsymbol{z}_i) \right| \leq \frac{1}{n} \left( 2 L_\ell^2 T + c L_\ell \beta_\ell T^2 \right)$. Then by McDiarmid's inequality, for any $\delta \in (0, 1)$, with probability at least $1 - \delta$,

$$\left| g_{\mathsf{K}_{\mathcal{S}'}}(\boldsymbol{z}_i) - \mathop{\mathbb{E}}_{\mathcal{S}'} \left[ g_{\mathsf{K}_{\mathcal{S}'}}(\boldsymbol{z}_i) \right] \right| \leq \left( 2 L_\ell^2 T + c L_\ell \beta_\ell T^2 \right) \sqrt{\frac{\log(\frac{2}{\delta})}{2n}}$$

Then, by a union bound, with probability at least $1 - \delta$, for all $i \in [n]$,

$$\left| g_{\mathsf{K}_{\mathcal{S}'}}(\boldsymbol{z}_i) - \mathop{\mathbb{E}}_{\mathcal{S}'} \left[ g_{\mathsf{K}_{\mathcal{S}'}}(\boldsymbol{z}_i) \right] \right| \leq \left( 2 L_\ell^2 T + c L_\ell \beta_\ell T^2 \right) \sqrt{\frac{\log(\frac{2n}{\delta})}{2n}}$$

This means that with probability at least $1 - \delta$, for all $i \in [n]$,

$$g_{\mathsf{K}_{\mathcal{S}'}}(\boldsymbol{z}_i) \in \left[ \mathop{\mathbb{E}}_{\mathcal{S}'} \left[ g_{\mathsf{K}_{\mathcal{S}'}}(\boldsymbol{z}_i) \right] - \left( 2 L_\ell^2 T + c L_\ell \beta_\ell T^2 \right) \sqrt{\frac{\log(\frac{2n}{\delta})}{2n}}, \mathop{\mathbb{E}}_{\mathcal{S}'} \left[ g_{\mathsf{K}_{\mathcal{S}'}}(\boldsymbol{z}_i) \right] + \left( 2 L_\ell^2 T + c L_\ell \beta_\ell T^2 \right) \sqrt{\frac{\log(\frac{2n}{\delta})}{2n}} \right]$$

Use the $g_{\mathsf{K}_{\mathcal{S}'}}$ in this range to construct the $\mathcal{G}_T^\delta$. Then for any $g_1(\mathbf{Z}), g_2(\mathbf{Z}) \in \mathcal{G}_T^{\delta,\mathcal{S}}$,

$$g_1(\mathbf{Z}) - g_2(\mathbf{Z}) = \sum_{i=1}^n |g_1(\boldsymbol{z}_i) - g_2(\boldsymbol{z}_i)|$$

$$\leq \sum_{i=1}^n 2\left(2L_\ell^2 T + cL_\ell \beta_\ell T^2\right) \sqrt{\frac{\log(\frac{2n}{\delta})}{2n}}$$

$$= \left(2L_\ell^2 T + cL_\ell \beta_\ell T^2\right) \sqrt{2n \log(\frac{2n}{\delta})}.$$

Take $\epsilon$ as this value, then $\mathcal{N}(\mathcal{G}_T^{\delta,\mathcal{S}}, \epsilon, \|\|_1) = 1$. Take this into Eq. (14),

$$\hat{\mathcal{R}}_{\mathcal{S}}(\mathcal{G}_T^\delta) \leq \inf_{\epsilon > 0} \left( \frac{\epsilon}{n} + \sqrt{\frac{2 \ln \mathcal{N}(\mathcal{G}_T^{\delta,\mathcal{S}}, \epsilon, \|\|_1)}{n}} \right)$$

$$\leq \left(2L_\ell^2 T + cL_\ell \beta_\ell T^2\right) \sqrt{\frac{2\log(\frac{2n}{\delta})}{n}}.$$

$\square$

## E.3 Norm-constrained neural network

For simplicity, we consider the one-dimensional output in this subsection, i.e. $k = 1$, and assume the loss function is $\rho$-lipschitz for the model output, $\|\ell(\hat{\boldsymbol{y}}, \boldsymbol{y}) - \ell(\hat{\boldsymbol{y}}', \boldsymbol{y})\| \leq \rho(\hat{\boldsymbol{y}} - \hat{\boldsymbol{y}}')$ for every $\hat{\boldsymbol{y}}, \hat{\boldsymbol{y}}' \in \mathbb{R}$. Consider one-layer NNs:

$$\mathcal{F} = \left\{ f(\boldsymbol{w}, \boldsymbol{x}) = \boldsymbol{w}^T \boldsymbol{x} : \boldsymbol{w} \in \mathbb{R}^d, \boldsymbol{x} \in \mathbb{R}^d \right\}.$$

**Proposition 2.** *For the function class of one-layer NN defined above,*

$$\sup_{\mathsf{K}_{\mathcal{S}'} \in \mathcal{K}_T} \mathrm{Tr}(\mathsf{K}_{\mathcal{S}'}(\mathbf{Z}, \mathbf{Z})) \leq \rho^2 T \sum_{i=1}^n \|\boldsymbol{x}_i\|^2.$$

For this one-layer NN, we do not need a norm constraint. For $L$-layer NNs with norm constraints:

$$\mathcal{F} = \left\{ f(\boldsymbol{w}, \boldsymbol{x}) = W^L \sigma(W^{L-1} \cdots \sigma(W^1 \boldsymbol{x})) : \left\| W_t^h \right\| \leq B_i, t \in [0, T] \right\}$$

where $W^h \in \mathbb{R}^{d_h \times d_{h-1}}$ for $h \in [L]$ with $d_L = 1, d_0 = d$. $\sigma$ is the element-wise activation function and is 1-lipschitz with $\sigma(0) = 0$. With these norm constraints of the parameters during the training, we can further bound the trace term in $U_1$ as follows.

**Theorem 9.** *For the function class of $L$-layer NN defined above,*

$$\sup_{\mathsf{K}_{\mathcal{S}'} \in \mathcal{K}_T} \mathrm{Tr}(\mathsf{K}_{\mathcal{S}'}(\mathbf{Z}, \mathbf{Z})) \leq \rho^2 T \sum_{i=1}^n \|\boldsymbol{x}_i\|^2 \prod_{j=1}^L B_j^2 \sum_{h=1}^L \frac{1}{B_h^2}.$$

This bound shows that this quantity has a linear relation with $T$. With a finer constraint of $\left\| W_t^i \right\|$ during the training, for example, $\left\| W_t^i \right\| \leq B_{i,t'}, t \in [t', t'+1]$, we can get tighter bound. Although similar to previous norm-based bounds that have a polynomial dependence with the norms of the parameters, our bound has a clear dependence on training time $T$, which is not achievable from previous approaches. But note these bounds can be very loose since they are worse-case bounds.

### E.3.1 Proof of Proposition 2

*Proof.*

$$\mathsf{K}_{\mathcal{S}'}(\boldsymbol{z}, \boldsymbol{z}') = \int_0^T \langle \ell'(f_t(\boldsymbol{x}), \boldsymbol{y}) \nabla_{\boldsymbol{w}} f_t(\boldsymbol{x}), \ell'(f_t(\boldsymbol{x}'), \boldsymbol{y}') \nabla_{\boldsymbol{w}} f_t(\boldsymbol{x}') \rangle \, dt$$

$$= \int_0^T \langle \ell'(f_t(\boldsymbol{x}), \boldsymbol{y}) \boldsymbol{x}, \ell'(f_t(\boldsymbol{x}'), \boldsymbol{y}') \boldsymbol{x}' \rangle \, dt$$

$$= \int_0^T \ell'(f_t(\boldsymbol{x}), \boldsymbol{y}) \ell'(f_t(\boldsymbol{x}'), \boldsymbol{y}') \langle \boldsymbol{x}, \boldsymbol{x}' \rangle \, dt$$

$$\leq \rho^2 \left| \langle \boldsymbol{x}, \boldsymbol{x}' \rangle \right| T.$$

Thus

$$\text{Tr}(\mathsf{K}_{\mathcal{S}'}(\mathbf{Z}, \mathbf{Z})) = \sum_{i=1}^n \mathsf{K}_T(\boldsymbol{z}_i, \boldsymbol{z}_i) \leq \sum_{i=1}^n \rho^2 \|\boldsymbol{x}_i\|^2 T = \rho^2 T \sum_{i=1}^n \|\boldsymbol{x}_i\|^2.$$

Since this holds for any $\mathsf{K}_{\mathcal{S}'} \in \mathcal{K}_T$,

$$\sup_{\mathsf{K}_{\mathcal{S}'} \in \mathcal{K}_T} \text{Tr}(\mathsf{K}_{\mathcal{S}'}(\mathbf{Z}, \mathbf{Z})) \leq \rho^2 T \sum_{i=1}^n \|\boldsymbol{x}_i\|^2.$$

$\square$

### E.3.2 Proof of Theorem 9

*Proof.* Denote

$$f^h(\boldsymbol{x}) = W^h g^{h-1}(\boldsymbol{x}) \in \mathbb{R}^{d_h}, \quad g^{h-1}(\boldsymbol{x}) = \sigma(f^{h-1}(\boldsymbol{x})) \in \mathbb{R}^{d_{h-1}}, \quad h \in [L]$$

$$\frac{\partial f(\boldsymbol{w}, \boldsymbol{x})}{\partial W^h} = b^h(\boldsymbol{x}) \left( g^{h-1}(\boldsymbol{x}) \right)^T \in \mathbb{R}^{d_h \times d_{h-1}}, \quad h \in [L]$$

where

$$b^h(\boldsymbol{x}) = \begin{cases} 1 \in \mathbb{R}, & h = L, \\ D^h(\boldsymbol{x}) \left( W^{h+1} \right)^T b^{h+1}(\boldsymbol{x}) \in \mathbb{R}^{d_h} & h \in [L-1], \end{cases}$$

$$D^h(\boldsymbol{x}) = diag(\dot{\sigma}(f^h(\boldsymbol{x}))) \in \mathbb{R}^{d_h \times d_h}, \quad h \in [L-1].$$

Then for any $h \in [L]$, we can compute

$$\left\langle \frac{\partial f(\boldsymbol{w}, \boldsymbol{x})}{\partial W^h}, \frac{\partial f(\boldsymbol{w}, \boldsymbol{x}')}{\partial W^h} \right\rangle$$

$$= \left\langle b^h(\boldsymbol{x}) \left( g^{h-1}(\boldsymbol{x}) \right)^T, b^h(\boldsymbol{x}') \left( g^{h-1}(\boldsymbol{x}') \right)^T \right\rangle \qquad \text{(inner product of matrices)}$$

$$= \text{Tr} \left( g^{h-1}(\boldsymbol{x}) b^h(\boldsymbol{x})^T b^h(\boldsymbol{x}') \left( g^{h-1}(\boldsymbol{x}') \right)^T \right)$$

$$= \text{Tr} \left( \left( g^{h-1}(\boldsymbol{x}') \right)^T g^{h-1}(\boldsymbol{x}) b^h(\boldsymbol{x})^T b^h(\boldsymbol{x}') \right)$$

$$= \left\langle g^{h-1}(\boldsymbol{x}), g^{h-1}(\boldsymbol{x}') \right\rangle \cdot \left\langle b^h(\boldsymbol{x}), b^h(\boldsymbol{x}') \right\rangle,$$

where we have for the first term,

$$\left\langle g^{h-1}(\boldsymbol{x}), g^{h-1}(\boldsymbol{x}') \right\rangle \leq \left\| g^{h-1}(\boldsymbol{x}) \right\| \left\| g^{h-1}(\boldsymbol{x}') \right\|$$

$$\leq \prod_{j=1}^{h-1} \left\| W^j \right\|^2 \|\boldsymbol{x}\| \|\boldsymbol{x}'\|.$$

The second term can be bounded as

$$
\begin{aligned}
\langle b^h(\boldsymbol{x}), b^h(\boldsymbol{x}') \rangle &\leq \left\| b^h(\boldsymbol{x}) \right\| \left\| b^h(\boldsymbol{x}') \right\| \\
&= \left\| D^h(\boldsymbol{x}) \left( W^{h+1} \right)^T b^{h+1}(\boldsymbol{x}) \right\| \left\| D^h(\boldsymbol{x}') \left( W^{h+1} \right)^T b^{h+1}(\boldsymbol{x}') \right\| \\
&\leq \left\| D^h(\boldsymbol{x}) \right\| \left\| W^{h+1} \right\| \left\| b^{h+1}(\boldsymbol{x}) \right\| \left\| D^h(\boldsymbol{x}') \right\| \left\| W^{h+1} \right\| \left\| b^{h+1}(\boldsymbol{x}') \right\| \\
&\leq \left\| W^{h+1} \right\|^2 \left\| b^{h+1}(\boldsymbol{x}) \right\| \left\| b^{h+1}(\boldsymbol{x}') \right\| \\
&\leq \prod_{j=h+1}^{L} \left\| W^j \right\|^2 .
\end{aligned}
$$

Thus in total,

$$
\begin{aligned}
\left\langle \frac{\partial f(\boldsymbol{w}, \boldsymbol{x})}{\partial W^h}, \frac{\partial f(\boldsymbol{w}, \boldsymbol{x}')}{\partial W^h} \right\rangle &= \left\langle g^{h-1}(\boldsymbol{x}), g^{h-1}(\boldsymbol{x}') \right\rangle \cdot \left\langle b^h(\boldsymbol{x}), b^h(\boldsymbol{x}') \right\rangle \\
&\leq \prod_{j=1}^{h-1} \left\| W^j \right\|^2 \left\| \boldsymbol{x} \right\| \left\| \boldsymbol{x}' \right\| \cdot \prod_{j=h+1}^{L} \left\| W^j \right\|^2 \\
&= \left\| \boldsymbol{x} \right\| \left\| \boldsymbol{x}' \right\| \frac{\prod_{j=1}^{L} \left\| W^j \right\|^2}{\left\| W^h \right\|^2}
\end{aligned}
$$

Since the tangent kernel $\left\langle \nabla_{\boldsymbol{w}} f(\boldsymbol{w}, \boldsymbol{x}), \nabla_{\boldsymbol{w}} f(\boldsymbol{w}, \boldsymbol{x}') \right\rangle = \sum_{h=1}^{L} \left\langle \frac{\partial f(\boldsymbol{w}, \boldsymbol{x})}{\partial W^h}, \frac{\partial f(\boldsymbol{w}, \boldsymbol{x}')}{\partial W^h} \right\rangle$, we obtain an upper bound for the tangent kernel,

$$
\left\langle \nabla_{\boldsymbol{w}} f(\boldsymbol{w}, \boldsymbol{x}), \nabla_{\boldsymbol{w}} f(\boldsymbol{w}, \boldsymbol{x}') \right\rangle \leq \left\| \boldsymbol{x} \right\| \left\| \boldsymbol{x}' \right\| \prod_{j=1}^{L} \left\| W^j \right\|^2 \sum_{h=1}^{L} \frac{1}{\left\| W^h \right\|^2}.
$$

Thus

$$
\begin{aligned}
\mathsf{K}_{\mathcal{S}'}(\boldsymbol{z}, \boldsymbol{z}') &= \int_0^T l'(f(\boldsymbol{w}_t, \boldsymbol{x}), \boldsymbol{y}) l'(f(\boldsymbol{w}_t, \boldsymbol{x}'), \boldsymbol{y}') \left\langle \nabla_{\boldsymbol{w}} f(\boldsymbol{w}_t, \boldsymbol{x}), \nabla_{\boldsymbol{w}} f(\boldsymbol{w}_t, \boldsymbol{x}') \right\rangle dt \\
&\leq \int_0^T \rho^2 \left\| \boldsymbol{x} \right\| \left\| \boldsymbol{x}' \right\| \prod_{j=1}^{L} \left\| W^j \right\|^2 \sum_{h=1}^{L} \frac{1}{\left\| W^h \right\|^2} dt \\
&\leq \int_0^T \rho^2 \left\| \boldsymbol{x} \right\| \left\| \boldsymbol{x}' \right\| \prod_{j=1}^{L} B_j^2 \sum_{h=1}^{L} \frac{1}{B_h^2} dt \\
&= \rho^2 T \left\| \boldsymbol{x} \right\| \left\| \boldsymbol{x}' \right\| \prod_{j=1}^{L} B_j^2 \sum_{h=1}^{L} \frac{1}{B_h^2}.
\end{aligned}
$$

Thus,

$$
\mathrm{Tr}(\mathsf{K}_{\mathcal{S}'}(\mathbf{Z}, \mathbf{Z})) \leq \rho^2 T \sum_{i=1}^{n} \left\| \boldsymbol{x}_i \right\|^2 \prod_{j=1}^{L} B_j^2 \sum_{h=1}^{L} \frac{1}{B_h^2}.
$$

Since this holds for any $\mathsf{K}_{\mathcal{S}'} \in \mathcal{K}_T$,

$$
\sup_{\mathsf{K}_{\mathcal{S}'} \in \mathcal{K}_T} \mathrm{Tr}(\mathsf{K}_{\mathcal{S}'}(\mathbf{Z}, \mathbf{Z})) \leq \rho^2 T \sum_{i=1}^{n} \left\| \boldsymbol{x}_i \right\|^2 \prod_{j=1}^{L} B_j^2 \sum_{h=1}^{L} \frac{1}{B_h^2}.
$$

$\square$

