# OpenReview forum: "Analyzing Generalization of Neural Networks through Loss Path Kernels"
_NeurIPS.cc/2023/Conference — NeurIPS 2023 poster_

### Official Review · Reviewer_kz2v · 2023-06-26

**Soundness:** 2 fair
**Presentation:** 2 fair
**Contribution:** 3 good
**Rating:** 5
**Confidence:** 4

**Summary:**

Thanks to authors for their submission. Their paper tackles generalization of neural network from neural tangent perspective applicable on gradient flow. The novelty of the paper stems from the new tangent kernel $\overline{K}(w,z,z')$ proposed. It is defined as an inner product of loss gradients w.r.t. weights and evaluated at two data points $z$ and $z'$. This leads to Loss Path Kernel defined as a path integral w.r.t. $t$ of $\overline{K}(w(t),z,z')$. After introduction and recollection of NTK theory the definitions are laid out. The paper continues with proving the gradient flow loss is general kernel machine followed up by Rademacher complexity bound derivation. The later section 5 extends these results to SGD. Finally results are used to design neural architecture search (NAS) and demonstrate favorable performance compared with state-of-the-art NAS algorithms through numerical experiments.

Paper follows an interesting idea of generalising tangent kernels to path integrals. It is timely and potentially interesting contribution. Being theoretical work, however, it lacks the clarity in presentation, rigor in definitions and theorems and this strongly undermines its validity. The amendments are recommended int he bellow.

**Strengths:**

+ Well introduced paper with well stated contributions in Table1 presenting paper's contributions to NTK theory.

+ Paper strives for applicability and results are used to design of neural architecture search (NAS) supported by (limited) numerical experiments.

**Weaknesses:**

- $\textbf{[Definition of the Loss Path Kernel]}$ For instance the line 108:The Theorem 2 "... shows that the loss of the NN at a certain fixed time ($T$) is a general kernel machine ...". But is the limit (when $T \rightarrow \infty$ and/or $dt \rightarrow 0$) also in RKHS? Under what conditions? The existence/convergence of the integral is not commented neither in the body nor in the appendices. I would urge authors to detail this out very thoroughly as assumptions needed for a definition to work would possibly limit the applicability of the results. For instance the Lipschitz gradients may be needed, because loss may be Lipschitz locally, but its norm may be growing with $t$, e.g., in case of overfitting, and the integral (LPK) may still diverge as $T \rightarrow \infty$.

- I try to provide alternative view: In case of the gradient flow over whole training data set considered by paper, i.e., gradient descent with infinitesimal steps, as opposed to stochastic (mini-batch) gradient descent, the weight updates are locally diffeomorphisms (locally invertible smooth maps - that is what paper implicitly assumes by infinitesimal steps). Then the kernel (LTK) $\overline{K}(z,z')$ (Definition 3) can be seen as a pull-back inner product used for instance in [Principles of riemannian geometry in neural networks, Hauser, Michael and Ray, Asok, Advances in neural information processing systems, vol=30,2017] and the crucial Loss Path Kernel (LPK) $K_T(z,z')$ (Definition 4) is defined as a length of the parameterised curve $f \odot w(t)$ w.r.t. this inner product $\textbf{if it exists}$! It does not in general unless inner products along the path "behave nicely".

- Following up on the above, assumptions required by Corollary 1 should be listed thoroughly. Line 203 reads  ..."assuming $\ell({w},z)$ is $L_{\ell}^2$-Lipschitz ... ". Is it needed? If so, is it enough for Definitions to be well defined (see above).

- $\textbf{[Corollary 1 (Generalization bound for NN)]}$ the lines 203-208 are very vague, using terms "usually, often etc." which degrade validity of the Corollary 1 and thus the main result of the paper. I strongly advise authors to avoid such terms. Especially in connection to the most important result of the paper. Try to clarify what is assumed, what is already existing result (please cite sources, if so) and what is hypothesis. For instance: l: 203-205: "... $\textbf{assuming}$ the $\ell(w, z)$ is $L_{\ell}$-Lipschitz. The second component is $\textbf{often}$ not the dominant term since GD is a stable algorithm when the training data are modified slightly [24, 8]. It $\textbf{usually}$ decreases with $n$ since GD becomes more stable when $n$ grows. ..."

- $\textbf{[ $1/n^2\sum_{I,j}K_T(z'_i,z'_j;S′)≤B^2$ ,see l 192]}$ This condition limits the datasets $S'$ LTK and LPK can be considered over. How does this limit the applicability/generality of results?

- $\textbf{[SGD, Section 5]}$ This section generalizes the previous results (bounds) to stochastic gradient descent. The corresponding gradient flow remains utterly deterministic, however. As opposed to previous works, where SGD is related to Langevin dynamics, [Stochastic modified equations and adaptive stochastic gradient algorithms, Li, Qianxiao, Tai, Cheng, Weinan, E, International Conference on Machine Learning, 2017], where path integrals involve higher terms due to quadratic variation induced by stochasticity. The paper neglects this. To improve the impact of the paper extended argument why deterministic flow (ODE) applies and what are assumptions needed.

**Questions:**

Q0: Figure 2c, Fig A4,d) and same for Fig. A5, d) Generalization gap is close to zero from the beginning and increasing later on (overfitting). It suggests the scale is large and thus Rademarcher upper bound is really vacuous compared to a real generalization gap (blue line), despite it improved on previous bounds as reported. Could authors rescale, redo experiments so Figure is more convincing? This is to support the strong claim of the paper mentioned on lines l43 and l55 again and l92 again” …” strong correlation (of the hereby derived upper bound) with generalization error” … and …”highly correlated” …

Q1: Could authors elaborate in detail on assumptions needed for Loss Path Kernel (LPK) $K_T(z,z')$ to be defined for arbitrary $T$? Or is $T$ limited (early stopping assumed)? See section Weaknesses for details.

Q2: The same as Q1 just for stochastic gradient descent versions in Section 5. Also see Weaknesses [SGD, Section 5].

Q3: Condition $\frac{1}{n^2}\sum_{i,j} K_T(z'_i, z'_j ; S') \leq B^2$ on the line $192$ and later similarly for SGD kernels (definition on line 242): This condition limits the datasets $S'$ LTK and LPK can be considered over. How does this limit the applicability/generality of results?

Q4: $\textbf{Gradient flow vs. Realistic NNs}$ How does the bound deteriorate when practical scenarios, e.g., finite and not infinitesimal learning rate, noise etc., are considered?

**Limitations:**

I find missing assumptions and not stated limitations to be the main weakness of the otherwise very interesting paper. See leading questions: Q1, Q2, Q3 and eventually Q4: $\textbf{Gradient flow vs. Realistic NNs}$

---

> ### Author Rebuttal · Authors · 2023-08-10
>
> We thank the reviewer for their careful reading of our paper and constructive comments!
>
> ---
> **Q1. Definition of loss path kernel and the existence of the integral.**
>
> A1. Indeed, this is a great question. Please allow us to address your concerns point-by-point below [this discussion will be added to the updated paper].
>
> - [Finite training time $T$]. Throughout this paper, we restrict our attention to a finite time $T$, where $T$ is bounded by a predetermined constant. We will mention this explicitly in the revised paper. We do not consider the asymptotic behavior of the loss function as $T \rightarrow \infty$. This is a practical setting since the training time for NNs typically has an upper limit to control computational expenses and prevent overfitting.
>
> - [Existence of gradient flow]. The existence of gradient flow (dt -> 0) and its integrals are implicitly assumed in previous NTK papers [1, 2, 3]. The gradient flow is well-defined for a wide variety of conditions on the function. For example, Lipschitz-continuity of the gradient or semi-convexity [4].
>
> - [Continuity of loss tangent kernel (LTK)]. By the assumption that the loss is continuously differentiable in line 107, the loss gradient $\nabla_w \ell(w(t), z)$ is continuous w.r.t. $w(t)$. Since $w(t)$ is differentiable (therefore continuous) w.r.t. $t$ by the gradient flow equation (Eq. (1), line 175), $\nabla_w \ell(w(t), z)$ is continuous w.r.t. $t$. After the inner product, the LTK $\bar{\mathsf{K}}(w(t); z, z')$ is still continuous w.r.t. $t$.
>
> - [Integrability of loss tangent kernel]. By the continuity of LTK on the compact set $[0, T]$, LTK is bounded and Riemann integrable on $[0, T]$. Therefore, the integral in LPK exists.
>
> In short, the existence of gradient flow and continuously differentiable loss together with finite training time $T$ are enough to guarantee the existence of LPK.
>
> As a side note, the validity of LPK as a kernel function is proved in Appendix B.1 lines 583-592.
>
> [1] Jacot, Arthur, Franck Gabriel, and Clément Hongler. Neural tangent kernel: Convergence and generalization in neural networks. Advances in neural information processing systems, 2018.
>
> [2] Du, Simon S., et al. Gradient descent provably optimizes over-parameterized neural networks. arXiv preprint arXiv:1810.02054, 2018.
>
> [3] Arora, Sanjeev, et al. On exact computation with an infinitely wide neural net. Advances in neural information processing systems, 2019.
>
> [4] Filippo Santambrogio. {Euclidean, metric, and Wasserstein} gradient flows: an overview. Bulletin of Mathematical Sciences, 2017.
>
> ---
> **Q2. Line 203 reads ..."assuming $\ell(w, z)$ is $L^2_\ell$-Lipschitz ... ". Is it needed?**
>
> A2. Corollary 1 does not require the Lipschitz continuity of the loss function. This assumption is mentioned not as a requirement, but to provide insight into the scale of the term present in Theorem 3.
>
> As a side note, continuously differentiable loss implies locally Lipschitz loss. Since $w(t): [0, T] \rightarrow \mathbb{R}^p$ is continuous on $[0, T]$, the path \{$w(t): t \in [0, T] $\} is a compact set. Then together with the loss is continuously differentiable, the loss is Lipschitz on the path $w(t)$ considered.
>
> ---
> **Q3. lines 203-208 are very vague, using terms "usually, often etc."**
>
> A3. Thank you for the suggestion! We will rephrase the sentence to eliminate the vague terms and make the statement more precise and rigorous. Please find the revised sentence below:
>
> ``The second term calculates the variation range of the LPK in the set $\mathcal{K}_T$. It will decrease with sample size $n$ as GD becomes more stable when $n$ grows [24, 8].’’
>
> ---
> **Q4 The norm constraint limits the datasets LTK and LPK can be considered over.**
>
> A4. Please refer to Q2 in the global response.
>
> ---
> **Q5. SGD remains utterly deterministic.**
>
> A5. The only assumption we have imposed is that the mini-batch indices $S_t$ are specified before the algorithm is run (see line 230–232). Under this assumption, we were able to connect the trajectory of the weights updated by SGD with a deterministic flow. It is worth noting that this assumption is commonly used in generalization theory to eliminate the randomness in the selection of mini-batches [5, 6, 7]. Our findings can be extended to the scenarios involving random mini-batch selection by first conditioning on the $S_t$ and then taking an expectation over the randomness of $S_t$. We will include the above discussion in the revised paper.
>
> [5] Neu, Gergely, et al. "Information-theoretic generalization bounds for stochastic gradient descent." Conference on Learning Theory. PMLR, 2021.
>
> [6] Wang, Ziqiao, and Yongyi Mao. "On the generalization of models trained with SGD: Information-theoretic bounds and implications." International Conference on Learning Representations, 2022.
>
> [7] Wang, Hao, Rui Gao, and Flavio P. Calmon. "Generalization Bounds for Noisy Iterative Algorithms Using Properties of Additive Noise Channels." J. Mach. Learn. Res. 24 (2023): 26-1.
>
> ---
> **Q6. Figure 2c, Fig A4,d) and same for Fig. A5, d) Generalization gap is close to zero from the beginning and increasing later on (overfitting)…**
>
> A6. To clarify, in Fig. 2(c), Fig. A4 (d), and Fig. A5 (d), the generalization gap and Rademacher upper bound are plotted in the same scale. Our generalization bound is tight with respect to the generalization gap; for example, the bound is  $\leq 0.08$ in Fig. 2(c). In contrast, existing generalization bounds are vacuous and much larger than 1 as shown in Table 3 and 4.
>
> In response to your concerns, we will make the following changes to make our statements more precise. For Figure 2(c), Fig A4 (d), and Fig. A5 (d), we will say that the bound is “tight" instead of "maintaining a strong correlation". The high correlation we meant is the correlation between Gene$(w, S)$ and the test error of different models shown in Figure 1.
>
> ---
> **Q7. Gradient flow vs. Realistic NNs**
>
> A7. Please refer to Q1 in the global response.

---

> > ### Author Response · Authors · 2023-08-13
> > **We are happy to address any remaining concerns**
> >
> > **Summary**
> > - In Q1, we have clarified the condition for the existence of the integral in LPK.
> > - In Q2, we have clarified some confusion about the explanation of the Theorem.
> > - In Q3, we will rephrase the sentence to eliminate the vague terms and make the statement more precise and rigorous.
> > - In Q4, we explained the meaning of the norm constraint and the way to eliminate it.
> > - In Q5, we clarified that deterministic SGD is a common assumption in generalization theory and our results can be extended to the random SGD case.
> > - In Q6, we clarified that the generalization gap and Rademacher upper bound are plotted in the same scale, and we will revise our statements to be more precise.
> > - In Q7, we clarified our experiments are conducted under practical scenarios and showed our results can be extended to the gradient descent with a finite learning rate setting.
> >
> > We hope our responses have addressed all your concerns. Please let us know if you have any further questions and we are happy to address them!

---

> > > ### Comment · Reviewer_kz2v · 2023-08-17
> > >
> > > Dear Authors,
> > >
> > > I appreciate your detailed response, clarifications and adjustments proposed.
> > >
> > > After reading responses and comments carefully I still think paper derives upper bound for solutions of gradient flow under additional loss smoothness assumptions. To be precise, the argument in your rebuttal (Q1, the $3^{rd}$ point, "... Since $w(t)$ is differentiable (therefore continuous) w.r.t.  by the gradient flow equation (Eq. (1), line 175) ...") stands upon the fact that it is assumed (implicitly in the paper) that $w(t)$ is solution to (Eq. (1), line 175). Then all the rest holds. But this means the paper derives upper bound for solutions of gradient flow only. This is already interesting result by all means.
> > >
> > > What I am concerned with, is the paper claims without proofs that the derived bound applies on much wider solution space, i.e., SGD or finite learning rate, which are provably deviating from gradient flow solutions, see [1,2,3]. Specifically, the Stochastic gradient descent section 5, finite learning rate or Experiments, see further comments bellow.
> > >
> > > It has been shown, cf. [1,2,3], that (discrete steps) gradient descent path diverges from gradient flow, see Fig. 1 in [1]. This gap is due to discretization error imposed by finite learning steps. In Global response (rebuttal), Q1, authors claim their analysis cover realistic scenarios including finite learning rates, using gradient flow. The same with Experiments, where relatively large learning rate together with ReLU (violating smoothness assumption on l.107) has been used.
> > >
> > > More over, the above and [1,2,3] mostly concern (full) gradient descent leaving stochastic GD alone as a more complex scenario. But again there is dedicated section 5 that claims results are valid for SGD.
> > >
> > > Overall, I bargain that derived improved bound may be the result of applying Rademacher complexity on smaller functional space, i.e., only on gradient flow solutions (as also noted by authors, in answer A1, 3rd bullet, to kz2v review), than space of realistic GD or SGD paths. In other words, real GD/SGD paths and solutions may lie out of functional space defined by solutions to gradient flow, which has been shown both theoretically and empirically to be the case in general, see [1,2,3], and derived bound does not apply.
> > >
> > > In my opinion, the current version of the paper to be acceptable, it needs to be solidified in above terms, i.e., either show that realistic training scenarios are gradient flow solutions and thus works [1,2,3] are incorrect, or retract from claiming applicability to realistic SGD scenarios, i.e., beyond the gradient flow solutions.
> > >
> > > I may be wrong, but none of the rebuttal answers did alleviate my concerns. Please do let me know, were I off in my thoughts. I'd be happy to adjust my evaluation.
> > >
> > > The following responses to rebuttal detail out the comments above:
> > > Q1: Thank you for clarifying and pointing to assumption on line 107, that the loss is continuously differentiable. However that brings back the non-applicability of the theory to popular ReLU NNs for instance as pointed by reviewer kZTv in Limitations. More over and confusingly, all experiments are conducted on ReLU NNs as per line 300 and thus violating the key assumption on line 107.
> > >
> > > Q5: Concerning the transfer from GD to stochastic GD I still have issues to understand how is continuity of the gradient w.r.t. $t$ ensured. In particular the gradient $\nabla_w {L_S}(w(t),S(t))$ is continuous w.r.t. $w(t)$ (by assumption on l. 107) yet it is not continuous w.r.t. $t$ due to different data samples (finite of size $m$ and chosen deterministically) it is evaluated upon in $[t,t+\delta t]$ steps. Thus $w(t)$ is not continuous w.r.t. $t$ even for assumption from line 230 avoiding sample randomness, i.e., $S_t$, "...the mini-batch indices are specified before the algorithm is run (see line 230–232).". In other words, solutions to SGD may lie outside of considered space of solutions to gradient flow Rademacher complexity is applied on and thus derived bound does not hold for SGD in general.
> > >
> > > As a consequence of the above, the last two bullets (arguments) in Q1 of the rebuttal do not hold and the LTK is not well defined for $\textbf{stochastic}$ GD. Or could you argue otherwise please?
> > >
> > > As written in the original review, I believe the paper brings up an interesting approach, but it should clearly state its limitations, for instance, no ReLUs and non-Lipschitz activations and clarify applicability on SGD (see above).
> > >
> > > [1] Miyagawa, T. 2023. Toward equation of motion for deep
> > > neural networks: Continuous-time gradient descent and dis-
> > > cretization error analysis. In Advances in Neural Informa-
> > > tion Processing Systems.
> > >
> > > [2] Elkabetz, Omer, and Nadav Cohen. "Continuous vs. discrete optimization of deep neural networks." Advances in Neural Information Processing Systems 34 (2021): 4947-4960.
> > >
> > > [3] Barrett, David GT, and Benoit Dherin. "Implicit gradient regularization." arXiv preprint arXiv:2009.11162 (2020).

---

> > > > ### Author Response · Authors · 2023-08-19
> > > > **Reply to Reviewer kz2v**
> > > >
> > > > Thank you for your careful reading and further comments. Please allow us to address your concerns point-by-point.
> > > >
> > > > ---
> > > > ### Main questions
> > > > ---
> > > > **Q8. [Finite learning rate] It has been shown, cf. [1,2,3], that (discrete steps) gradient descent path diverges from gradient flow, see Fig. 1 in [1]. This gap is due to discretization error imposed by finite learning steps. In Global response (rebuttal), Q1, authors claim their analysis cover realistic scenarios including finite learning rates, using gradient flow.**
> > > >
> > > > A8. First, we would like to clarify that in our submission, we never claim that our analysis covers gradient descent with finite learning rates. Instead, we wrote, “We show for the first time that the loss of NNs trained by (stochastic) *gradient flow* is equivalent to a general kernel machine.” (line 48) and “a generalization bound for NNs whose weights follow *gradient flow* in (1) at time T.” (line 188) and “We derived a generalization bound for NNs trained from full-batch *gradient flow*. Here we extend our analysis to stochastic *gradient flow* and derive a corresponding generalization bound.” (line 227).
> > > >
> > > > In our global response, we *did not* claim our analysis based on gradient flow can cover the finite learning rate case either. Rather, we connected the trajectory of gradient descent with the general kernel machine by introducing a *new* loss path kernel. This loss path kernel will become a *summation* of the loss tangent kernel instead of integration due to the discrete nature of gradient descent.
> > > >
> > > > In response to your concerns, we will explicitly outline the settings addressed in our analysis, specifically gradient flow and stochastic gradient flow, in the revised manuscript. Additionally, we will discuss the limitations of our analysis in the last section.
> > > >
> > > > ---
> > > > **Q9. [SGD results]: Concerning the transfer from GD to stochastic GD I still have issues to understand how is continuity of the gradient w.r.t. $t$ ensured …**
> > > >
> > > > A9. To clarify, for each time interval $[t, t+1]$, we assume that the same batch of data is used (please refer to lines 230-232). Within this time interval, $w(t)$ is continuous w.r.t. $t$ and we can apply Theorem 2 to establish an equivalence between NN and the general kernel
> > > > machine.
> > > >
> > > > For the next time interval $[t+1, t+2]$, even if a different data batch is used, the gradient flow ODE initializes the $w(t+1)$ with the solution from the end point of the previous interval, $[t, t+1]$. This ensures the continuity of $w(t)$ across distinct time intervals. Finally, the interval size $[t, t+1]$ is adaptable to any finite fixed length.
> > > >
> > > > In short, the continuity of $w(t)$ within each time interval, combined with the initialization of the ODE, will ensure the continuity of $w(t)$ in the entire time interval $[0, T]$.
> > > >
> > > >
> > > > ---
> > > > **Q10. Experiment questions: experiments use a relatively large learning rate and ReLU**
> > > >
> > > > A10.
> > > > - [relatively large learning rate] In experiments (I) and (II), we computed the bound for the general kernel machine (equivalently the NN trained by gradient flow). Specifically, the kernels are computed from gradient flow and used to compute the bound (lines 307-309). Importantly, under our experimental settings, we have shown that the loss of gradient flow and GD with finite learning rate are consistently close throughout the entire training process (please refer to Fig. 2(a) and Fig. A.4(a)(b), Fig. A.5(a)(b) in appendix). Given this consistent closeness between the loss of gradient flow and GD with finite learning rate, our results, although formally an upper bound for gradient flow, effectively serve as an (approximated) upper bound for NNs trained by GD with finite learning rate. We will clarify this experimental context in the revised version of our paper.
> > > >
> > > > - [ReLU] We re-ran Experiment (I) and (II) by replacing ReLU with Softplus function $\text{Softplus}(x) = \frac{1}{\beta}\ln(1 + e^{\beta x})$, which is a continuously differentiable activation function and is a smooth approximation for ReLU. We set $\beta = 10$ in the experiment. Please refer to the tables below for the new experimental results. Our observation is consistent with prior experimental results: our bound is tight w.r.t. generalization gap and can capture how noisy label influences the generalization behaviors of NNs. We will update our Experiment (I) and (II) with these new results in the revised paper.
> > > >
> > > > **Experiment (I)**
> > > >
> > > > | Training time T                                          | 10    | 100    | 1000  |
> > > > |----------------------------------------------------------|-------|--------|-------|
> > > > | generalization gap                                       | 0.002 | 0.0014 | 0.009 |
> > > > | $\hat{\mathcal{R}}^{gd}\_{\mathcal{S}}(\mathcal{G}_{T})$ | 0.003 | 0.071  | 0.028 |

---

> > > > > ### Author Response · Authors · 2023-08-19
> > > > > **Reply to Reviewer kz2v (continued)**
> > > > >
> > > > > **Experiment (II)**
> > > > > | Portion of label noise                                  | 0     | 0.2   | 0.4   | 0.6   | 0.8   | 1.0   |
> > > > > |---------------------------------------------------------|-------|-------|-------|-------|-------|-------|
> > > > > | Generalization gap                                      | 0.026 | 0.109 | 0.162 | 0.188 | 0.194 | 0.218 |
> > > > > | $\hat{\mathcal{R}}^{gd}\_{\mathcal{S}}(\mathcal{G}_{T})$ | 0.039 | 0.226 | 0.373 | 0.434 | 0.508 | 0.531 |
> > > > >
> > > > >
> > > > > In response to your concerns, we will clearly state the applicability of our analysis and conclude with a comprehensive discussion of its limitations in the final section.
> > > > >
> > > > >
> > > > > ---
> > > > > ### Other questions
> > > > >
> > > > > ---
> > > > > **Q11. Overall, I bargain that derived improved bound may be the result of applying Rademacher complexity on smaller functional space, i.e., only on gradient flow solutions (as also noted by authors, in answer A1, 3rd bullet, to kz2v review), than space of realistic GD or SGD paths. In other words, real GD/SGD paths and solutions may lie out of functional space defined by solutions to gradient flow, which has been shown both theoretically and empirically to be the case in general, see [1,2,3], and the derived bound does not apply.**
> > > > >
> > > > > A11. Indeed, you are right: the generalization bounds we provided are for neural networks trained by gradient flow (please refer to lines 188, 223, and lines 227-228, 360-361) and this function class might be different from the neural network trained with finite learning rate. Nevertheless, in Experiment (I) under the experiment settings, we have demonstrated that the loss of gradient flow and GD with finite learning rate are consistently close in training processing (please refer to Fig. 2(a) and Fig. A.4(a)(b), Fig. A.5(a)(b) in appendix). This has also been observed in [see e.g. 1, Fig. 1 and Fig. 3; 2, Fig. 2 red line and Fig. 4 green line]. As indicated by [1], “the degree of approximation depends on the curvature around the gradient flow trajectory. We then show that over deep neural networks with homogeneous activations, gradient flow trajectories enjoy favorable curvature, suggesting they are well approximated by gradient descent.” and “Experiments suggest that over simple deep neural networks, gradient descent with conventional step size is indeed close to gradient flow.”
> > > > >
> > > > >
> > > > > We demonstrated the improvement of our bound in comparison to previous bounds in Table 3 and Table 4. We achieved a tighter bound because our generalization theory takes into account the training trajectory decided by the optimization method, resulting in a smaller function class. In contrast, previous VC-dimension-based and norm-based bounds do not consider the optimization algorithms used for training neural networks (NNs).
> > > > >
> > > > >
> > > > >
> > > > > [1] Elkabetz, Omer, and Nadav Cohen. "Continuous vs. discrete optimization of deep neural networks." Advances in Neural Information Processing Systems 34 (2021): 4947-4960.
> > > > >
> > > > > [2] Miyagawa, T. 2023. Toward equation of motion for deep neural networks: Continuous-time gradient descent and dis- cretization error analysis. In Advances in Neural Information Processing Systems.
> > > > >
> > > > > ---
> > > > > **Q12. As written in the original review, I believe the paper brings up an interesting approach, but it should clearly state its limitations, for instance, no ReLUs and non-Lipschitz activations and clarify applicability on SGD (see above).**
> > > > >
> > > > > A12. Thank you for your suggestion! In response to your concerns, we will explicitly state the applicability of our theory (smooth activation functions and NN trained by gradient flow) both after Theorem 3, 5, and in the last section. We will also add the clarification about SGD (see our A.9) to Section 5.
> > > > >
> > > > >
> > > > > ---
> > > > > We hope our responses have addressed all your concerns. Please let us know if you have any additional questions or suggestions that can help us further improve our paper! Thank you!

---

> > > > > ### Comment · Reviewer_kz2v · 2023-08-20
> > > > >
> > > > > Thank you for your responses.
> > > > >
> > > > > Regarding A8, "...we never claim that our analysis covers gradient descent with finite learning rates..." and "... In our global response, we did not claim our analysis based on gradient flow can cover the finite learning rate case either. ...". But read your second sentence of the Abstract: "... In this paper, we study the generalization capability of neural networks trained with (stochastic) gradient descent." It seems to me as a reader that you are suggesting gradient flow describes "neural networks trained (stochastic) gradient descent", which we have already agreed is not true \textbf{in general}, for instance see [1,2,3]. Hence my previous comments and I still insist your paper claims more than it should and thorough revision is required.
> > > > >
> > > > > A9. Ok, thank you. But by this perspective we have different (arbitrary large, since there is no assumption on loss w.r.t. training data made) losses for every batch, i.e., [t,t+1] and possibly unbounded, in general. To solve this we need to consider only training sets with constrained RKHS norms (with bounds $B_t$) as you do in section 5 and these upper bounds are present $B_t$ as they should. In other words complexity introduced by discontinuity of loss (due to evaluation on different batches) is now covered in $B_t$, correct? If so, this makes bound being dependent also on specific batch selection regime, maybe worth mentioning?
> > > > > Overall thank you for clarification, I admit stochastic gradient FLOW is covered.
> > > > >
> > > > > In this light, I suggest renaming section 5 to stock. grad. flow along the general comments above that finite step size is provably out of "flow" regime.
> > > > >
> > > > > To all other answers, great and thank you.
> > > > >
> > > > > So if authors do thorough revision of wording rephrasing "descent" for "flow" retracting to claims covered (theoretically) practical scenarios (finite learning rates) throughout the paper, I will change score to "borderline accept".

---

> > > > > > ### Author Response · Authors · 2023-08-21
> > > > > > **Reply to Reviewer kz2v**
> > > > > >
> > > > > > Thank you for your follow-up. We are glad to learn that our response has addressed your main concerns.
> > > > > >
> > > > > > ---
> > > > > > **Q13. Rephrasing "descent" with "flow".**
> > > > > >
> > > > > > A13. Thank you for your suggestion! In response to your concern, we will replace “descent” with “flow” in the abstract, 3rd paragraph of the introduction, and the titles of Sections 4 and 5 in the updated paper.
> > > > > >
> > > > > > In our experiment, we will clearly state the experimental setup and observations we had (the loss of gradient flow and GD with finite learning rate are consistently close throughout the entire training process in Fig. 2(a)). We will also include a discussion about the limitations of our analysis in the last section.
> > > > > >
> > > > > > ---
> > > > > > **Q14. Different (arbitrary large, since there is no assumption on loss w.r.t. training data made) losses for every batch and possibly unbounded. The bound of stochastic gradient flow is dependent on the specific batch selection regime.**
> > > > > >
> > > > > > A14. Indeed, given that we assume the mini-batch indices are given (line 231), the bound of the stochastic gradient flow depends on the batch selection---we will clarify this in the revised version of the paper. Nevertheless, the choice of batch selection can be general, as the loss path kernel $\mathsf{K}\_{t, t+1}(z_i', z_j'; \mathcal{S}')$ can be bounded with the local Lipschitz constant of the loss function (please refer to our A2) and, consequently, all $\mathcal{S}’$ will have bounded RKHS norm $\frac{1}{m^2} \sum_{i, j \in \mathcal{S}_t} \mathsf{K}\_{t, t+1}(z_i', z_j'; \mathcal{S}')$. Additionally, we made the assumption that the loss function $\ell(w, z)$ is bounded in $[0, 1]$ (line 107) to prevent the training loss from blowing up.
> > > > > >
> > > > > >
> > > > > > ---
> > > > > > We will make sure to include the promised changes in the revised paper. Thank you again for your careful reading of our paper and constructive feedback. If our responses and proposed adjustments addressed your concerns, please kindly reconsider your evaluation!

---

> > > > > > > ### Comment · Reviewer_kz2v · 2023-08-21
> > > > > > >
> > > > > > > Ok, deal :-). I've increased my evaluation to 5. Thank you for your responses and overall discussion. Appreciated.

---

> > > > > > > > ### Author Response · Authors · 2023-08-21
> > > > > > > > **Thank you for raising the score!**
> > > > > > > >
> > > > > > > > Thank you for raising the score! We really appreciate the insightful feedback and your engagement in discussion with us!

---

### Official Review · Reviewer_W7Wt · 2023-06-29

**Soundness:** 4 excellent
**Presentation:** 3 good
**Contribution:** 4 excellent
**Rating:** 8
**Confidence:** 3

**Summary:**

The paper proposes a new complexity measure for neural networks based on tracking the changes of the weight vector of the entire model.  The work is rooted in theory by linking the proposed measure through the Neural Tangent Kernel framework to Rademacher complexity resulting in a new, meaningfully tight bounds on generalisation.


**Strengths:**

Explanation of the success of overparametrised neural network models using statistical learning theory is still very important problem we haven't solved, and this works seems like a significant development in that space.

The proposed complexity measure is linked theoretically to a generalisation bound.

This looks like a significant amount work, extending and generalising existing theory in the Neural Tangent Kernel framework.

Empirical results, though shown on small examples, show that the derived generalisation bound is meaningfully tight (i.e. non-vacuous).

For the most part (with some exceptions mentioned below) the paper is very well written, and guides the reader well through very complicated subject matter.

**Weaknesses:**

The maths are very dense...and the 32 page supplement, while admirable, is way too much to get through in the time given for this reviewing period.  I didn't get through the maths, so I couldn't verify it.  It almost feels like this should be a journal rather than a conference paper.

I did get lost a little bit in the maths of the main part of the paper.  It should be possible to follow this paper at high level, taking the proofs of theorems at face value, and for the most part authors do a great job of guiding the reader....but still, there are some things (see questions below) that I found confusing.

The proposed method is computationally very expensive, and so not that easy to use in practice...but approximations and computational improvements might come in the future, and the theory (assuming the proofs (which I couldn't verify) are solid) is sound and of great interest.

**Questions:**

As already stated, I didn't follow all the maths, but I wanted to verify my high level understanding of the work.  It seems to me, that the fundamental principle here is that while the hypothesis space of the network is immensely complex (hence very high VC dimension), that during training there is only a small subset of hypotheses "visited"...and for the purpose of generalisation guarantees, what matters is the set of hypotheses "considered" (when training), not the set of hypotheses available.  And the proposed loss path kernel measure is a measure of the complexity of the hypothesis space "visited".  Is this a fair high level summary?

I am completely lost at what $\mu^{\otimes n}$ is - is it supposed to be obvious from the context?  And it all gets so complicated by Theorem 5...that I can't really tell from the math how tight the bound is.  Or, since this will always depend on the training path, the tightness of the bound can only be assessed empirically?

---

> ### Author Rebuttal · Authors · 2023-08-09
>
> We thank the reviewer for the kind comments and positive feedback!
>
> ---------
> **Q1. The proposed method is computationally very expensive, and so not that easy to use in practice**
>
> A1. Thank you for raising the question about the computational cost of our method. For small models, our method is not computationally expensive. In Experiment (I), estimating the bound with 20 $S’$ (solving 20 gradient flow ODE) costs 500s and training NN costs 0.29s. The GPU memory required by estimating the bound is 2406MB and training NN requires 1044MB.
>
> For large models, the computational cost of exactly calculating the bound might be expensive but the approximation we’ve proposed for NAS (i.e., $U_{sgd}$ in Eq. (4)) is computationally efficient. In the table below, for Experiment (III), we report the averaged computational cost (GPU hours) of our approach for one architecture and the computational cost of training one NN architecture to convergence. Note our approach calculates Gene$(w, S)$ only after training for 1 or 2 epochs, leading to significant savings in computational time.
>
> | GPU hours                                   | CIFAR-10 | CIFAR-100 |
> |---------------------------------------------|----------|-----------|
> | RS + Gene$(w, S)_2$ (Ours)                  | 0.036    | 0.037     |
> | Training one NN architecture to convergence | 1.83     | 2.56      |
>
> We will include the above results in the revised paper.
>
>
> ---------
> **Q2. Verify high-level understanding. It seems to me, that the fundamental principle here is that while the hypothesis space of the network is immensely complex (hence very high VC dimension), that during training there is only a small subset of hypotheses "visited"...and for the purpose of generalisation guarantees, what matters is the set of hypotheses "considered" (when training), not the set of hypotheses available. And the proposed loss path kernel measure is a measure of the complexity of the hypothesis space "visited". Is this a fair high level summary?**
>
> A2. We appreciate your careful read of our paper and, yes, this summary nicely describes our work! We will include the high-level summary and intuition in the revision for clarity. Thank you again!
>
>
> ---------
> **Q3. Lost at what $\mu^{\otimes n}$ is. Can't really tell from the math how tight the bound is. Or, since this will always depend on the training path, the tightness of the bound can only be assessed empirically?**
>
> A3.
> - $\mu^{\otimes n}$ is the joint probability distribution of $n$ i.i.d. samples drawn from $\mu$.
> - The tightness of our bound can be partly seen from the lower bound of the Rademacher complexity in Appendix B.3. Our lower bound matches the trace term in the upper bound $U_1$, which shows the bound is relatively tight. We also conducted experiments to demonstrate the tightness of our bounds, a standard practice adopted in generalization theory literature [e.g., 1, 2, 3].
>
> [1] Dziugaite, Gintare Karolina, and Daniel M. Roy. "Computing nonvacuous generalization bounds for deep (stochastic) neural networks with many more parameters than training data." arXiv preprint arXiv:1703.11008 (2017).
>
> [2] Zhou, Wenda, et al. "Non-vacuous generalization bounds at the imagenet scale: a PAC-bayesian compression approach." arXiv preprint arXiv:1804.05862 (2018).
>
> [3] Jiang, Yiding, et al. "Fantastic generalization measures and where to find them." The International Conference on Learning Representations, 2020.
>
> ---------
> **Summary**
> - In Q1, we have clarified that our method is computationally efficient for small models, and for large models, the approximation we proposed for the NAS is computationally efficient.
> - In Q2, we acknowledged the reviewer’s high-level summary of the paper.
> - In Q3, we have explained the meaning of $\mu^{\otimes n}$ and the tightness of our bound.
>
> We hope our responses have addressed all your concerns. Please let us know if you have any further questions!

---

> > ### Comment · Reviewer_W7Wt · 2023-08-17
> > **Thanks for your reply**
> >
> > Thank you for your reply - glad to know my high-level understanding of the presented concepts was not completely wrong.  I am quite happy to stick with my strong recommendation to accept.

---

> > > ### Author Response · Authors · 2023-08-19
> > > **Thanks for your recommendation!**
> > >
> > > Absolutely, it was a clear summary of our work! Thank you once again for your insightful response and the constructive feedback provided in your initial review. We will make sure to include the promised changes in the revised paper.

---

### Official Review · Reviewer_kZTv · 2023-07-03

**Soundness:** 3 good
**Presentation:** 2 fair
**Contribution:** 3 good
**Rating:** 7
**Confidence:** 3

**Summary:**

This paper introduces a new generalization bound based on dynamic NTK, called loss path kernel in the paper. The loss path kernel is a kernel based on the integration of a loss tangent kernel (NTK with loss function) so that the generalization bound can be determined by the training dataset and training trajectory of parameters. Theoretically, the paper provides the loss path kernel generalization bound guarantee and beats their bound over previous work. On the other hand, empirically, the authors verify their theorem with numerical experiments and plug their loss path kernel into NAS and get good results.

**Strengths:**

- The paper studies a very important question of the generalization gap of general neural networks. The definition of the loss path kernel is intuitive and straightforward. From my perspective, the new generalization bound based on the loss path kernel has its value. It considers the training trajectory in the bound rather than calculating the whole function hypothesis class so that the generalization bound can be tighter.

- The paper provides numerical experiments to support their theorem. The numerical simulation and implementation in NAS show that their analysis can be used in practice and their theorem is not vacuous.


**Weaknesses:**

- The empirical part of writing is not clear enough. I cannot get the message between Theorem 3 and Figure 2 (c). Also, Figure 1 looks strange. The paper does not provide a convincing explanation for the outlier. The implementation of the NAS part is missing, e.g., the search space. The best architecture was not reported.

- The NAS part can be stronger. The paper only considers the random sampling of 100 architectures. However, in practice, we do not use random sampling in NAS anymore because it is not efficient. There are many dynamic sampling ways in NAS, e.g., [1]. The paper may improve its results by using these methods.

[1] Guo, Zichao, Xiangyu Zhang, Haoyuan Mu, Wen Heng, Zechun Liu, Yichen Wei, and Jian Sun. "Single path one-shot neural architecture search with uniform sampling." In ECCV 2020.


**Questions:**

- Why do we need Definition 2? It seems that the analysis did not use the general kernel machine.
- The loss path kernel needs to calculate integration, in GD or SGD. I wonder how to calculate them in practice, e.g., NAS, by math formulation or sampling.
- The paper only considers gradient flow. Can we extend the results to normal gradient descent with a constant learning rate?


**Limitations:**

It seems that the analysis does not hold for the ReLU network as it is not continuously differentiable.

======

Change score from 6 to 7 on Aug 10.

---

> ### Author Rebuttal · Authors · 2023-08-09
>
> We thank the reviewer for the positive feedback, thoughtful comments, and for appreciating the novelty and value of the work!
>
> ---
> **Q1. The empirical part of writing is not clear enough. I cannot get the message between Theorem 3 and Figure 2 (c). Also, Figure 1 looks strange. The paper does not provide a convincing explanation for the outlier. The implementation of the NAS part is missing, e.g., the search space. The best architecture was not reported.**
>
> A1. Thank you for the questions and suggestions! Please allow us to clarify them point by point.
>
> - As shown in Corollary 1, $\hat{\mathcal{R}}^{gd}\_{\mathcal{S}}(\mathcal{G}\_T)$ is an upper bound of the generalization gap $L_\mu(w_T) - L_S(w_T)$. In Figure 2 (c), we plot both $\hat{\mathcal{R}}^{gd}\_{\mathcal{S}}(\mathcal{G}\_T)$ and the generalization gap to demonstrate that $\hat{\mathcal{R}}^{gd}\_{\mathcal{S}}(\mathcal{G}\_T)$ is a tight upper bound of the generalization gap.
>
> - Cause of the outlier: $U_{sgd}$ is calculated from the loss gradients along the training trajectory. NAS-Bench-201 is a NAS benchmark that contains 15625 NN architectures. When we randomly sample 100 architectures, there is a chance that we will get some “not-so-good” architectures, which have large loss gradients during training and cause a large $U_{sgd}$.
>
> - The implementation of our NAS algorithm is detailed in lines 336-357. The search space is NAS-Bench-201 [1], as indicated in line 275 and Table 2. We follow the same setting in this line of work (TENAS, LGA) for fair comparison. We will report the best architecture searched by our algorithm in the revised paper.
>
> [1] Dong, X. and Yang, Y. Nas-bench-201: Extending the scope of reproducible neural architecture search. arXiv preprint arXiv:2001.00326, 2020.
>
> ---
> **Q2. The NAS part can be stronger with dynamic sampling**
>
> A2. Thank you for pointing out the reference and for your valuable suggestion. We will incorporate it into the updated paper and will also attempt to reproduce our results using dynamic sampling. Additionally, we would like to highlight that the performance of our approach (93.79 for CIFAR-10) is already close to the “optimal” (94.37 for CIFAR-10) in Experiment (III) Table 2, where the “optimal” means the best test accuracy achievable in the NAS-Bench-201 search space.
>
> ---
> **Q3. Why do we need Definition 2? It seems that the analysis did not use the general kernel machine.**
>
> A3. We include Definition 2 for ease of reference. We use it in Theorem 2 and Theorem 4 whose results are either a general kernel machine or a sum of general kernel machines.
>
> ---
> **Q4. How to calculate LPK in practice, e.g., NAS, by math formulation or sampling.**
>
> A4. For small models as those in Experiments (I) and (II), we can solve the gradient flow equation (Eq. (1), line 175) to get the model parameters and calculate the LTK accordingly. Then calculating the LPK just requires solving another ODE with the calculated LTK. These ODEs can be computed with torchdiffeq package.
>
> For large models as in NAS Experiment (III), solving the gradient flow ODE is computationally infeasible. As explained in line 276-278, we applied a trapezoidal rule to approximate the integration, where  $\mathsf{K}\_{t, t+1}(z, z') = \int_t^{t+1} \left\langle \nabla_{w} \ell(w(s), z), \nabla_{w} \ell(w(s), z') \right\rangle ds$ is approximated by $\frac{\eta}{2}[\left\langle \nabla_{w} \ell(w_{t}, z), \nabla_{w} \ell(w_{t}, z') \right\rangle + \left\langle \nabla_{w} \ell(w_{t+1}, z), \nabla_{w} \ell(w_{t+1}, z') \right\rangle]$.
>
> ---
> **Q5. Can we extend the results to normal gradient descent with a constant learning rate?**
>
> A5. This is a great question! Please refer to Q1 in the global response, where we have clarified that our experiments are conducted under practical scenarios and showed our results can be extended to the gradient descent with a finite learning rate setting.
>
>
> ---
> **Q6. The analysis does not hold for the ReLU network as it is not continuously differentiable.**
>
> A6. Our assumption of continuously differentiable NN is mainly for the loss path kernel to be a valid kernel. As a valid kernel, it needs to be continuous for its input (See Proposition 1 and lines 583-589). We believe with a finer analysis, our theory could be extended to non-smooth cases such as ReLU NNs. For example, finite input space does not have the continuity requirement according to Proposition 1. Another approach is writing the LPK explicitly as an inner product of feature mappings according to the definition of the kernel to verify its validity.
>
> Finally, we would like to highlight that our theory is very general as it holds for any continuously differentiable neural networks. In contrast, many prior work's analyses are tailored to ReLU networks and have other requirements, e.g. ultra-wide models and specific loss functions [2, 3].
>
> [2] Arora, S., Du, S., Hu, W., Li, Z., and Wang, R. Fine-grained analysis of optimization and
> generalization for overparameterized two-layer neural networks. In International Conference on Machine Learning, pp. 322–332. PMLR, 2019.
>
> [3] Cao, Y. and Gu, Q. Generalization bounds of stochastic gradient descent for wide and deep neural networks. Advances in neural information processing systems, 32, 2019.
>
> ---
> **Summary**
> - In Q1, we clarified some confusion about our experiments.
> - In Q2, we will incorporate dynamic sampling to further improve the NAS algorithm in the revised paper.
> - In Q3, we clarified that Definition 2 is for ease of reference.
> - In Q4, we explained how to calculate LPK in practice.
> - In Q5, we clarified that our experiments are conducted under practical scenarios and showed our results can be extended to the gradient descent with a finite learning rate setting.
> - In Q6, we explained how to extend our theory to ReLU NNs.
>
> We hope our responses have addressed all your concerns. Please let us know if you have any further questions!

---

> > ### Comment · Reviewer_kZTv · 2023-08-11
> > **Increasing my score from 6 to 7**
> >
> > Thank you for the rebuttal. The rebuttal well-solved my questions. I have read all reviewers' comments and responses. I'd like to increase my score from 6 to 7.
> >
> > Three items to make the paper stronger. (1) Put the gradient descent with a finite learning rate analysis in the main body. (2) The NAS experiments can be more thorough and well-explained. (3) The paper only considers infinite-width NN as a theoretical application. It would be good to provide a finite-width 2-layer NN trained under some well-defined statistical learning problems, e.g., Mixture of Gaussians, Parity function, as a case study. This may provide more theoretical insights and comparisons with previous work.

---

> > > ### Author Response · Authors · 2023-08-11
> > > **Thank you! We are glad to hear that your questions get resolved.**
> > >
> > > Thank you for your prompt reply and raising the score! We’re glad to know that our rebuttal has addressed your concerns. We also appreciate the further insights you’ve provided. We will enrich the main body with a discussion about extending our theory to finite learning rate analysis. We will also include a more detailed discussion of the NAS experiments, and try to apply our theory to other theoretical applications.

---

### Official Review · Reviewer_Zsen · 2023-07-07

**Soundness:** 4 excellent
**Presentation:** 4 excellent
**Contribution:** 2 fair
**Rating:** 7
**Confidence:** 3

**Summary:**

The submission provides data-and-architecture-dependent Rademacher complexity generalization bounds for neural networks. Unlike previous work, the approach takes the evolution of the neural tangent kernel under gradient flow into account. First gradient flow under an evolving kernel is expressed as learning with a general kernel machine on a kernel named Loss Path Kernel. Second, a data-dependent complexity measure of the general kernel machine is computed that is finally used to compute the generalization bound.

**Strengths:**

The approach is novel as far as I know, although the literature on this topic is vast and it is hard to assess novelty. While computing the bound requires access to the true data distribution and training the network, a heuristic approximation of it from the training data and a few epochs of training is shown to correlate with performance in the experiments. Background, relevant work, and the proposed method are brilliantly presented. I did not notice any glaring issues in the math although checking the long proofs in appendix in detail is not possible given the review workload. Altogether I'm leaning towards acceptance.

**Weaknesses:**

The major weaknesses of this work are the limited intuition it provides, an issue with the presentation of the experiments, and possibly high computation required to estimate the bound. I will elaborate on these points below. The second and the third point are critical and I ask the authors to improve them in the revision.

W1. The results abstract away the details of training into a black-box evolving NTK and, although this improves the domain of applicability of the result, the obtained result does not give any insights about the role depth, width, or other choices of architecture on generalization. If theoretical connections between these choices and the evolution of NTK are known in the literature, this paper can be improved by theoretically studying the role of different architectural choices on the loss path kernel and its complexity.

W2. The main motivation for the bound is that it is not limited to infinite-width or single output neural networks. A brief subsection 6.1 right before the experiments focuses on infinite-width neural networks to avoid the dependency on training. The subsection and its placement in the paper confused me about the following experiments. It seems like the experiments afterwards consider a finite-width neural network. If this is the case, I do not understand why this subsection is placed right before them and in the same section.

W3. The experiments allude to intractability of a basic procedure to estimate the bound and computational speedups from approximation. The revision should report the computation or provide measures of complexity or at least clarify in the paper that estimating the bound is not as expensive as training the network. Otherwise one could perform architecture search by trying one or a few train/validation splits.

Minor comment (did not influence score):

i. If the Lipschitz constant is local, it can be influenced by the data or the trajectory of optimization in ways that are not captured in the analysis. This comment did not influence the score as it applies to previous work (Arora et al) as well and I believe fixing them is outside the scope of this work.

ii. Practical neural networks are typically trained with relatively large learning rates and especially the early evolution of NTK under large learning rate and gradient flow are known to be very different. This limits the significance of a study on gradient flow.

------------
Post-rebuttal: Raised the score to 7 since the revision will address W2 and W3.

**Questions:**

See weaknesses

**Limitations:**

See weaknesses

---

> ### Author Rebuttal · Authors · 2023-08-09
>
> We thank the reviewer for the positive feedback, thoughtful review, and for appreciating the merits of the work!
>
> ---
> **Q1. No insights about the role of depth, width, or other choices of architecture on generalization.**
>
> A1. Thank you for the insightful question! Please refer to Appendix D.3, where we have derived a norm-based bound by extending our analyses. Prior work has generally observed that wider neural networks tend to have smaller weight norms [see e.g., 1]. Therefore, our norm-based bound is expected to be smaller for such NNs. Nevertheless, we acknowledge that the exact influence of depth and width on our bound remains an active area of research that requires more exploration.
>
> Besides, our bound can be used to study the influence of width and depth empirically. We have conducted an additional experiment in the rebuttal to study the influence of width on our bound. With the same setting of Experiment (I), we train and compute our bound for NNs with widths 100 and 1000 at $T=10000$ (convergence). The 1000-width NN has a smaller bound of 0.024 compared with the 100-width NN whose bound is 0.032. This shows wider NN tends to have smaller values of our bounds and better generalization ability.
>
> Beyond depth and width, our bounds offer insights into how learning algorithms impact the generalization capabilities of neural networks. For instance, the bound $U_1$ presented in Theorem 3 suggests that the generalization gap could be influenced by the local Lipschitz constant along the training trajectory. This is because the loss path kernel can be upper bounded by the local Lipschitz constant. Meanwhile, $U_2$ indicates that generalization is contingent upon the variations in loss when trained using different data.
>
> Finally, we'd like to highlight that our results are general and can be applied to delve into various specific cases. As an illustration, we explored stable algorithms in Appendix D.2 and norm-constrained NNs in Appendix D.3. These investigations shed light on the roles of the Lipschitz constant, smoothness constant, and weight norm in neural network generalization — pivotal elements in generalization theory.
>
> We will include the above discussion in the revision, which we believe will help the readers to gain more insights on our results. Thank you for the question!
>
> [1] Neyshabur, B., Li, Z., Bhojanapalli, S., LeCun, Y., and Srebro, N. The role of over-parametrization in generalization of neural networks. In International Conference on Learning Representations, 2019.
>
> ---
> **Q2. Confusion about why subsection 6.1 of infinite-width NN is placed right before subsection 6.2 of NAS**
>
> A2. Subsection 6.1 of infinite-width NN is an application of our results in theory and subsection 6.2 of NAS is an empirical application. We include subsection 6.1 to apply our bound to infinite-width NN so that we can compare it with existing generalization bounds that are tailored to infinite-width NNs.
>
> In response to your concerns, we will move this subsection to Appendix D, where we have discussed other applications of our generalization bounds to study stable algorithms and derive norm-based generalization bounds.
>
> ---
> **Q3. Computation time of estimating the bound**
>
> A3. Thanks for your suggestion! In Experiment (I), estimating the bound with 20 $S’$ (solving 20 gradient flow ODE) costs 500s and training NN costs 0.29s. The GPU memory required by estimating the bound is 2406MB and training NN requires 1044MB.
>
> For experiment (III), we report the averaged computational cost (GPU hours) of our approach for one architecture and the computational cost of training one NN architecture to convergence in the below table. Note our approach calculates Gene$(w, S)$ only after training for 1 or 2 epochs, which saves computational cost a lot.
>
> | GPU hours                                   | CIFAR-10 | CIFAR-100 |
> |---------------------------------------------|----------|-----------|
> | RS + Gene$(w, S)_2$ (Ours)                  | 0.036    | 0.037     |
> | Training one NN architecture to convergence | 1.83     | 2.56      |
>
> We will report the above results in the revised paper.
>
> ---
> **Q4. Local Lipschitz constant can be influenced by the data or the trajectory of optimization in ways that are not captured in the analysis.**
>
> A4. We thank the reviewer for this insightful point. Indeed, the local Lipschitz constant can be influenced by the training data. To clarify, the original formulation of our bounds does not directly depend on the local Lipschitz constant. The loss tangent kernel in our bounds captures the influence of training data and the trajectory of parameters in optimization to a certain extent. Nonetheless, we do agree with the reviewer that a thorough exploration of the relationship between our bounds and the local Lipschitz constant is beyond the scope of this work. We will include a discussion in the revision.
>
> ---
> **Q5. Practical neural networks are typically trained with relatively large learning rates …**
>
> A5. Please refer to Q1 in the global response, where we have clarified that our experiments are conducted under practical scenarios and showed our results can be extended to the gradient descent with a finite learning rate setting.
>
> ---
> We hope our responses have addressed all your concerns. Please let us know if you have any further questions!

---

> > ### Comment · Reviewer_Zsen · 2023-08-14
> >
> > Thank you for the response. It addresses my comments and I will raise the score to 7.

---

> > > ### Author Response · Authors · 2023-08-15
> > > **Thank you for your response!**
> > >
> > > Thank you so much for your response and raising the score! We are glad to know that we have addressed your concerns. We will make sure to include the promised changes in the revision.

---

### Author Rebuttal · Authors · 2023-08-10

### **Global Response**

We would like to thank all the reviewers for taking the time and effort to review our paper! We are delighted to learn that our paper was positively received, and the reviewers found that:
 - the background, relevant work, and the proposed method are brilliantly presented (Reviewer Zsen);
- the new generalization bound based on the loss path kernel has its value and the numerical simulation and implementation in NAS show that their analysis can be used in practice and their theorem is not vacuous (Reviewer kZTv);
- this looks like a significant amount work, extending and generalising existing theory in the Neural Tangent Kernel framework (Reviewer W7Wt);
- and paper follows an interesting idea of generalising tangent kernels to path integrals and it is timely and potentially interesting contribution (Reviewer kz2v).

We also recognize that the reviewers are busy handling multiple papers, so their thoughtful feedback is even more appreciated.

Below, we provide an answer to a common question shared by Reviewer Zsen, kZTv, and kz2v in this global response. We also address the concerns and questions raised by each reviewer and detail our plans for updating the paper. We will add the changes in the revision (both in the main text and appendix).

Please don’t hesitate to let us know if you have any additional feedback or questions regarding our response. We would be happy to address any remaining concerns with you in the discussion period if any. If our responses have addressed your concerns and questions, we would appreciate it if you could kindly let us know and consider raising your review score.

Thanks for your time and review!

---
**Q1. Extension to practical gradient descent with a finite learning rate.**

A1. We thank the reviewers for this important question. First, we would like to highlight that all our experiments have been conducted under practical scenarios. Specifically, in Experiments (I) and (II) (Section 7, line 306), we trained NNs with a relatively large learning rate (lr=10 for NTK parameterization), while the gradient flow and gradient descent overlapped well (see Fig. 2 (a)) and the bounds were non-vacuous (Fig. 2 (b) and (c), Fig. 3).

Following the reviewer’s suggestion, we show below that it is possible to extend our results to gradient descent with a finite learning rate setting. In this case, the loss path kernel will be defined as a summation instead of an integration over training trajectory as follows,

$$\mathsf{K}\_T(z, z') = \sum_{t=0}^{T-1}\left\langle \nabla_{w} \ell(w_{t}, z), \nabla_{w} \ell(w_{t}, z') \right\rangle.$$

Under the assumption that the loss $\ell(w, z)$ is $\beta_\ell$-smooth and convex, we can get a similar bound as the gradient flow one with an additional term that involves the learning rate $\eta$ and smoothness constant $\beta_\ell$:

$$\mathcal{E}(\mathcal{S}, \eta, T) = \frac{\beta_\ell \eta^2}{2 n^2} \sum_{i = 1}^{n}\sum_{j = 1}^{n} \mathsf{K}_T(w_t; z_i, z_j).$$

When $\eta \rightarrow 0$, $\mathcal{E}(\mathcal{S}, \eta, T) \rightarrow 0$.


---
**Q2 The norm constraint [$\frac{1}{n^2} \sum_{i, j} \mathsf{K}_T(z_i', z_j';\mathcal{S}') \leq B^2$, line 192] limits the datasets $\mathcal{S}'$ LTK and LPK can be considered over. How does this limit the applicability/generality of results?**

A2. This norm constraint balances a tradeoff between the tightness of the bound and the expressiveness of the set $\mathcal{G}\_T$ (that is, the number of datasets $\mathcal{S}'$ over which LTK and LPK can be applied to). A small $B$ results in a tighter bound, whereas a large $B$ allows for more datasets to be covered. In an extreme case, we can choose $B^2 = \sup_{\mathcal{S}'} \frac{1}{n^2} \sum_{i, j} \mathsf{K}_T(z_i', z_j';\mathcal{S}') $ to encompass all possible $\mathcal{S}'$.

Finally, it is worth noting that similar kinds of norm constraints have appeared in previous norm-based bounds [1, 2, 3, 4]. By applying a method akin to that found in [2, Lemma A.9], one can cover the parameter space and use a union bound to eliminate this norm constraint.

[1] Bartlett, P. L. and Mendelson, S. Rademacher and gaussian complexities: Risk bounds and structural results. Journal of Machine Learning Research, 3(Nov):463–482, 2002.

[2] Bartlett, P. L., Foster, D. J., and Telgarsky, M. J. Spectrally-normalized margin bounds for neural networks. Advances in neural information processing systems, 30, 2017.

[3] Neyshabur, Behnam, Ryota Tomioka, and Nathan Srebro. "Norm-based capacity control in neural networks." Conference on learning theory. PMLR, 2015.

[4] Neyshabur, B., Li, Z., Bhojanapalli, S., LeCun, Y., and Srebro, N. The role of over-parameterization in generalization of neural networks. In International Conference on Learning Representations, 2019.

---

### Decision · Program_Chairs · 2023-09-21

**Decision:**

Accept (poster)

**Comment:**

The authors propose a new kernel that is used for analyzing the generalization behavior of neural networks trained by gradient flow, called the loss path kernel.  They introduce a data- and path-dependent generalization bound based on this kernel with a Rademacher-complexity based analysis.  They show that there is relatively large correlation between the proposed generalization bound and observed generalization bounds in (gradient descent-) trained neural networks in some settings.

The reviewers were largely supportive of this work.  One reviewer's main concern was the connection between the analysis for gradient flow and the applicability to finite-step gradient descent, but the authors' rebuttal largely addressed this concern.

The other 3 reviewers expressed support for the novelty of the approach of a path-based generalization bound.  I concur with these reviewers and think this novel approach is worth publication and further investigation; I recommend acceptance.